# ARE TRANSFORMERS ABLE TO REASON BY CONNECTING SEPARATED KNOWLEDGE IN TRAINING DATA?

**Yutong Yin**[*]  **Zhaoran Wang**[*]

## ABSTRACT

Humans exhibit remarkable compositional reasoning by integrating knowledge from various sources. For example, if someone learns ( B = f(A) ) from one source and ( C = g(B) ) from another, they can deduce ( C=g(B)=g(f(A)) ) even without encountering ( ABC ) together, showcasing the generalization ability of human intelligence. In this paper, we introduce a synthetic learning task, "FTCT" (Fragmented at Training, Chained at Testing), to validate the potential of Transformers in replicating this skill and interpret its inner mechanism[1]. During training, data consist of separated knowledge fragments from an overall causal graph. In testing, Transformers must combine these fragments to infer complete causal traces. Our findings demonstrate that few-shot Chain-of-Thought prompting enables Transformers to perform compositional reasoning on FTCT by revealing correct combinations of fragments, even if such combinations were absent in training data. Furthermore, the emergence of compositional reasoning ability is strongly correlated with model complexity and training-testing data similarity. We propose, both theoretically and empirically, that Transformers learn an underlying generalizable program from training, enabling effective compositional reasoning during testing.

## 1 INTRODUCTION

Humans exhibit a generalized reasoning ability that integrates knowledge from diverse sources. For example, if one learns ( B = f(A) ) from one source and ( C = g(B) ) from another, they can deduce ( C = g(B)=g(f(A)) ) without direct exposure to ( ABC ). We formally define this capability as compositional reasoning—the skill to integrate discrete pieces of knowledge from multiple sources to form a coherent reasoning, even in the absence of explicit examples connecting these pieces during learning. This ability is a manifestation of a broader concept known as systematic compositionality—understanding and generating an infinite number of expressions by combining a finite set of known components and rules (Fodor & Pylyshyn, 1988; Chomsky, 2002). Transformer-based large language models demonstrate signs of compositional reasoning by producing comprehensive content that includes elements not likely to co-occur within the training data, suggesting the emergence of general intelligence (Press et al., 2022; Zhou et al., 2022a; Bubeck et al., 2023). However, the complexity and ambiguity of their natural language training and testing data make it hard to scientifically validate the compositional reasoning ability and explore the underlying mechanisms.

This paper validates the potential of Transformers in doing compositional reasoning on synthetic dataset and investigates the inner mechanisms eliciting such ability. Specifically, we address three key questions: 1) When are Transformers able to perform compositional reasoning by connecting fragmented knowledge in training data? 2) How do different training factors impact the emergence of this ability? 3) What internal mechanisms enable Transformers to develop this ability?

We first introduce the "FTCT" (Fragmented at Training, Chained at Testing) dataset, on which we investigate the performance of Transformers to address these questions. This dataset simulates knowledge relationships through graph-like causal structures, where vertices represent knowledge points and edges represent the relationships between their values. Multi-step reasoning paths are represented by chains consisting of connected vertices with values calculated by edges between them.

---

[*]Northwestern University; yutongyin2028@u.northwestern.edu, zhaoranwang@gmail.com.

[1]The code is in `https://github.com/roger-yt/Fragmented-at-Training-Chained-at-Testing`.

The training data comprises only fragmented knowledge segments (child chains of short lengths), while testing requires the model to connect these segments into long, complete causal chains. To examine the impact of prompting on compositional reasoning, we concatenate similar examples to enable reasoning with few-shot Chain-of-Thought (CoT) (Wei et al., 2022c) prompts. Detailed construction of the FTCT dataset is shown in Section 3. This approach evaluates the model's ability to generalize learned knowledge to novel combinations.

Regarding question 1), we find that few-shot CoT prompts enable Transformers to perform compositional reasoning. Testing Transformers trained on FTCT revealed poor compositional reasoning abilities in zero-shot scenarios, but significant improvement with few-shot CoT prompts. Few-shot CoT examples provide Transformers with correct vertices order to imitate, while Transformers iteratively deduce both vertices and their correct values based on preceding reasoning paths. Notably, the CoT prompts during testing time consist of complete reasoning paths—an out-of-distribution (OOD) scenario compared to the fragmented knowledge in training data. The capability to understand and utilize OOD CoT prompts indicates Transformers' compositional reasoning ability.

Regarding question 2), we examine the influence of data distribution and model complexity. Data-wise, compositional reasoning emerges as the similarity between training and testing data increases. In FTCT, this is measured by the relative knowledge ratio—the ratio of child chain length in training data to complete chain length in testing data. A phase transition occurs when the ratio reaches 0.3, marked by notable performance enhancement. Model-wise, multi-layer attention mechanisms prove essential, with the compositional reasoning emerging at a complexity of at least 2 layers and 2 heads. Single-layer Transformers fail to replicate vertices order from CoT prompts, and Multilayer Perceptrons (MLPs) struggle with sparse information of value relationships.

Regarding question 3), we find that Transformers develop compositional reasoning ability by learning an underlying program during training. This program, comprising in-context learning and parent retrieving (detailed in Section 5.1), minimizes both training and testing loss by capturing the common latent structure of fragmented knowledge and chained reasoning paths. Theoretically, we prove that Transformers have the expressivity to simulate such an underlying program. Empirically, through attention heatmap plotting and linear probing (Hewitt & Manning, 2019; Clark, 2019; Allen-Zhu & Li, 2023), we provide evidence that Transformers simulate the underlying program through two mechanisms—induction heads and attention assignment. These mechanisms respectively facilitate in-context learning and parent retrieving.

To summarize, within the context of the FTCT learning task, this article answers the above three questions as following:

1) Few-shot CoT prompting enables Transformers to perform compositional reasoning by providing the correct order of vertices to imitate, improving from poor zero-shot performance.

2) Compositional reasoning abilities emerge as the similarity between training and testing data increases (with a relative knowledge ratio $\geq 0.3$) and require multi-layer attention (with a minimum of 2 layers and 2 heads).

3) Transformers develop compositional reasoning by learning an underlying program during training, facilitated through induction heads and attention assignment, to integrate fragmented knowledge into coherent reasoning paths.

## 2    RELATED WORKS

**Step-by-step reasoning.** Chain-of-Thought (CoT) prompting (Nye et al., 2021; Wei et al., 2022c;b; Kojima et al., 2022) enables language models to conduct step-by-step reasoning, significantly boosting their performance in complex tasks like mathematical deduction and code generation (Cobbe et al., 2021; Suzgun et al., 2022; Zhou et al., 2022b; Lee et al., 2023; Achiam et al., 2023; Romera-Paredes et al., 2024). Our research emphasizes few-shot CoT prompting (Wei et al., 2022c), which initiates reasoning by integrating CoT examples into prompts. Interpretability studies suggest CoT's efficacy arises from models' enhanced expressivity via intermediate reasoning steps (Feng et al., 2024; Li et al., 2024b;a). Besides the expressivity perspective, we additionally examine CoT generalization in out-of-distribution (OOD) settings, showing few-shot CoT prompts can elicit correct reasoning even with previously unseen prompts. Another study (Prystawski et al., 2024) evaluates data's role in CoT's capacity for generalized reasoning. Our FTCT structure draws inspiration from

their Bayesian networks, additionally inserting contextual noise and complicating value relationships. While they focus on locality structure's impact on CoT efficacy, we investigate how various training factors influence compositional reasoning emergence and conduct an in-depth analysis of the mechanisms within Transformer structures that elicit such capability.

**In-context learning.** In-context learning (ICL) (Brown, 2020; Garg et al., 2022; Min et al., 2022; Wei et al., 2023) enables language models to perform various tasks by interpreting examples within the provided prompt, without needing explicit tuning. This capability allows Transformers trained on our FTCT to emulate the order of vertices in few-shot examples. Several theoretical studies (Xie et al., 2021; Li et al., 2023; Wang et al., 2024; Hahn & Goyal, 2023; Wies et al., 2024; Zhang et al., 2023) treat ICL as implicit Bayesian inference. Another set of works (Dai et al., 2022; Von Oswald et al., 2023; Akyürek et al., 2022; Ahn et al., 2024) argue that ICL functions similarly to gradient descent from a function approximation perspective. Notably, a mechanism within Transformers, known as "induction heads" (Elhage et al., 2021; Olsson et al., 2022; Bietti et al., 2024), is identified as the direct cause of ICL capabilities (detailed explanation is in Section 6.1). We demonstrate that induction heads exist in Transformers trained on FTCT, enabling the model to replicate the order of vertices from few-shot examples through ICL.

**Compositional generalization.** Compositional generalization refers to machine learning models' ability to solve new problems by integrating known components from training data (Lake & Baroni, 2018; Keysers et al., 2019; Kim & Linzen, 2020; Hupkes et al., 2020). Prior works have validated the potential of Transformers in compositional tasks where answers are directly output without intermediate reasoning steps (Hupkes et al., 2020; Arora & Goyal, 2023; Yu et al., 2023; Xu et al., 2024; Treutlein et al., 2024). In contrast, our FTCT dataset with deep causal structure allows exploration of explicit reasoning's impact on compositional generalization. While some empirical studies have shown that step-by-step reasoning enhances large language models' compositional abilities on real-world tasks (Press et al., 2022; Zhou et al., 2022a; Khot et al., 2022), the complexity of natural language corpora used by them complicates the scientific validation, which can be done credibly by our synthetic data. Recent studies have explored Transformers' generalized reasoning on various controllable synthetic tasks (Ramesh et al., 2023; Allen-Zhu & Li, 2023; Ye et al., 2024). In contrast, our FTCT task not only ensures controlled experimentation but also introduces measures of training-testing data similarity and establishes a distinct parent-child causal relationship, facilitating analysis of the underlying mechanisms in terms of data distribution and model structure. Although some studies have highlighted Transformers' shortcomings in achieving compositional generalization due to structural constraints (Dziri et al., 2024; Peng et al., 2024) or misleading statistical features (Zhang et al., 2022), our findings reveal that few-shot CoT prompting elicits compositional generalization in sequential reasoning tasks, indicating the importance of data structure and prompting skills.

**Transformer expressivity.** Transformers have been proven to exhibit a high degree of expressivity in approximating universal sequence-to-sequence functions (Yun et al., 2019), Turing machines (Pérez et al., 2019; Wei et al., 2022a), and programmable computers (Giannou et al., 2023). Prior research has demonstrated that Transformers can simulate various algorithms related to in-context learning (Dai et al., 2022; Von Oswald et al., 2023; Akyürek et al., 2022; Ahn et al., 2024), step-by-step reasoning (Li et al., 2024a; Feng et al., 2024), and causal graph inference (Nichani et al., 2024; Edelman et al., 2024; Makkuva et al., 2024). Building on this foundation, we explicitly construct a Transformer architecture that efficiently minimizes both the training and testing loss of FTCT. To streamline the proof, we adopt the associative memory perspective (Bietti et al., 2024), using this approach to construct the weight matrices as the sum of outer products of orthogonal embeddings.

## 3 FRAGMENTED AT TRAINING, CHAINED AT TESTING

We introduce the structure of FTCT dataset and corresponding training and testing loss, illustrating the reason why it measures the model's compositional reasoning ability. Figure 1 demonstrates the data generation procedure.

### 3.1 CAUSAL STRCTURE

We represent knowledge relationships with a directed graph $\mathcal{G} = (\mathcal{V}, \mathcal{E})$, where knowledge points are simulated by vertices $\mathcal{V}$, a subset of the alphabet $\mathcal{V}_{\text{all}} := \{A, B, \ldots, Z, a, b, \ldots, z\}$. Each vertex $v$ in

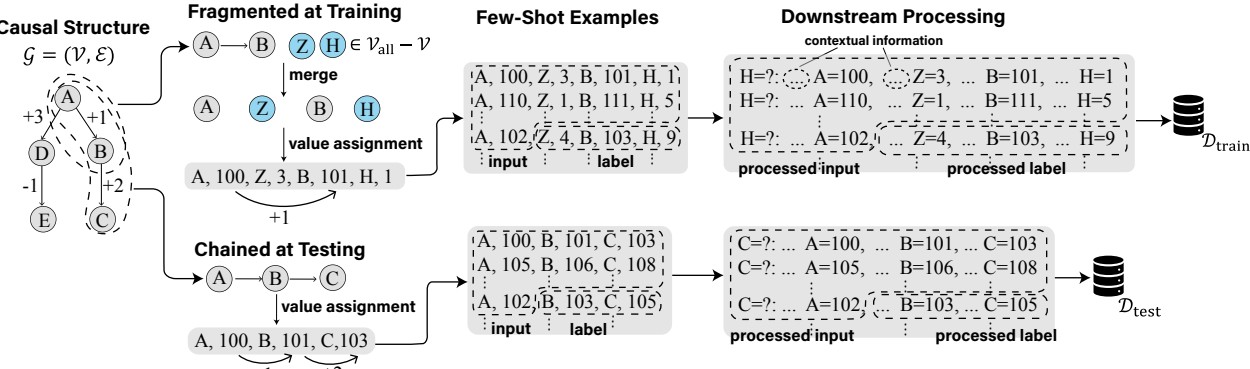

Figure 1: Overview of the FTCT Data Generation Process. The generation begins with the introduction of "Causal Structure", representing the relationships of knowledge points. The "Fragmented at Training" stage shows how shorter child chains with noise vertices are formed to simulate incomplete knowledge during training. The "Chained at Testing" stage presents the longest chains used in testing to assess compositional reasoning ability. The "Few-Shot Examples" part demonstrates the concatenation of multiple sequences for both training and testing to enable few-shot learning. In the end, the "Downstream Processing" adapts sequences into natural language-like sentences for intuitive reasoning format.

$\mathcal{V}$ has a value $q(v)$ from its associated set $\mathtt{VALS}(v) \subset \mathbb{Z}$. Relationships between knowledge points are represented by the edges set $\mathcal{E}$. Each edge $e := (v_1, v_2) \in \mathcal{E}$ defines the relationship between the parent vertex $v_1$ and the child vertex $v_2$ by the operation $\mathtt{op}(e)$, satisfying $q(v_2) = \mathtt{op}(e) \circ q(v_1)$. We assume that $\mathtt{op}(e)$ only represents addition or subtraction operation like $+a$ or $-b$. Multi-step reasoning paths are represented by the chains $\mathcal{T}(\mathcal{G}) := \{[v_1, v_2, \ldots, v_n] \mid n \in \mathbb{N}, (v_i, v_{i+1}) \in \mathcal{E}\}$. The depth of $\mathcal{G}$, denoted as $N$, is the length of the longest chain in $\mathcal{T}(\mathcal{G})$. The "Causal Structure" in Figure 1 illustrates a causal structure with depth $N = 3$.

## 3.2 DATA GENERATION

**Step 1.1. Fragmented at Training:** To simulate disconnected knowledge, training data excludes the longest chain in $\mathcal{T}(\mathcal{G})$ and instead includes shorter child chains with length $M < N$, interspersed with $M'$ noise vertices from $\mathcal{V}_{\text{all}} - \mathcal{V}$. Child chains vertices are merged with noise vertices, preserving their order. Child chain vertices receive values based on their edge operations, while noise vertices get random values. The final vertex-value sequence is formatted as $\mathtt{seq} := [v_1, q_1, \ldots, v_m, q_m]$, where $m = M + M'$. As shown in "Fragmented at Training" in Figure 1, the training data includes sequences like [A, 100, Z, 3, B, 101, H, 1], where [A, B] is a child chain from $\mathcal{T}(\mathcal{G})$ with values following the "+1" operation, and [Z, H] are sampled from $\mathcal{V}_{\text{all}} - \mathcal{V}$ with randomly assigned values.

**Step 1.2. Chained at testing:** To test models' compositional reasoning ability, the testing data consists of longest chains (length $N$) from $\mathcal{T}(\mathcal{G})$ without noise, formulating the sequence $\mathtt{seq} := [v_1, q_1, \ldots, v_N, q_N]$, where $(v_1, \ldots, v_N) \in \mathcal{T}(\mathcal{G})$. Refer to "Chained at Testing" in Figure 1.

**Step 2. Few-shot learning:** For both training and testing datasets, multiple sequences with the same vertices order are concatenated into few-shot document, which is formatted as

$$\mathtt{doc}^k := [\mathtt{seq}^{(1)}, \backslash \mathtt{n}, \ldots, \backslash \mathtt{n}, \mathtt{seq}^{(k)}], \quad \text{where } \mathtt{seq}^{(i)} := [v_1, q_1^{(i)}, \ldots, v_L, q_L^{(i)}]$$

where $L$ is the sequence length which can be either $m$ or $N$, and $k$ is the shots number ranging from 0 to $K$. The $k$-shot input and label are formatted as

$$\mathtt{inp}^k := \mathtt{doc}^k + [v_1, q_1^{(k+1)}], \quad \mathtt{lab}^k := [v_2, q_2^{(k+1)}, \ldots, v_L, q_L^{(k+1)}].$$

The model should generate $\mathtt{lab}^k$ autoregressively from $\mathtt{inp}^k$. Especially, the zero-shot input $\mathtt{inp}^0$ requires reasoning without any preceding examples. See "Few-Shot Examples" in Figure 1.

**Step 3. Downstream processing:** This process adapts sequences into natural language-like sentences by adding punctuation, contextual details, and stating the reasoning goal upfront. As illustrated in "Downstream Processing" in Figure 1, a few-shot document like [A, 110, Z, 1, B, 111, H, 5, \n, A, 102, Z, 4, B, 103, H, 9] transforms into the sentence

$$\underbrace{\text{"H=?: ... A=110, ... Z=1, ... B=111, ... H=5 \textbackslash n H=?: ... A=102,}}_{\text{processed input}} \underbrace{\text{... Z=4, ... B=103, ... H=9"}}_{\text{processed label}}$$

with "..." indicating tokens' context. The processed input and label are denoted as $\widetilde{\text{inp}}^k$ and $\widetilde{\text{lab}}^k$.

For brevity, we define $\mathcal{D}_{\text{train}}$ as the distribution of inputs and labels generated by **Step 1.1, 2, 3**, and $\mathcal{D}_{\text{test}}$ as the distribution of inputs and labels generated by **Step 1.2, 2, 3**. The detailed data generation with specific sampling methods is in Appendix B.

### 3.3 TRAINING AND TESTING LOSS

**Training loss:** For any input and label sampled from $\mathcal{D}_{\text{train}}$, we train the language model to autoregressively generate label given input. With the length of label $\widetilde{\text{lab}}^k$ defined as $d^k$, the training loss is formatted as

$$L_{\text{train}} := -\mathbb{E}_{\widetilde{\text{lab}}, \widetilde{\text{inp}} \sim \mathcal{D}_{\text{train}}} \sum_{k=0}^{K-1} \sum_{t=1}^{d^k-1} \log\left(P_{\text{model}}\left(\widetilde{\text{lab}}^k_{t+1} \mid \widetilde{\text{inp}}^k + \widetilde{\text{lab}}^k_{1:t}\right)\right).$$

**Testing loss:** For any input and label sampled from $\mathcal{D}_{\text{test}}$, given the input, we test how well a language model can generate sentence having the same vertex-value pairs as the label. Specifically, We define a decoding function $\text{dec}(\widetilde{\text{lab}}) := \text{lab} = [v_2, q_2, \ldots, v_N, q_N]$ to decode the vertex-value information from the processed label. Given the input, the sentence generated by the model is defined as $\text{model}(\widetilde{\text{inp}})$. We measure model's testing loss with $k$-shot prompt by

$$L^k_{\text{test}} := -\mathbb{E}_{\widetilde{\text{inp}}^k, \widetilde{\text{lab}}^k \sim \mathcal{D}_{\text{test}}} \mathbf{1}_{\{\text{dec}(\text{model}(\widetilde{\text{inp}}^k)) = \text{dec}(\widetilde{\text{lab}}^k)\}}. \tag{1}$$

Transformers trained on $\mathcal{D}_{\text{train}}$ have not seen the complete chains in $\mathcal{D}_{\text{test}}$ from either the reasoning paths or the few-shot examples. As a result, the fact that they still achieve low testing loss indicates the emergence of the compositional reasoning ability. The empirical version of training and testing loss are in Appendix A.

## 4 EMPIRICAL FINDINGS

### 4.1 FEW-SHOT CoT PROMPTING ENABLES COMPOSITIONAL REASONING

Our empirical findings highlight the essential role of few-shot CoT prompts in enabling compositional reasoning in Transformers during testing. We evaluate model's compositional reasoning ability using the following criterion:[2]

**Whole chain accuracy:** Measures if the model's generation contains all vertices and values along the reasoning chain in a correct order. For $(\widetilde{\text{inp}}^k, \widetilde{\text{lab}}^k)$ sampled from $\mathcal{D}_{\text{test}}$, it measures whether $\text{dec}(\text{model}(\widetilde{\text{inp}}^k))$ contains all elements from $\text{dec}(\widetilde{\text{lab}}^k)$ in a correct order.

Further, we decompose the compositional reasoning ability into two sublevel abilities—the ability of generating correct vertices order and the ability of deducing correct values given preceding paths, which are evaluated respectively by these criteria:

**Testing vertices accuracy:** Measures if the model correctly outputs all vertices in $\text{dec}(\widetilde{\text{lab}}^k)$.

**Testing values accuracy:** Measures if the model outputs correct values of intermediate vertices, given correct preceding reasoning paths sampled from $\mathcal{D}_{\text{test}}$. For the causal structure in Figure 1, this is tested by prompting models with sentences like "... A=100, ... B=". The model is considered to output accurate values if and only if it outputs "101" as the next token.

---

[2]We also assessed final value accuracy (Appendix H.1), which considers the model correct if it outputs the correct value of the last vertex. Models' performance under such metric mirrors that of whole chain accuracy, underscoring the necessity of proper reasoning paths for correct final answers.

We trained 3-layer 3-head GPT-2-like Transformers (details in Appendix G) on FTCT training set with varying graph depths and child chain lengths. Figure 2 (left) shows the testing performance for Transformers trained on graph depths $N = 5, 10, 15$, with $k$-shot CoT prompting ($k$ from 0 to 4). Different curve colors represent different child chain lengths $M = 2, 3, 4, 6$. Our conclusions are:

**Few-shot CoT enables compositional reasoning by revealing correct vertices order.** Whole chain accuracy is low with zero-shot prompts but increases sharply with more shots. At zero-shot, values accuracy is optimal while vertices accuracy is near zero. This indicates that few-shot CoT prompts enhance models' compositional reasoning by revealing the correct vertices order. Notably, such order has not appeared in the training data. The ability to understand and imitate the OOD vertices order stems from Transformers' in-context learning via induction heads (Section 5, 6).

**Transformer outputs correct values with OOD reasoning paths.** High testing values accuracy shows Transformers' robust performance in deducing correct values with the OOD compositional reasoning paths. Such an ability ensures models to iteratively output correct values of the vertices generated by few-shot CoT, leading to correct reasoning paths. This ability stems from Transformers' parent retrieving mechanism brought by proper attention assignment (Section 5, 6).

**An adequate number of shots is necessary.** Performance improvements plateau or decline after one-shot examples, possibly due to increased dissimilarity between training and testing data with more CoT examples. A detailed discussion is in Appendix K.

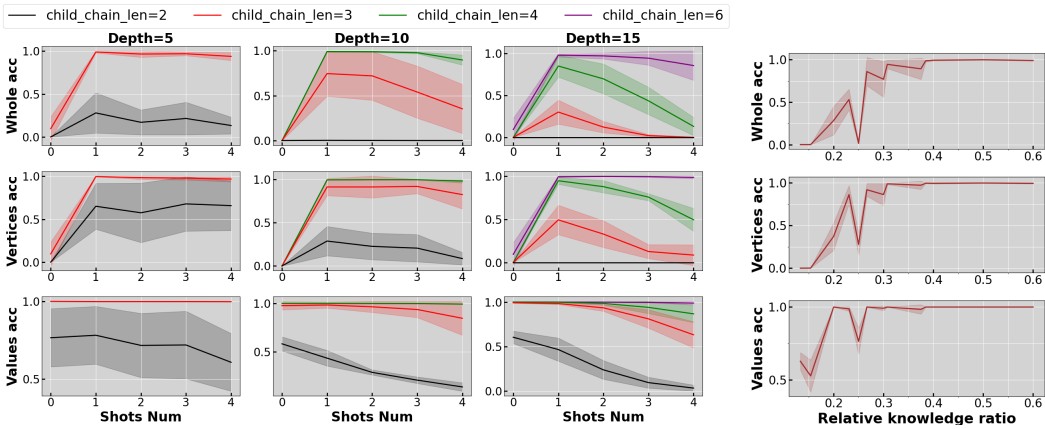

Figure 2: *Left:* Zero and few-shot testing performance of Transformers trained on FTCT with various causal depths and child chain lengths. *Right:* Relationship between the relative knowledge ratio and model's compositional reasoning ability. *Each row* shows testing performance under a specific criterion: The 1st row indicates whole chain accuracy (correct prediction of all vertices and values), the 2nd row shows testing vertices accuracy (correct order of vertices), and the 3rd row shows testing values accuracy (correct values of vertices with correct preceding reasoning paths).

## 4.2 THE SIMILARITY BETWEEN TRAINING AND TESTING DATA DETERMINES THE EMERGENCE OF COMPOSITIONAL REASONING

For each FTCT task, we measure the similarity between training and testing data by the relative knowledge ratio $\lambda := M/N$, where $M$ is the child chain lengths and $N$ is the causal graph depth. We find that compositional reasoning emerges as $\lambda$ increases. Figure 2 (right) illustrates the relationship between $\lambda$ and model's compositional reasoning ability. For each $\lambda$, the compositional reasoning ability is measured by the optimal few-shot testing performance of Transformers trained on tasks whose relative knowledge ratio is $\lambda$. A phase transition occurs: compositional reasoning remains weak when $\lambda < 0.3$ and distinctly emerges when $\lambda \geq 0.3$. In essence, a larger $\lambda$ makes few-shot CoT prompts more similar to training data, thereby enhancing testing performance. However, the fact that testing accuracy approaches one with $\lambda = 0.3$—a ratio significantly smaller than 1—underscores the non-triviality of our results.

Experiments on larger models like GPT-2-small (12 layers 12 heads) and GPT-2-large (36 layers 20 heads) show the same pattern (Appendix J), demonstrating the generalizability of our conclusions.

| Model | | Params | Training | | | Testing | | |
|---|---|---|---|---|---|---|---|---|
| | | | Valid loss | Vertices acc | Values acc | Vertices acc | Values acc | Whole acc |
| TF | 3L3H | 54M | **1.446** | **1.000** | **1.000** | **0.996** | **1.000** | **0.996** |
| | 2L2H | 48M | **1.433** | **1.000** | **1.000** | **1.000** | **1.000** | **1.000** |
| | 2L1H | 48M | **1.436** | 0.870 | **1.000** | 0.768 | **1.000** | 0.766 |
| | 1L2H | 42M | 1.559 | 0.138 | **0.994** | 0.776 | **1.000** | 0.776 |
| | 1L1H | 42M | 1.624 | 0.025 | **0.974** | 0.026 | **1.000** | 0.022 |
| MLP | 4L | 145M | 3.464 | 0.063 | 0.104 | 0.484 | 0.290 | 0.002 |
| | 2L | 143M | 2.147 | 0.065 | 0.146 | **0.850** | 0.346 | 0.022 |

Table 1: Performance of different models trained on FTCT task with causal structure depth $N = 5$ and child chain length upper bound $M = 3$. "Params" stands for numbers of parameters of different models; "Training" column includes three criteria described in Section 4.3 measuring the in-distribution performance;"Testing" column includes three criteria described in Section 4.1 measuring the compositional reasoning performance. We only show the performance of models prompted by CoT with the optimal shots number.

### 4.3 MULTI-LAYER ATTENTION MECHANISM ENABLES COMPOSITIONAL REASONING

We show that compared with other simpler structures, multi-layer Transformers excel at imitating vertices order from few-shot examples and deducing correct values with preceding reasoning paths, leading to outstanding compositional reasoning ability. In addition to compositional reasoning metrics (Section 4.1), we introduce the following criteria to evaluate in-distribution performance:

**Validation loss:** Tracks the validation loss during training.

**Training vertices accuracy:** Assesses how well the model imitates the vertices order from the few-shot examples sampled from $\mathcal{D}_{\text{train}}$.

**Training values accuracy:** Measures if the model outputs correct values of vertices from child chains, given preceding reasoning paths sampled from $\mathcal{D}_{\text{train}}$.

We assess the performance of various models on the FTCT task with a causal structure depth of 5 and a child chain length of 3. Models include Transformers (TF) of various layers and heads, and multi-layer perceptrons (MLPs) of different depths (details in Appendix G). Table 1 summarizes the results. For brevity, we only show the performance of models prompted by CoT with the optimal shots number.

**Depth of Transformer enables the imitation of vertices.** Table 1 indicates that Transformers with at least 2 layers and 2 heads achieve optimal in-distribution and compositional reasoning performance. As complexity decreases, performance deteriorates, notably with a significant drop in the vertices accuracy, both training and testing, while values accuracy remains optimal. Such phenomenon indicates that depth in Transformers is crucial for imitating vertices order in few-shot examples, hence enhancing compositional reasoning. This is because induction heads for in-context learning are less likely in single-layer Transformers (Section 6).

**Attention mechanism enables the deduction of sparse values information.** For MLPs with appropriate window sizes (details in Appendix G.2), both the training and testing values accuracy remain low. Conversely, even the simplest Transformer (1 layer, 1 head) achieves nearly optimal values accuracy, suggesting that MLPs struggle to capture sparse value information in noisy contexts as effectively as Transformers. Interestingly, MLPs perform well in generating vertices order during testing but not during training, possibly due to the extra noise vertices in the training data, suggesting a different knowledge memorization approach that warrants further study.

## 5 TRANSFORMER DOES COMPOSITIONAL REASONING VIA THE UNDERLYING PROGRAM

As discussed in Section 4.3, the multi-layer attention mechanism of Transformers is crucial for compositional reasoning. However, how Transformers achieve this ability through training remains

unclear. In this section, we explain this mystery by showing the capability of Transformer in learning an underlying program that minimizes both training and testing loss. In Section 6 we provide empirical evidence of this underlying program in the Transformer's hidden states. We focus on the few-shot testing loss $\{L_{\text{test}}^k\}_{k=1}^{K-1}$ and a modified training loss accounting only for shots number $k \geq 1$, which is defined as

$$\widehat{L}_{\text{train}} := -\mathbb{E}_{\widetilde{\text{lab}}, \widetilde{\text{inp}} \sim \mathcal{D}_{\text{train}}} \sum_{k=1}^{K-1} \sum_{t=1}^{d^k-1} \log \left( P_{\text{model}}\left( \widetilde{\text{lab}}_{t+1}^k \mid \widetilde{\text{inp}}^k + \widetilde{\text{lab}}_{1:t}^k \right) \right). \tag{2}$$

## 5.1 Underlying Program

We construct a text-generating program which provably achieves optimal performance on both training and testing data. Key components are summarized here, with an algorithm (Algorithm 1) in the Appendix. Given any input sentence $z_{1:T}$ that contains at least 1-shot example, the program executes the following two parts iteratively:

**In-context learning:** If the last token $z_T$ is a comma ",", $z_{T-3}$ must be a vertex $v_i$. The program identifies all $v_i$ in the previous few-shot examples and attends to their next vertex $v_{i+1}$, returning $v_{i+1}$'s contextual tokens.

**Parent retrieving:** If the last token $z_T$ is an equation token "=", $z_{T-1}$ must be a vertex $v_j$. The program retrieves the parent of $v_j$ from the preceding context. If $v_j$ belongs to the child chain from $\mathcal{T}(\mathcal{G})$ and has a parent $v_{j_1}$ with value $q_{j_1}$ in the preceding context, the program returns $q_j = \text{op}(v_{j_1}, v_j) \circ q_{j_1}$ with probability one. Otherwise, it returns value $q_j$ randomly sampled from $v_i$'s value set $\text{VALS}(v_i)$.

For example, in the testing sentence shown in Figure 1, given the input

$$\text{"C=?: } \dots \text{A=100, } \dots \text{B=101, } \dots \text{C=103 } \backslash \text{n C=?: } \dots \text{A=105,"} ,$$

the program, through in-context learning, attends to "B" in the preceding example and outputs its contextual information as the next token. The program continues generating tokens using other minor parts from Algorithm 1 until the output

$$\text{"C=?: } \dots \text{A=100, } \dots \text{B=101, } \dots \text{C=103 } \backslash \text{n C=?: } \dots \text{A=105, } \dots \text{, B="} .$$

The program then retrieves "A" as the parent of "B" by parent retrieving part, returning the value $106 = 105 + 1$. The Following lemmas show that the underlying program minimizes both the few-shot training loss and few-shot testing loss.

**Lemma 5.1.** *For any sentence $z_{1:T} := \widetilde{\text{inp}}^k + \widetilde{\text{lab}}_{1:t}^k$, where input $\widetilde{\text{inp}}^k$ and label $\widetilde{\text{lab}}^k$ are sampled from $\mathcal{D}_{train}$ with $k \geq 1$, and $t$ is an arbitrary position within the label, denote the distribution output by the program as $P_{prog}(\cdot \mid z_{1:T})$. We have that $P_{prog}(\cdot \mid z_{1:T}) = P_{train}(\cdot \mid z_{1:T})$. Hence, $P_{prog}$ minimizes the few-shot training loss $\widehat{L}_{train}$ (eq. (2)).*

**Lemma 5.2.** *For any input $\widetilde{\text{inp}}^k$ and label $\widetilde{\text{lab}}^k$ sampled from $\mathcal{D}_{test}$ with $k \geq 1$, denoting the sentence generated by the program as $prog(\widetilde{\text{inp}}^k)$, we have $dec(prog(\widetilde{\text{inp}}^k)) = dec(\widetilde{\text{lab}}^k)$. Hence, $prog(\cdot)$ minimizes the few-shot testing loss $\{L_{test}^k\}_{k=1}^{K-1}$ (eq. (1)).*

Lemma 5.1 and 5.2 show that, despite the different distributions of fragmented training data and chained testing data, their next-token distributions given few-shot examples can be represented by a common program. As demonstrated in Sections 5.2 and 6, the Transformer learns this common latent structure through training, achieving the compositional reasoniong ability.

## 5.2 Transformer is Expressive Enough to Simulate the Underlying Program

We prove that Transformer is expressive enough to simulate the underlying program by explicitly constructing a 2 layers Transformer. Representing the model parameters by $\theta$, we state

**Lemma 5.3.** *There exists a 2 layer Transformer with parameters $\theta^*$ that approximates Algorithm 1 with arbitrarily small error.*

| | | |
|---|---|---|
| Layer 2 | ... ... E = 138 , ... A = 141 **,** ... o = 135 , ... i = 130 , ... D = 138 , ... \n ... E = 125 **,** | |
| Layer 1 | ... ... E = 138 , ... A = 141 **,** | ... ... E = 125 **,** |
| Layer 2 | ... ... A = 141 , ... o = 135 **,** ... i = 130 , ... D = 138 , ... \n ... E = 125 , ... A = 128 **,** | |
| Layer 1 | ... ... E = 138 , ... A = 141 , ... o = 135 **,** | ... ... E = 125 , ... A = 128 **,** |
| Layer 2 | ... ... o = 135 **,** ... i = 130 **,** ... D = 138 , ... \n ... E = 125 , ... A = 128 , ... o = 122 **,** | |
| Layer 1 | ... ... E = 138 , ... A = 141 , ... o = 135 , ... i = 130 **,** | ... ... E = 125 , ... A = 128 , ... o = 122 **,** |
| Layer 2 | ... ... i = 130 , ... D = 138 **,** ... \n ... E = 125 , ... A = 128 , ... o = 122 , ... i = 117 **,** | |
| Layer 1 | ... ... A = 141 , ... o = 135 , ... i = 130 , ... D = 138 **,** | ... E = 125 , ... A = 128 , ... o = 122 , ... i = 117 **,** |

Table 2: Attention weights of induction heads on the sentence sampled from test dataset.

| Data | | 0-shot prompt | | 1-shot prompt | | 2-shot prompt | | 3-shot prompt | | 4-shot prompt | |
|---|---|---|---|---|---|---|---|---|---|---|---|
| | | Parent | Others | Parent | Others | Parent | Others | Parent | Others | Parent | Others |
| Depth 5 | train | **78.6** | 11.6 | **79.3** | 10.3 | **81.9** | 9.3 | **80.8** | 9.5 | **77.9** | 10.2 |
| Child 3 | test | **78.3** | 4.3 | **77.1** | 4.5 | **78.8** | 4.5 | **79.8** | 4.1 | **77.8** | 4.5 |
| Depth 10 | train | **77.6** | 10.3 | **80.8** | 10.0 | **79.3** | 9.7 | **78.3** | 9.5 | **80.5** | 9.4 |
| Child 4 | test | **81.5** | 3.1 | **79.3** | 2.6 | **78.4** | 2.7 | **78.8** | 2.5 | **74.6** | 2.5 |
| Depth 15 | train | **80.6** | 8.8 | **83.8** | 9.4 | **80.6** | 10.0 | **81.8** | 10.9 | **79.7** | 9.6 |
| Child 5 | test | **78.2** | 2.4 | **80.3** | 2.4 | **78.8** | 2.1 | **76.9** | 2.9 | **73.9** | 2.5 |

Table 3: Probing accuracy in predicting the replaced values of different vertices. "Parent" column records the accuracy in predicting values of parent vertices while "Others" column records the accuracy in predicting values of non-parent vertices.

The ability of a 2-layer Transformer to simulate the underlying program aligns with the empirically observed performance in Table 1. By expressing the training and testing loss as functions of the model parameters $\theta$, we summarize Lemmas 5.1, 5.2, and 5.3 into the following theorem.

**Theorem 5.4.** *There exists a Transformer model parameterized by $\theta^*$ that satisfies*

$$\begin{cases} \left| \widehat{L}_{train}(\theta^*) - \min_\theta \widehat{L}_{train}(\theta) \right| < \epsilon, & \text{where } \epsilon \text{ is an arbitrarily small value,} \\ L_{test}^k(\theta^*) = 0, & \text{where } k = 2, \dots, K. \end{cases}$$

Theorem 5.4 shows that the parameters $\theta^*$ approximate the minimizers of the few-shot training loss, indicating they can be learned from fragmented knowledge. Additionally, $\theta^*$ minimizes the few-shot testing loss, reflecting strong compositional reasoning ability. The reason why $\theta^*$ (approximately) minimizes both the training and testing loss is that it simulates the underlying program common to both training and testing data. It is noteworthy that the theorem only establishes the existence of such program-simulating $\theta^*$. However, empirical evidence in Section 6 demonstrates that the hidden states of Transformers trained in practice exhibit patterns corresponding to the underlying program.

## 6 EMPIRICAL EVIDENCE OF THE UNDERLYING PROGRAM

We present empirical evidence showing that Transformers are simulating the underlying program through two mechanisms—induction head and attention assignment, which respectively facilitate the in-context learning and parent retrieving.

### 6.1 INDUCTION HEADS

By plotting Transformer's attention heat map, we provide empirical evidence showing the existence of induction heads that enables in-context learning. As described in previous works (Elhage et al., 2021; Olsson et al., 2022), induction heads are two heads of the Transformer in different layers that collaborate to copy patterns. For example, with input sentences like "... [A][B]... [A]", the first head in a shallow layer copies information from the first [A] to [B], while the second head in a

deeper layer recognizes [B] and retrieves its context from [A], guiding the model to output [B] as the next token. In our task, we discovered similar induction heads operating in a slightly different manner. Given an input sentence formatted as (for clarity, we highlight comma tokens at different positions with boxes and different colors):

$$\text{``}\ldots v_i = q_i, \ldots v_{i+1} = q_{i+1}\boxed{,} \ldots \backslash \texttt{n} \ldots v_i = q_i' \boxed{,}\text{''}$$

The head in the shallower layer copies the information of $v_i$ and $v_{i+1}$ to ",". The head in the deeper layer attends "," along with the information of $v_{i+1}$ to ",", making the model to output the contextual information of $v_{i+1}$.

To empirically demonstrate this pattern, we trained a 3-layer, 3-head Transformer on the FTCT task with a causal structure depth of 13 and a child chain length of 6, generating attention heatmaps for each layer. Complete heatmap plots are available in Appendix L, and an abbreviated version is shown in Table 2 which displays the average attention weights of heads in different layers for their respective tokens. For each comma in the black frame, the distribution of its attention weights to preceding tokens is shown using colored boxes—the brighter the color, the more attention paid. In Layer 1, each comma attends to its previous two vertices, recording their information in its hidden state. In Layer 2, each comma uses this information to identify the preceding comma whose vertex is next to be output.

## 6.2 ATTENTION ASSIGNMENT

By linear probing (Hewitt & Manning, 2019; Clark, 2019; Allen-Zhu & Li, 2023), we empirically show that the parent retrieving is facilitated by proper attention assignment—focusing on the value of parent vertex while ignoring others.

For each sentence sampled from either the training or testing data, we identify the equation token "=", where its corresponding vertex has a parent in the preceding context. Specifically, we examine input formatted as:

$$\text{``}\ldots v_{j_1} = q_{j_1}, \ldots v_j = \text{''}$$

where $v_{j_1}$ is the parent of $v_j$. We construct the probing dataset by each time picking a position $i < j$ (including $j_1$), replacing $q_i$ with randomly sampled $q_i'$, and recording the Transformer's hidden state for this modified sentence. We train a linear function (details in Appendix G.3) to predict $q_i'$ from the hidden states. If the Transformer attends to $q_i$, the linear function should predict $q_i'$ with high accuracy. If not, the accuracy should be low.

Table 3 shows the results for 3-layer, 3-head Transformers trained on multiple FTCT tasks. For sentences sampled from training or testing data, linked to prompts with shot numbers 0-4, the probing accuracy of predicting replaced values is tested. The results demonstrate that for both training and testing data, the probing function achieves high accuracy in predicting parents' replaced values, while showing low accuracy for other positions. This indicates that Transformers successfully retrieve parent vertices within fragmented knowledge, ignoring irrelevant information.

## 7 CONCLUSION

Our research validates the potential of Transformers in doing compositional reasoning on synthetic data and investigates the inner mechanism eliciting such ability. We demonstrate that few-shot CoT prompting enables Transformers to perform compositional reasoning by providing the information of correct order of knowledge points. We also find that compositional reasoning ability emerges when the training-testing data similarity and the model complexity are above certain thresholds. We further show that Transformers develop compositional reasoning by learning an underlying program during training, which minimizes both training and testing loss. This program leverages in-context learning and parent retrieving mechanisms, facilitated by induction heads and attention assignment.

Through experiments on synthetic data, we demonstrate the potential of Transformers to develop generalized reasoning skills, indicating that the impressive performance of contemporary large language models extends beyond mere memorization of vast data. While our conclusions may not directly apply to real-world models trained on extensive natural language datasets, we believe that our analysis offers valuable insights into the training processes and understanding of today's large language models.

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

# A EMPIRICAL LOSS

## A.1 EMPIRICAL TRAINING LOSS

With $B$ samples in total, at each sample step $b$ we sample $\{\widetilde{\text{inp}}^{k,b}, \widetilde{\text{lab}}^{k,b}\}_{k=1,b=1}^{K,B}$ from $\mathcal{D}_{\text{train}}$. With the length of label $\widetilde{\text{lab}}^{k,b}$ defined as $d^{k,b}$, the empirical training loss is formatted as

$$L_{\text{train}} := -\frac{1}{B} \sum_{b=1}^{B} \sum_{k=0}^{K-1} \sum_{t=1}^{d^{k,b}-1} \log\left(P_{\text{model}}\left(\widetilde{\text{lab}}_{t+1}^{k,b} \mid \widetilde{\text{inp}}^{k,b} + \widetilde{\text{lab}}_{1:t}^{k,b}\right)\right).$$

## A.2 EMPIRICAL TESTING LOSS

With $B$ samples in total, at each sample step $b$ we sample $\{\widetilde{\text{inp}}^{k,b}, \widetilde{\text{lab}}^{k,b}\}_{k=1,b=1}^{K,B}$ from $\mathcal{D}_{\text{test}}$. The empirical testing loss with $k$-shot prompt is

$$L_{\text{test}}^{k} := -\frac{1}{B} \sum_{b=1}^{B} \mathbf{1}_{\{\text{dec}(\text{model}(\widetilde{\text{inp}}^{k,b}))=\text{dec}(\widetilde{\text{lab}}^{k,b})\}}.$$

# B DETAILED DATA GENERATION

**Step 0. Causal structure:** The dataset is based on directed graph structures $\mathcal{G} = (\mathcal{V}, \mathcal{E})$, where $\mathcal{V}$ is a subset of the alphabet vocabulary $\mathcal{V}_{\text{all}} := \{A, B, \ldots, Z, a, b, \ldots, z\}$. Each vertex $v \in \mathcal{V}_{\text{all}}$ has a value $q(v)$ from its set $\text{VALS}(v) \subset \mathbb{Z}$. For any edge $e := (v_1, v_2) \in \mathcal{E}$, $v_1$ is the parent and $v_2$ is the child. Each edge $e$ maps $\text{VALS}(v_1)$ to $\text{VALS}(v_2)$ via $\text{op}(e) : \text{VALS}(v_1) \to \text{VALS}(v_2)$, satisfying $q(v_2) = \text{op}(e) \circ q(v_1)$. We assume $\text{op}(e)$ represents operations like $(+a)$ or $(-b)$. The set of chains $\mathcal{T}(\mathcal{G}) := \{[v_1, v_2, \ldots, v_n] \mid n \in \mathbb{N}, (v_i, v_{i+1}) \in \mathcal{E}\}$ represents sequences of connected vertices. The depth of $\mathcal{G}$, denoted as $N$, is the length of the longest chain in $\mathcal{T}(\mathcal{G})$.

**Step 1.1. Fragmented at Training:** The training data consists of child chains sampled from $\mathcal{T}(\mathcal{G})$ with length $M < N$ as well as $M'$ noise vertices sampled from $\mathcal{V}_{\text{all}} - \mathcal{V}$. Child chains are merged with noise vertices while maintaining their relative order. Within the merged sequence, vertices belonging to the child chains are assigned values following their defined operations, while noise vertices are given random values. Suppose the child chain is $s_1 := [u_1, \cdots, u_M] \in \mathcal{T}(\mathcal{G})$ and the sequence of noise vertices is $s_2 := [u'_1, \cdots, u'_{M'}]$. The merged sequence is $[v_1, \cdots, v_m]$, where $m = M + M'$, $[v_{i_1}, \cdots, v_{i_M}] = s_1$ and $[v_{j_1}, \cdots, v_{j_{M'}}] = s_2$. We then sample $[q_1, \cdots, q_m]$ as the corresponding values of vertices. For $[q_{i_1}, \cdots, q_{i_M}]$ that corresponds to $s_1$, it follows the operators over the chain, where

$$q_{i_1} \sim \text{Uniform}(\text{VALS}(u_1)) \text{ and } q_{i_h} = \text{op}((u_{h-1}, u_h)) \circ q_{i_{h-1}} \text{ for } h = 2, \cdots, M.$$

For $[q_{j_1}, \cdots, q_{j_{M'}}]$ that corresponds to $s_2$, they are just random noise so we have

$$q_{j_h} \sim \text{Uniform}(\text{VALS}(u'_h)) \text{ for } h = 1, \cdots, M'.$$

The final sequence of vertex-value pairs is formatted as $\text{seq} := [v_1, q_1, \ldots, v_m, q_m]$, where $m = M + M'$.

**Step 1.2. Chained at testing:** In contrast to the training dataset, the testing data consists of complete chains of length $N$ from $\mathcal{T}(\mathcal{G})$ without noise vertices, formulating the sequence $\text{seq} := [v_1, q_1, \ldots, v_N, q_N]$, where $(v_1, \ldots, v_N) \in \mathcal{T}(\mathcal{G})$, $q_1 \sim \text{Uniform}(\text{VALS}(v_1))$ and $q_i = \text{op}((v_{i-1}, v_i)) \circ q_{i-1}$ for $i = 2, \cdots, N$

**Step 2. Few-shot learning:** For both training and testing datasets, multiple sequences with the same vertices order are concatenated into few-shot document, which is formatted as

$$\text{doc}^k := [\text{seq}^{(1)}, \backslash n, \ldots, \backslash n, \text{seq}^{(k)}], \quad \text{where } \text{seq}^{(i)} := [v_1, q_1^{(i)}, \ldots, v_L, q_L^{(i)}].$$

Here $L$ is the number of vertex-value pairs that can be either $m$ or $N$, and $k$ is the shot numbers, ranging from 0 to $K$. The $k$-shot input and label are formatted as

$$\text{inp}^k := \text{doc}^k + [v_1, q_1^{(k+1)}], \quad \text{lab}^k := [v_2, q_2^{(k+1)}, \ldots, v_L, q_L^{(k+1)}].$$

Given the input $\text{inp}^k$, the model is expected to autoregressively generate the label $\text{lab}^k$ by referring to the $k$-shot examples in $\text{doc}^k$. Especially, the zero-shot input $\text{inp}^0$ requires reasoning without any preceding examples.

**Step 3. Downside processing:** In order to make sentences resemble natural language, we apply the following modifications: First, insert equation token $o^{\text{eq}}$ between each vertex $v_i$ and value $q_i$. Second, insert comma token $o^{\text{cm}}$ between each value $q_i$ and next vertex $v_{i+1}$. Third, insert contextual tokens $[c(v_i)_1, \ldots, c(v_i)_{l_i}]$ before each vertex $v_i$. Lastly, define the question token $o^{\text{qu}}$ and insert $[v_L, o^{\text{eq}}, o^{\text{qu}}]$ to the beginning of the sequence, indicating the final goal to be reasoned. For each sequence $\text{seq} = [v_1, q_1, \ldots, v_L, q_L]$, the processed form of it is like

$$\widetilde{\text{seq}} := [v_L, o^{\text{eq}}, o^{\text{qu}}, c(v_1)_1, \ldots, c(v_1)_{l_1}, v_1, o^{\text{eq}}, q_1, o^{\text{cm}}, \ldots, o^{\text{cm}}, c(v_L)_1, \ldots, c(v_L)_{l_L}, v_L, o^{\text{eq}}, q_L].$$

Similarly, with the processed document defined as $\widetilde{\text{doc}}^k := \{\widetilde{\text{seq}}^{(i)}\}_{i=1}^k$, the processed input and label are formatted as

$$\widetilde{\text{inp}}^k := \widetilde{\text{doc}}^k + [v_L, o^{\text{eq}}, o^{\text{qu}}, c(v_1)_1^{(k+1)}, \ldots, c(v_1)_{l_1}^{(k+1)}, v_1, o^{\text{eq}}, q_1^{(k+1)}, o^{\text{cm}}]$$

$$\widetilde{\text{lab}}^k := [c(v_2)_1^{(k+1)}, \ldots, c(v_2)_{l_2}^{(k+1)}, v_2, o^{\text{eq}}, q_2^{(k+1)}, o^{\text{cm}}, \ldots, o^{\text{cm}}, c(v_L)_1^{(k+1)}, \ldots, c(v_L)_{l_L}^{(k+1)}, v_L, o^{\text{eq}}, q_L^{(k+1)}]$$

The contextual tokens are sampled in the following way: For each vertex $v \in \mathcal{V}_{\text{all}}$, we maintain its context token set $\text{CONT}(v)$. For any $v_1 \neq v_2$, we regulate $\text{CONT}(v_1) \cap \text{CONT}(v_2) = \varnothing$. For each vertex $v_i$ in the sequence, its contextual tokens $c(v_i)_1, \cdots c(v_i)_{l_i}$ are sampled from $\text{CONT}(v_i)$ without replacement, with $l_i$ following the uniform distribution $\text{Uniform}([|\text{CONT}(v_i)|])$.

## C    CONCRETE EXAMPLES OF SENTENCES IN FTCT

We provide concrete examples of sentences in the FTCT, using the causal structure illustrated in Figure 1. Each vertex in $\mathcal{V}_{\text{all}}$ is associated with its own set of contextual tokens, represented as follows:

> A: {"intern", "accomplishment", "whistle", "subsystem", "Rewards", "patents", "ARCH" }
>
> B: {"correl", "register", "044", "ask", "latex", "Coins", "google" }
>
> C: {"ü", "exc", "increasing", "REAM", "she", "-.", "operated"}
>
> Z: {"Influ", "interf", "Ideally", "Pad", "adders", "confusion", "XP"}
>
> H: {"Higher", "fren", "romptu", "smoke", "shake", "Frank", "treasure"}
>
> v: {"ACH", "Catalyst", "pens", "emer", "4000", " Cars", "easiest"}

In this representation, [A, B, C] are knowledge points from $\mathcal{V}$, and [Z, H, v] are random noise from $\mathcal{V}_{\text{all}} - \mathcal{V}$. The contextual tokens carry no semantic meaning, and the contextual token sets for different vertices do not overlap. The 3-shot training input containing [A, B] is like

> "B=?: confusion Pad interf Ideallyadders XPZ=6, accomplishmentintern subsystemARCH
>
>   Rewards patentsA=120,romptuH=8,044 Coins google correl latexB=123
>
> B=?: confusion Pad Ideally XPZ=7, accomplishment patents subsystem RewardsARCHA=122,
>
>   treasureFrankH=3, googleregister044 CoinsaskB=125
>
> B=?: Pad Influ interfZ=5,".

Its corresponding label is

"accomplishment subsystem whistle RewardsARCHinternA=147,smokeH=6, latexregister googleB=150".

The 3-shot training input containing [B, C] is like

> "C=?: correl044 latexregisterB=126, 4000v=6,increasingoperatedREAMC=135
>
> C=?: Coins correlaskB=108, pens Catalystv=6, exc-.increasingC=117
>
> C=?: correl Coins googleregisterB=144,".

Its corresponding label is

"Carsemerv=5,-. excoperatedC=153".

---

**Algorithm 1** Generalized Reasoning

1: **Input:** Causal graph $\mathcal{G} = (\mathcal{V}, \mathcal{E})$, all vertices $\mathcal{V}_{\text{all}}$, input sentence $z_{1:T}$, shot number $f \geq 1$.
2: **if** $z_T = o^{\text{cm}}$ **then**
3:     That means $z_{T-3}$ must be a vertex $v_i$. There are $f$ vertices $v_i$ in the previous sentence. Find all $v_i$ in the previous sentence and pay attention to there next vertex $v_{i+1}$, deduce the contextual information $\text{CONT}(v_{i+1})$, return $\text{Uniform}(\text{CONT}(v_{i+1}))$.
4: **else if** $z_T = c(v_j)_k$ **then**
5:     That means $z_{T-k+1:T} = [c(v_j)_1, \cdots, c(v_j)_k]$. Pay attention to them and return
    $\text{Uniform}(\text{CONT}(v_j) + \{v_j\} - \{c(v_j)_{k'}\}_{k'=1}^{k})$
6: **else if** $z_T = v_j$ **then**
7:     return distribution $P(\cdot)$ where $P(o^{\text{eq}}) = 1$.
8: **else if** $z_T = o^{\text{eq}}$ **then**
9:     Pay attention to its corresponding vertex $v_j = z_{T-1}$.
10:     **if** $v_j \in \mathcal{V}$ **then**
11:       If $v_j$ is the first vertex from $\mathcal{V}$ that appears in the sentence, then return $\text{Uniform}(\text{VALS}(v_j))$. Otherwise, search in previous tokens and pay attention to the nearest $v_{j_1}, q_{j_1}^f$ that $v_{j_1} \in \mathcal{V}$. Such a $v_{j_1}$ must be the parent of $v_j$. Then return distribution $P(\cdot)$ where $P(\text{op}(v_{j_1}, v_j) \circ q_{j_1}) = 1$.
12:     **else**
13:       Return $\text{Uniform}(\text{VALS}(v_j))$.
14:     **end if**
15: **else if** $z_T = q_j$ **then**
16:     Return distribution $P(\cdot)$ where $P(o^{\text{cm}}) = 1$
17: **end if**

---

The 3-shot testing input is like

"C=?: accomplishment patents Rewards subsystemA=134, Coins correlB=137,-.REAM exc sheC=146

C=?: subsystem patents RewardsA=103, latex google correlB=106,REAM exc sheC=115

C=?: patents accomplishmentA=124,"

Its corresponding label is

"ask google044 correl CoinsB=127, sheincreasingREAM-.üoperatedC=136".

## D GENERALIZED PROGRAM

### D.1 DETAILED ALGORITHM

The detailed algorithmic description of the generalized text generating program is shown in Algorithm 1

### D.2 PROOF OF LEMMA 5.1

For any sentence $z_{1:T}$, its last token $z_T$ must be one of the tokens $\{o^{\text{cm}}, c(v_j)_k, v_j, o^{\text{eq}}, q_j\}$. Then we discuss the distribution of the next token in $\mathcal{D}_{\text{train}}$ when $z_T$ is any one of them.

- When $z_T = o^{\text{cm}}$, that means $z_{T-3}$ must be a vertex $v_i$ and the next token will be one of the contextual tokens of $v_{i+1}$. To determine which specific vertex $v_{i+1}$ is, we can refer to previous examples to find what is the next vertex after every $v_i$. After determining $v_{i+1}$, in $\mathcal{D}_{\text{train}}$, the next token follows the distribution of $c(v_{i+1})_1$. Because $c(v_{i+1})_1, \cdots, c(v_{i+1})_{l_{i+1}}$ are sampled from $\text{CONT}(v_{i+1})$ without replacement, so $c(v_{i+1})_1 \sim \text{Uniform}(\text{CONT}(v_{i+1}))$, deducing $P_{\text{train}}(\cdot \mid z_{1:T}) = \text{Uniform}(\text{CONT}(v_{i+1}))$.

- When $z_T = c(v_j)_h$, the previous $h - 1$ tokens must be $c(v_j)_1, \cdots c(v_j)_{h-1}$. Because $l_j$ is sampled from $\text{Uniform}(1, |\text{CONT}(v_j)|)$, then we have $P(l_j = \cdot \mid l_j \geq h) \sim \text{Uniform}(h, |\text{CONT}(v_j)|)$. So with probability $1/(|\text{CONT}(v_j)| - h + 1)$ the next token is $v_j$, with probability $(|\text{CONT}(v_j)| - h)/(|\text{CONT}(v_j)| - h + 1)$ the next token is sampled

from Uniform($\text{CONT}(v_j) - \{c(v_j)_1, \cdots, c(v_j)_h\}$). Combining them together, we have that $P_{\text{train}}(\cdot \mid z_{1:T}) = \text{Uniform}(\text{CONT}(v_j) \bigcup \{v_j\} - \{c(v_j)_1, \cdots, c(v_j)_h\})$.

- When $z_T = v_j$, the next token is $o^{\text{eq}}$, deducing $P_{\text{train}}(o^{\text{eq}} \mid z_{1:T}) = 1$.

- When $z_T = o^{\text{eq}}$, $z_{T-1}$ must be a vertex $v_j$ and the next token must be a value $q_j$. If $v_j \in \mathcal{V}_{\text{all}} - \mathcal{V}$, then $q_j$ is sampled from Uniform($\text{VALS}(v_j)$), deducing $P_{\text{train}}(\cdot \mid z_{1:T}) = \text{Uniform}(\text{VALS}(v_j))$. If $v_j \in \mathcal{V}$ and $v_j$ is not the first vertex from $\mathcal{V}$ in this sentence, there must exist $v_{j_1} \in \mathcal{V}$ which is the parent of $v_j$ previously. In this situation, $q_j = \text{op}(v_{j_1}, v_j) \circ q_{j_1}$ with probability one, deducing $P_{\text{train}}(\text{op}(v_{j_1}, v_j) \circ q_{j_1} \mid z_{1:T}) = 1$. Otherwise, if $v_j$ is the first vertex from $\mathcal{V}$ in this sentence, then $q_j$ is sampled from Uniform($\text{VALS}(v_j)$), deducing $P_{\text{train}}(\cdot \mid z_{1:T}) = \text{Uniform}(\text{VALS}(v_j))$.

- When $z = q_j$, the next token must be $o^{\text{cm}}$, deducing $P_{\text{train}}(o^{\text{cm}} \mid z_{1:T}) = 1$.

In any situation, $P_{\text{prog}}(\cdot \mid z_{1:T})$ always equals to $P_{\text{train}}(\cdot \mid z_{1:T})$.

### D.3 PROOF OF LEMMA 5.2

First write down $\widetilde{\text{inp}}^k$ and $\widetilde{\text{lab}}^k$:

$$\widetilde{\text{seq}} = [v_L, o^{\text{eq}}, o^{\text{qu}}, c(v_1)_1, \ldots, c(v_1)_{l_1}, v_1, o^{\text{eq}}, q_1, o^{\text{cm}}, \ldots, o^{\text{cm}}, c(v_L)_1, \ldots, c(v_L)_{l_L}, v_L, o^{\text{eq}}, q_L]$$

$$\widetilde{\text{doc}}^k = [\widetilde{\text{seq}}^{(1)}, \backslash\text{n}, \ldots, \backslash\text{n}, \widetilde{\text{seq}}^{(k)}]$$

$$\widetilde{\text{inp}}^k = \widetilde{\text{doc}}^k + [\backslash\text{n}] + [v_L, o^{\text{eq}}, o^{\text{qu}}, c(v_1)_1^{(k+1)}, \ldots, c(v_1)_{l_1}^{(k+1)}, v_1, o^{\text{eq}}, q_1^{(k+1)}, o^{\text{cm}}]$$

$$\widetilde{\text{lab}}^k = [c(v_2)_1^{(k+1)}, \ldots, c(v_2)_{l_2}^{(k+1)}, v_2, o^{\text{eq}}, q_2^{(k+1)}, o^{\text{cm}}, \ldots, o^{\text{cm}}, c(v_L)_1^{(k+1)}, \ldots, c(v_L)_{l_L}^{(k+1)}, v_L, o^{\text{eq}}, q_L^{(k+1)}]$$

After decoding, we have $\text{dec}(\widetilde{\text{lab}}^k) = [v_2, q_2^{(k+1)}, \cdots, v_N, q_N^{(k+1)}]$. According to the program, we have

$$P_{\text{prog}}(\cdot \mid \widetilde{\text{inp}}^k) = \text{Uniform}(\text{CONT}(v_2)).$$

In this situation, the greedy decoding will randomly sample a $c(v_2)_1^k$ uniformly and concatenate it to the end of $\text{inp}^k$, then we have that

$$P_{\text{prog}}(\cdot \mid [\widetilde{\text{inp}}^k, c(v_2)_1^k, \cdots, c(v_2)_i^k]) = \text{Uniform}(\text{CONT}(v_2) + \{v_2\} - \{c(v_2)_1^k, \cdots, c(v_2)_i^k\}).$$

It will keep sampling elements from $\text{CONT}(v_2)$ until it samples the vertex $v_2$. Suppose that it samples $r_2^k$ times until it samples $v_2$. Then we have that

$$\arg\max P_{\text{prog}}(\cdot \mid [\widetilde{\text{inp}}^k, c(v_2)_1^{(k+1)}, \cdots, c(v_2)_{r_2^{k+1}}^{(k+1)}, v_2]) = o^{\text{eq}}$$

$$\arg\max P_{\text{prog}}(\cdot \mid [\widetilde{\text{inp}}^k, c(v_2)_1^{(k+1)}, \cdots, c(v_2)_{r_2^{k+1}}^{(k+1)}, v_2, o^{\text{eq}}]) = q_2^{(k+1)}$$

$$\arg\max P_{\text{prog}}(\cdot \mid [\widetilde{\text{inp}}^k, c(v_2)_1^{(k+1)}, \cdots, c(v_2)_{r_2^{f+1}}^{(k+1)}, v_2, o^{\text{eq}}, q_2^{(k+1)}]) = o^{\text{cm}}$$

Following such routine, the greedy decoding on algorithm 1 we finally sample

$$\text{prog}(\widetilde{\text{inp}}^k) = [c(v_2)_1^{(k+1)}, \cdots, c(v_2)_{r_2^{k+1}}^{(k+1)}, v_2, o^{\text{eq}}, q_2^{(k+1)}, \cdots, c(v_N)_1^{(k+1)}, \cdots, c(v_N)_{r_N^{k+1}}^{(k+1)}, v_N, o^{\text{eq}}, q_N^{(k+1)}].$$

So we have

$$\text{dec}(\text{prog}(\widetilde{\text{inp}}^k)) = [v_2, q_2^{(k+1)}, \cdots, v_N, q_N^{(k+1)}] = \text{dec}(\widetilde{\text{lab}}^k).$$

## E TRANSFORMER ARCHITECTURE

Here we construct a simplified Transformer architecture that is convenient for theoretical analysis. Denote the set $\mathcal{A}$ as the set of all possible tokens that may appear in our task. Given any input sentence $z_{1:T} \in \mathcal{A}^T$, it is processed by the following structures.

### E.1 EMBEDDINGS

**Token Embeddings.** The token embedding $W_{\text{tok}} : \mathcal{A} \to \mathbb{R}^{d_{\text{tok}}}$ is a mapping that maps each possible token $z \in \mathcal{A}$ to a $d_{\text{tok}}$ dimensional real vector $W_{\text{tok}}(z)$.

**Positional Embeddings.** For each $z_t$ that is the $t$th token of the sentence, it has a $d_p$ dimensional positional embedding $p_t \in \mathbb{R}^{d_p}$.

**Comprehensive Embeddings.** The comprehensive embedding $W_{\text{comp}} : \mathcal{A} \times [T_{\max}] \to \mathbb{R}^{d_{\text{comp}}}$ is a mapping that maps both the token information $z$ and the positional information $t$ to a $d_{\text{comp}}$ dimensional vectore $W_{\text{comp}}(z, t)$.

To simplify the analysis, we unify the dimensions of different embeddings as $d_1 := d_{\text{tok}} = d_p = d_{\text{comp}}$. Then we regulate that any two different embeddings are orthonormal[3], which means that for all $e_1, e_2 \in \{W_{\text{tok}}(z)\}_{z \in \mathcal{A}} \bigcup \{p_t\}_{t \in [T_{\max}]} \bigcup \{W_{\text{comp}}(z, t)\}_{z \in \mathcal{A}, t \in [T_{\max}]}$ and $e_1 \neq e_2$, we have that $e_1^\top e_1 = 1$ and $e_1^\top e_2 = 0$. Then for any token $z_t$ at the $t$th position of the input sentence $z_{1:T}$, its embedding is

$$W_E(z_t, t) := (\underbrace{W_{\text{tok}}^\top(z_t) + p_t^\top + W_{\text{comp}}^\top(z_t, t)}_{d_1}, \underbrace{0, \cdots, 0}_{4d_1})^\top$$

where the last $4d_1$ dimensional zero vector is for writing in additional information. For simplicity, we denote $x_t := W_E(z_t, t) \in \mathbb{R}^d$ where $d = 5d_1$.

### E.2 ATTENTION MECHANISM

For an $L$-layers $H$-heads Transformer, it has the set $\{(W_Q^{l,h}, W_K^{l,h}, W_V^{l,h})\}_{l=1, h=1}^{L, H}$ as its query, key, value matrices, where $W_Q^{l,h}, W_K^{l,h}, W_V^{l,h} \in \mathbb{R}^{d \times d}$. It also has output matrices $W_{O_1} \in \mathbb{R}^{(|\mathcal{A}|+d_2) \times d}$, $W_{O_2} \in \mathbb{R}^{|\mathcal{A}| \times (|\mathcal{A}|+d_2)}$ and bias $W_B \in \mathbb{R}^{|\mathcal{A}|+d_2}$ to map the hidden state to the logits for output tokens. Here $d_2$ is an extra dimension for information writing in of MLP, whose specific value will be given in appendix F.3. For each embedded input sentence $x_{1:T}$, it goes through the following attention mechanism:

$$x_{1:T}^{(0)} \leftarrow x_{1:T}$$

$$x_t^{(l)} \leftarrow x_t^{(l-1)} + \sum_{h=1}^{H} W_V^{l,h} x_{1:t}^{(l-1)} \text{softmax}\left( \left(x_{1:t}^{(l-1)}\right)^\top \left(W_K^{l,h}\right)^\top W_Q^{l,h} x_t^{(l-1)} \right)$$

$$:= x_t^{(l-1)} + \sum_{h=1}^{H} \text{attn}^{(l,h)}\left(x_{1:t}^{(l-1)}\right), \quad \text{for } t = 1, \cdots, T, l = 1, \cdots, L$$

$$x_t^{\text{out}} \leftarrow W_{O_2} \text{relu}\left(W_{O_1} x_t^{(L)} + W_B\right) := \text{MLP}(x_t^{(L)}), \quad \text{for } t = 1, \cdots, T$$

Define the parameter vector $\theta := \{(W_Q^{l,h}, W_K^{l,h}, W_V^{l,h})\}_{l=1, h=1}^{L, H} \bigcup \{W_{O_1}, W_{O_2}, W_B\}$. Given any input sentence $z_{1:T}$, the output probability of the Transformer with parameter $\theta$ is a distribution over the indices set of all tokens $\mathcal{A}$:

$$P_\theta(\cdot \mid z_{1:T}) := \text{softmax}(x_T^{\text{out}}).$$

Here we assume that the temperature of every $\text{softmax}$ layer here tends to infinity, meaning that $\text{softmax}(x) = \text{Uniform}(\arg\max(x))$.

At test time, the Transformer does greedy decoding, which outputs the token with the largest probability every time. Denoting the decoded sentence output by the Transformer as $\text{model}_\theta(z_{1:T}) := z'_{1, T_1}$, it follows

$$z'_1 = \arg\max P_\theta(\cdot \mid z_{1:T}), \quad z'_i = \arg\max P_\theta(\cdot \mid [z_{1:T}, z'_{1:i}]) \text{ for } i = 2, \cdots T_1.$$

---

[3]In practice, the orthonormality can be approximated by randomly mapping embeddings to high-dimensional Gaussian vectors with low variance (Bietti et al., 2024).

## F  PROOF OF LEMMA 5.3

### F.1  FIRST LAYER

We directly give the construction. We set $W_K^{l,h} = \boldsymbol{I}_d$ for all $l, h$, letting $W_Q^{l,h}$ plays the role of both key and query. Moreover, we set $T_{\max}$ as the maximum length of all possible input sentences. Referring to (Bietti et al., 2024), we construct $\{W_Q^{1,h}\}_{h=1}^4$ as associative memory:

$$
W_Q^{1,1} = \underbrace{\sum_{t=4}^{T_{\max}} (p_{t-3}^\top, \boldsymbol{0}_{4d_1})^\top (W_{\text{comp}}(o^{\text{cm}}, t)^\top, \boldsymbol{0}_{4d_1}) + \sum_{t=4}^{T_{\max}} (p_{t-3}^\top, \boldsymbol{0}_{4d_1})^\top (W_{\text{comp}}(o^{\text{dlm}}, t)^\top, \boldsymbol{0}_{4d_1})}_{z_T = o^{\text{cm}}/o^{\text{dlm}}}
$$
$$
+ \underbrace{\sum_{v \in \mathcal{V}_{\text{all}}} (W_{\text{tok}}(o^{\text{eq}})^\top, \boldsymbol{0}_{4d_1})^\top (W_{\text{tok}}(v)^\top, \boldsymbol{0}_{4d_1})}_{z_T \in \mathcal{V}_{\text{all}}} + \underbrace{\sum_{a \in \mathcal{A} - \mathcal{V}_{\text{all}} - \{o^{\text{cm}}, o^{\text{dlm}}\}} (W_{\text{tok}}(o^{\text{qu}})^\top, \boldsymbol{0}_{4d_1})^\top (W_{\text{tok}}(a)^\top, \boldsymbol{0}_{4d_1})}_{z_T = \text{others}}
$$

$$
W_Q^{1,2} = \underbrace{\sum_{v \in \mathcal{V}_{\text{all}}} \sum_{c_1, c_2 \in \text{CONT}(v)} \sum_{t=1}^{T_{\max}} \sum_{l=0}^{|\text{CONT}(v)|} (W_{\text{comp}}(c_2, t-l)^\top, \boldsymbol{0}_{4d_1})^\top (W_{\text{comp}}(c_1, t)^\top, \boldsymbol{0}_{4d_1})}_{z_T = \text{contextual information}}
$$
$$
+ \underbrace{\sum_{v \in \mathcal{V}_{\text{all}}} \sum_{t_1=5}^{T_{\max}} \sum_{t_2=1}^{t_1-4} t_2 \cdot (W_{\text{comp}}(v, t_2)^\top, \boldsymbol{0}_{4d_1})^\top (W_{\text{comp}}(o^{\text{cm}}, t_1)^\top + W_{\text{comp}}(o^{\text{dlm}}, t_1), \boldsymbol{0}_{4d_1})}_{z_T = o^{\text{cm}}/o^{\text{dlm}}}
$$
$$
+ \underbrace{\sum_{a \in \mathcal{A} - \bigcup_{v \in \mathcal{V}_{\text{all}}} \text{CONT}(v) - \{o^{\text{cm}}, o^{\text{dlm}}\}} (W_{\text{tok}}(o^{\text{qu}})^\top, \boldsymbol{0}_{4d_1})^\top (W_{\text{tok}}(a)^\top, \boldsymbol{0}_{4d_1})}_{z_T = \text{others}}
$$

$$
W_Q^{1,3} = \underbrace{\sum_{t=2}^{T_{\max}} (p_{t-1}^\top, \boldsymbol{0}_{4d_1})^\top (W_{\text{comp}}(o^{\text{eq}}, t)^\top, \boldsymbol{0}_{4d_1})}_{z_T = o^{\text{eq}}} + \underbrace{\sum_{a \in \mathcal{A} - \{o^{\text{eq}}\}} (W_{\text{tok}}(o^{\text{qu}})^\top, \boldsymbol{0}_{4d_1})^\top (W_{\text{tok}}(a)^\top, \boldsymbol{0}_{4d_1})}_{z_T = \text{others}}
$$

$$
W_Q^{1,4} = \underbrace{\sum_{q \in \bigcup_{v \in \mathcal{V}_{\text{all}}} \text{VALS}(v)} \sum_{t=3}^{T_{\max}} (p_{t-2}^\top, \boldsymbol{0}_{4d_1})^\top (W_{\text{comp}}(q, t)^\top, \boldsymbol{0}_{4d_1})}_{z_T = \text{values}}
$$
$$
+ \underbrace{\sum_{a \in \mathcal{A} - \bigcup_{v \in \mathcal{V}_{\text{all}}} \text{VALS}(v)} (W_{\text{tok}}(o^{\text{qu}})^\top, \boldsymbol{0}_{4d_1})^\top (W_{\text{tok}}(a)^\top, \boldsymbol{0}_{4d_1})}_{z_T = \text{others}}
$$

We also construct $\{W_V^{1,h}\}_{h=1}^4$ as

$$
W_V^{1,1} = \sum_{v \in \mathcal{V}_{\text{all}}} (\boldsymbol{0}_{d_1}, W_{\text{tok}}(v)^\top, \boldsymbol{0}_{3d_1})^\top (W_{\text{tok}}(v)^\top, \boldsymbol{0}_{4d_1}) + (\boldsymbol{0}_{3d_1}, W_{\text{tok}}(o^{\text{eq}})^\top, \boldsymbol{0}_{d_1})^\top (W_{\text{tok}}(o^{\text{eq}})^\top, \boldsymbol{0}_{4d_1})
$$

$$
W_V^{1,2} = \sum_{v \in \mathcal{V}_{\text{all}}} \sum_{c \in \text{CONT}(v)} (\boldsymbol{0}_{3d_1}, W_{\text{tok}}(c)^\top, \boldsymbol{0}_{d_1})^\top (W_{\text{tok}}(c)^\top, \boldsymbol{0}_{4d_1}) + \sum_{v \in \mathcal{V}_{\text{all}}} (\boldsymbol{0}_{2d_1}, W_{\text{tok}}(v)^\top, \boldsymbol{0}_{2d_1})^\top (W_{\text{tok}}(v)^\top, \boldsymbol{0}_{4d_1})
$$

$$
W_V^{1,3} = \sum_{v \in \mathcal{V}_{\text{all}}} \sum_{t}^{T_{\max}} (\boldsymbol{0}_{d_1}, W_{\text{comp}}(v, t)^\top, \boldsymbol{0}_{2d_1}, W_{\text{tok}}(v)^\top)^\top (W_{\text{comp}}(v, t)^\top, \boldsymbol{0}_{4d_1})
$$

$$
W_V^{1,4} = \sum_{v \in \mathcal{V}_{\text{all}}} \sum_{t}^{T_{\max}} (\boldsymbol{0}_{2d_1}, W_{\text{comp}}(v, t)^\top, \boldsymbol{0}_{2d_1})^\top (W_{\text{comp}}(v, t)^\top, \boldsymbol{0}_{4d_1})
$$

Using the above keys and values to calculate the hidden states directly is a little bit abstract. As a result, we explain the intuition of our construction taking $\text{attn}^{1,1}$ and $\text{attn}^{1,2}$ as examples. For any input sequence $z_{1:T}$ and its embedding $x_{1:T}$, we first analyze $\text{attn}^{1,1}(x_{1:T})$:

- If $z_T = o^{\text{cm}}/o^{\text{dlm}}$, it will be caught by the first two items in $W_Q^{1,1}$ and attend to its corresponding vertex $z_{T-3} = v_{j-1}$. The value matrix $W_V^{1,1}$ maps the value vector of each vertex $v \in \mathcal{V}_{\text{all}}$ to $(\mathbf{0}_{d_1}, W_{\text{tok}}(v)^\top, \mathbf{0}_{3d_1})^\top$. Hence, we have that $\text{attn}^{1,1}(x_{1:T}) = (\mathbf{0}_{d_1}, W_{\text{tok}}(v_{j-1})^\top, \mathbf{0}_{3d_1})^\top$.

- If $z_T \in \mathcal{V}_{\text{all}}$, it will be caught by the third item in $W_Q^{1,1}$, attending to the equation token $o^{\text{eq}}$. The value matrix $W_V^{1,1}$ maps the value vector of each $o^{\text{eq}}$ in the sentence to $(\mathbf{0}_{d_1}, W_{\text{tok}}(o^{\text{eq}})^\top, \mathbf{0}_{3d_1})^\top$. Hence, we have that $\text{attn}^{1,1}(x_{1:T}) = (\mathbf{0}_{d_1}, W_{\text{tok}}(o^{\text{eq}})^\top, \mathbf{0}_{3d_1})^\top$.

- If $z_T$ is other type of token, it will be caught by the last item in $W_Q^{1,1}$, attending to the question mark $o^{\text{qu}}$. The value matrix $W_V^{1,1}$ maps $o^{\text{qu}}$ to zero vector $(\mathbf{0}_{5d_1})^\top$. Hence, we have that $\text{attn}^{1,1}(x_{1:T}) = (\mathbf{0}_{5d_1})^\top$.

Then we analyze $\text{attn}^{1,2}(x_{1:T})$:

- If $z_T = c(v_j)_l$, it will be caught by the first item in $W_Q^{1,2}$, attending to the previous contextual tokens that are also generated by $v_j$ including itself: $\{c(v_j)_{l'}\}_{l'=1}^l$. The value matrix $W_V^{1,1}$ maps every contextual token $c$ to $(\mathbf{0}_{3d_1}, W_{\text{tok}}(c)^\top, \mathbf{0}_{d_1})^\top$. Hence, we have that $\text{attn}^{1,2}(x_{1:T}) = (\mathbf{0}_{3d_1}, \frac{1}{l} \sum_{l'=1}^l W_{\text{tok}}(c(v_j)_{l'})^\top, \mathbf{0}_{d_1})^\top$.

- If $z_T = o^{\text{cm}}/o^{\text{dlm}}$, it will be caught by the second item in $W_Q^{1,2}$, attending to all the previous tokens that are also vertices except for the nearest one $v_{j-1} = z_{T-3}$. Each token is multiplied by a weight $t_2$, the nearer the token is to $z_T$, the higher the weight will be. Hence, as the temperature of the softmax goes to infinity, $z_T$ attends to the second nearest vertex to it, which is denoted by $v_{j-2}$. The value matrix $W_V^{1,2}$ maps the value vector of each vertex $v \in \mathcal{V}_{\text{all}}$ to $(\mathbf{0}_{2d_1}, W_{\text{tok}}(v)^\top, \mathbf{0}_{2d_1})^\top$. Hence, we have that $\text{attn}^{1,2}(x_{1:T}) = (\mathbf{0}_{2d_1}, W_{\text{tok}}(v_{j-2})^\top, \mathbf{0}_{2d_1})^\top$.

- If $z_T$ is other type of token, it will be caught by the last item in $W_Q^{1,2}$, attending to the question mark $o^{\text{qu}}$. The value matrix $W_V^{1,2}$ maps $o^{\text{qu}}$ to zero vector $(\mathbf{0}_{5d_1})^\top$. Hence, we have that $\text{attn}^{1,2}(x_{1:T}) = (\mathbf{0}_{5d_1})^\top$.

The hidden states generated by $\text{attn}^{1,3}$ and $\text{attn}^{1,4}$ can be calculated following the same routine. Finally we can summarize that for any input sequence $z_{1:T}$ and its embedding $x_{1:T}$, we have

$$
x_T^{(1)} = x_T + \sum_{h=1}^4 \text{attn}^{1,h}(x_{1:T})
$$
$$
= \begin{cases}
1. (W_E(z_T, T)^\top, W_{\text{tok}}(v_{j-1})^\top, W_{\text{tok}}(v_{j-2})^\top, \mathbf{0}_{2d_1})^\top, & \text{if } z_T = o^{\text{cm}}/o^{\text{dlm}} \text{ and } z_{T-3} = v_{j-1} \\
\quad \text{If } j = 2, \text{ then } v_{j-2} \text{ is the last vertex of the sentence, otherwise it is the vertex before } v_{j-1}. \\
2. (W_E(z_T, T)^\top, \mathbf{0}_{2d_1}, \frac{1}{l} \sum_{l'=1}^l W_{\text{tok}}(c(v_j)_{l'})^\top, \mathbf{0}_{d_1})^\top, & \text{if } z_T = c(v_j)_l \\
3. (W_E(z_T, T)^\top, \mathbf{0}_{2d_1}, W_{\text{tok}}(o^{\text{eq}})^\top, \mathbf{0}_{d_1})^\top, & \text{if } z_T = v_j \\
4. (W_E(z_T, T)^\top, W_{\text{comp}}(v_j, T-1)^\top, \mathbf{0}_{2d_1}, W_{\text{tok}}(v_j)^\top)^\top, & \text{if } z_T = o^{\text{eq}} \text{ and } z_{T-1} = v_j \\
5. (W_E(z_T, T)^\top, \mathbf{0}_{d_1}, W_{\text{comp}}(v_j, T-2)^\top, \mathbf{0}_{2d_1})^\top, & \text{if } z_T = q_j \text{ and } z_{T-2} = v_j.
\end{cases}
$$

### F.2 SECOND LAYER

Then we construct the second layer $\{W_Q^{2,h}, W_V^{2,h}\}_{h=1}^2$. First is the queries:

$$W_Q^{2,1} = \underbrace{\sum_{\substack{q \in \bigcup_{v \in \mathcal{V}_{\text{all}}} \text{VALS}(v)}} \sum_{t=1}^{T_{\max}} (p_t^\top, \mathbf{0}_{4d_1})^\top (W_{\text{comp}}(q,t)^\top, \mathbf{0}_{4d_1})}_{z_T = q_j}$$

$$+ \underbrace{\sum_{v \in \mathcal{V}_{\text{all}}} (\mathbf{0}_{2d_1}, W_{\text{tok}}(v)^\top, \mathbf{0}_{2d_1})^\top (\mathbf{0}_{d_1}, W_{\text{tok}}(v)^\top, \mathbf{0}_{3d_1})}_{z_T = o^{\text{cm}}, o^{\text{dlm}}}$$

$$+ \sum_{a \in \mathcal{A} - \bigcup_{v \in \mathcal{V}_{\text{all}}} \text{VALS}(v) - \{o^{\text{cm}}, o^{\text{dlm}}\}} (W_{\text{tok}}(o^{\text{qu}})^\top, \mathbf{0}_{4d_1})^\top (W_{\text{tok}}(a)^\top, \mathbf{0}_{4d_1})$$

$$W_Q^{2,2} = \underbrace{\sum_{v_1 \neq v_2 \in \mathcal{V}} \sum_{t_2=2}^{T_{\max}} \sum_{t_1=1}^{t_2-1} t_1 \cdot (W_{\text{comp}}(o^{\text{dlm}}, t_1)^\top, \mathbf{0}_{d_1}, W_{\text{comp}}(v_1, t_1)^\top, \mathbf{0}_{2d_1})^\top (\mathbf{0}_{d_1}, W_{\text{comp}}(v_2, t_2)^\top, \mathbf{0}_{3d_1})}_{z_T = o^{\text{eq}}}$$

$$+ (W_{\text{tok}}(o^{\text{qu}})^\top, \mathbf{0}_{4d_1})^\top \left( \sum_{v \in \mathcal{V}_{\text{all}} - \mathcal{V}} \sum_{t=1}^{T_{\max}} (\mathbf{0}_{d_1}, W_{\text{comp}}(v,t)^\top, \mathbf{0}_{3d_1}) + \sum_{a \in \mathcal{A} - \{o^{\text{eq}}\}} (W_{\text{tok}}(a)^\top, \mathbf{0}_{4d_1}) \right)$$

Then is the values:

$$W_V^{2,1} = \sum_{q \in \bigcup_{v \in \mathcal{V}_{\text{all}}} \text{VALS}(v)} (\mathbf{0}_{3d_1}, W_{\text{tok}}(q)^\top, \mathbf{0}_{d_1})^\top (W_{\text{tok}}(q)^\top, \mathbf{0}_{4d_1}) + \sum_{v \in \mathcal{V}_{\text{all}}} (\mathbf{0}_{3d_1}, W_{\text{tok}}(v)^\top, \mathbf{0}_{d_1})^\top (\mathbf{0}_{d_1}, W_{\text{tok}}(v)^\top, \mathbf{0}_{3d_1})$$

$$W_V^{2,2} = (\mathbf{0}_{4d_1}, \sqrt{2} W_{\text{tok}}(o^{\text{dlm}})^\top)^\top (W_{\text{tok}}(o^{\text{dlm}})^\top, \mathbf{0}_{4d_1}) + \sum_{q \in \bigcup_{v \in \mathcal{V}_{\text{all}}} \text{VALS}(v)} \sum_{t=1}^{T_{\max}} (\mathbf{0}_{4d_1}, W_{\text{tok}}(q)^\top)^\top (W_{\text{comp}}(q,t)^\top, \mathbf{0}_{4d_1})$$

$$+ \sum_{v \in \mathcal{V}_{\text{all}}} \sum_{t=1}^{T_{\max}} (\mathbf{0}_{4d_1}, W_{\text{tok}}(v)^\top)^\top (\mathbf{0}_{2d_1}, W_{\text{comp}}(v,t)^\top, \mathbf{0}_{2d_1})$$

For any input sequence $z_{1:T}$ and its embedding $x_{1:T}$, we first analyze $\text{attn}^{2,1}(x_{1:T})$:

- If $z_T = q_j$, it will be caught by the first item in $W_Q^{2,1}$, attending to itself. The value matrix $W_V^{2,1}$ maps each $q$ to the vector $(\mathbf{0}_{3d_1}, W_{\text{tok}}(q)^\top, \mathbf{0}_{d_1})$. Hence, we have that $\text{attn}^{2,1}(x_{1:T}^{(1)}) = (\mathbf{0}_{3d_1}, W_{\text{tok}}(q_j)^\top, \mathbf{0}_{d_1})$.

- If $z_T = o^{\text{cm}}/o^{\text{dlm}}$, it will be caught by the second item in $W_Q^{2,1}$. It attends to a previous token $z_{T_1}$ whose value is equal to $z_T$ (which means it is also a comma or delimiter). Denote $v_{j-1} = z_{T-3}$ as the corresponding vertex of $z_T$, what is special is that $z_{T_1-3} = v_j$. That means that $z_{T_1}$ is the comma (or delimiter) of the next token of $v_j$ in a previous shot. The reason why $z_T$ can attend to $z_{T_1}$ is because of their hidden states:

$$x_T^{(1)} = (W_E(z_T, T)^\top, \underline{W_{\text{tok}}(v_{j-1})^\top}, W_{\text{tok}}(v_{j-2})^\top, \mathbf{0}_{2d_1})^\top$$
$$x_{T_1}^{(1)} = (W_E(z_{T_1}, T_1)^\top, W_{\text{tok}}(v_j)^\top, \underline{W_{\text{tok}}(v_{j-1})^\top}, \mathbf{0}_{2d_1})^\top$$

  That means $(x_T^{(1)})_{d_1+1:2d_1} = (x_{T_1}^{(1)})_{2d_1+1:3d_1}$, which is caught by the second item in $W_Q^{2,1}$. Such a mechanism is known as induction head Giannou et al. (2023); Olsson et al. (2022); Bietti et al. (2024). The value matrix $W_V^{2,1}$ maps each $o^{\text{cm}}/o^{\text{dlm}}$ to $(\mathbf{0}_{3d_1}, W_{\text{tok}}(v_{j-1})^\top, \mathbf{0}_{d_1})^\top$, where $v_{j-1}$ is its corresponding vertex. Hence, we have that $\text{attn}^{2,1}(x_{1:T}^{(1)}) = (\mathbf{0}_{3d_1}, W_{\text{tok}}(v_j)^\top, \mathbf{0}_{d_1})^\top$.

- If $z_T$ is other type of token, it will be caught by the last item in $W_Q^{2,1}$, attending to the question mark $o^{\mathrm{qu}}$. The value matrix $W_V^{2,1}$ maps $o^{\mathrm{qu}}$ to zero vector $(\mathbf{0}_{5d_1})^\top$. Hence, we have that $\mathrm{attn}^{2,1}(x_{1:T}^{(1)}) = (\mathbf{0}_{5d_1})^\top$.

Then we analyze $\mathrm{attn}^{2,2}(x_{1:T})$:

- If $z_T = o^{\mathrm{eq}}$, it will be caught by the first item in $W_Q^{2,2}$. Denote the corresponding vertex of $z_T$ as $v_j = z_{T-1}$. If $v_j \in \mathcal{V}_{\mathrm{all}} - \mathcal{V}$, then it attends to the question mark $o^{\mathrm{qu}}$, being same as the second situation. If $v_j \in \mathcal{V}$, it attends to the nearest delimiter $o^{\mathrm{dlm}}$ or the nearest value $q_{j_1}$ whose corresponding vertex $v_{j_1} \in \mathcal{V}$. If the nearest is $o^{\mathrm{dlm}}$, that means $v_j$ has no parent in the sentence, we have that $\mathrm{attn}^{2,2}(x_{1:T}^{(1)}) = (\mathbf{0}_{4d_1}, \sqrt{2}W_{\mathrm{tok}}(o^{\mathrm{dlm}})^\top)^\top$. If the nearest is $q_{j_1}$, that means $v_j$ has a parent $v_{j_1}$ in the sentence, we have that $\mathrm{attn}^{2,2}(x_{1:T}^{(1)}) = (\mathbf{0}_{4d_1}, W_{\mathrm{tok}}(v_{j_1})^\top + W_{\mathrm{tok}}(q_{j_1})^\top)^\top$.

- If $z_T$ is other type of token, it will be caught by the last item in $W_Q^{2,1}$, attending to the question mark $o^{\mathrm{qu}}$. The value matrix $W_V^{2,1}$ maps $o^{\mathrm{qu}}$ to zero vector $(\mathbf{0}_{5d_1})^\top$. Hence, we have that $\mathrm{attn}^{2,1}(x_{1:T}^{(1)}) = (\mathbf{0}_{5d_1})^\top$.

To summarize, we have that

$$
x_T^{(2)} = x_T^{(1)} + \sum_{h=1}^2 \mathrm{attn}^{(2,h)}\left(x_{1:T}^{(1)}\right)
$$
$$
= \begin{cases}
\mathbf{1}.(W_E(o^{\mathrm{cm}}, T)^\top, W_{\mathrm{tok}}(v_{j-1})^\top, W_{\mathrm{tok}}(v_{j-2})^\top, W_{\mathrm{tok}}(v_j)^\top, \mathbf{0}_{d_1})^\top, & \text{if } z_T = o^{\mathrm{cm}} \text{ and } z_{T-3} = v_{j-1} \\
\mathbf{2}.(W_E(c(v_j)_l, T)^\top, \mathbf{0}_{2d_1}, \frac{1}{l}\sum_{l'=1}^l W_{\mathrm{tok}}(c(v_j)_{l'})^\top, \mathbf{0}_{d_1})^\top, & \text{if } z_T = c(v_j)_l \\
\mathbf{3}.(W_E(v_j, T)^\top, \mathbf{0}_{2d_1}, W_{\mathrm{tok}}(o^{\mathrm{eq}})^\top, \mathbf{0}_{d_1})^\top, & \text{if } z_T = v_j \\
\mathbf{4.1}.(W_E(o^{\mathrm{eq}}, T)^\top, W_{\mathrm{comp}}(v_j, T-1)^\top, \mathbf{0}_{2d_1}, W_{\mathrm{tok}}(v_j)^\top + \sqrt{2}W_{\mathrm{tok}}(o^{\mathrm{dlm}})^\top)^\top \\
\quad \text{if } z_T = o^{\mathrm{eq}} \text{ and } z_{T-1} = v_j \text{ has no parent in the sentence} \\
\mathbf{4.2}.(W_E(o^{\mathrm{eq}}, T)^\top, W_{\mathrm{comp}}(v_j, T-1)^\top, \mathbf{0}_{2d_1}, W_{\mathrm{tok}}(v_j)^\top + W_{\mathrm{tok}}(v_{j_1})^\top + W_{\mathrm{tok}}(q_{j_1})^\top)^\top \\
\quad \text{if } z_T = o^{\mathrm{eq}} \text{ and } v_{j_1} \text{ is the parent of } z_{T-1} = v_j \\
\mathbf{5}.(W_E(q_j, T)^\top, \mathbf{0}_{d_1}, W_{\mathrm{comp}}(v_j, T-2)^\top, W_{\mathrm{tok}}(q_j)^\top, \mathbf{0}_{d_1})^\top, & \text{if } z_T = q_j
\end{cases}
$$

### F.3 OUTPUT LAYER

The output layer is a two-layer MLP whose input dimension is $5d_1$, hidden dimension is $|\mathcal{A}| + d_2$ and output dimension is $|\mathcal{A}|$. Here the extra dimension $d_2$ is for mapping elements in the set

$$
\mathcal{B} := \mathcal{V} \bigcup \left( \bigcup_{(v_1,v_2)\in\mathcal{E}} \bigcup_{q_1\in\mathrm{VALS}(v_1)} \{(v_1, v_2, q_1)\} \right),
$$

so $d_2 = |\mathcal{B}|$. For simplicity of the notation, we here introduce shorthands of several $|\mathcal{A}|$-dimensional vectors. There is an one-one mapping from the set of all tokens $\mathcal{A}$ to their indices $[|\mathcal{A}|]$. Specifically, each token $a \in \mathcal{A}$ has its unique index $\mathrm{idx}(a) \in [|\mathcal{A}|]$. Let $e^{(i)}$ be an $|\mathcal{A}|$-dimensional vector where a 1 at position $i$ and 0 at other positions. With a slight misuse of the notation, we define $e(a) := e^{(\mathrm{idx}(a))}$ for all $a \in \mathcal{A}$. For any subset $\mathcal{A}' \subset \mathcal{A}$, we denote the uniform logit on it as $e_{\mathrm{uni}}(\mathcal{A}') := \sum_{a\in\mathcal{A}'} e(a)$. For the extra set $\mathcal{B}$, we define $g^{(i)}$ be a $|\mathcal{B}|$-dimensional vector where a 1 at position $i$ and 0 at other positions. Similarly, we can define $g(b) := g^{(\mathrm{idx}(b))}$ for all $b \in \mathcal{B}$.

Denoting the set of vertices in $\mathcal{V}$ which does not have parents as $\mathcal{V}_{\text{init}}$, we can define the output matrices as

$$
W_{O_1} = \underbrace{\sum_{v \in \mathcal{V}_{\text{all}}} (e_{\text{uni}}(\text{CONT}(v))^\top, \mathbf{0}_{d_2})^\top (\mathbf{0}_{3d_1}, W_{\text{tok}}(v)^\top, \mathbf{0}_{d_1})}_{1.\, z_T = o^{\text{cm}}}
$$

$$
+ \underbrace{\sum_{v \in \mathcal{V}_{\text{all}}} \sum_{c \in \text{CONT}(v)} (e_{\text{uni}}(\text{CONT}(v))^\top + e(v)^\top - e(c)^\top, \mathbf{0}_{d_2})^\top (\mathbf{0}_{3d_1}, W_{\text{tok}}(c)^\top, \mathbf{0}_{d_1})}_{2.\, z_T = c(v_j)_l}
$$

$$
+ \underbrace{(e(o^{\text{eq}})^\top, \mathbf{0}_{d_2})^\top (\mathbf{0}_{3d_1}, W_{\text{tok}}(o^{\text{eq}}), \mathbf{0}_{d_1})}_{3.\, z_T \in \mathcal{V}}
$$

$$
+ \underbrace{\sum_{(v_1, v_2) \in \mathcal{E}} \sum_{q_1 \in \text{VALS}(v_1)} (\mathbf{0}_{|\mathcal{A}|}, g((v_1, v_2, q_1))^\top)^\top (\mathbf{0}_{4d_1}, W_{\text{tok}}(v_2)^\top + W_{\text{tok}}(v_1)^\top + W_{\text{tok}}(q_1)^\top)}_{4.\, z_T = o^{\text{eq}}}
$$

$$
+ \underbrace{\sum_{v \in \mathcal{V}} (\mathbf{0}_{|\mathcal{A}|}, g(v)^\top)^\top (\mathbf{0}_{4d_1}, W_{\text{tok}}(v)^\top + \sqrt{2} W_{\text{tok}}(o^{\text{dlm}})^\top)}_{5.\, z_T = o^{\text{eq}}}
$$

$$
+ \underbrace{\sum_{v \in \mathcal{V}_{\text{all}} - \mathcal{V}} (e_{\text{uni}}(\text{VALS}(v))^\top, \mathbf{0}_{d_2})^\top (\mathbf{0}_{4d_1}, W_{\text{tok}}(v)^\top)}_{6.\, z_T = o^{\text{eq}}}
$$

$$
+ \underbrace{\sum_{q \in \bigcup_{v \in \mathcal{V}_{\text{all}}} \text{VALS}(v)} (e(o^{\text{cm}})^\top, \mathbf{0}_{d_2})^\top (\mathbf{0}_{3d_1}, W_{\text{tok}}(q)^\top, \mathbf{0}_{d_1})}_{7.\, z_T = q_j}
$$

$$
W_B = (\mathbf{0}_{|\mathcal{A}|}, -\mathbf{2}_{d_2})
$$

$$
W_{O_2} = \sum_{v \in \mathcal{V}_{\text{all}}} \left( e(v)(e(v)^\top, \mathbf{0}_{d_2}) + \sum_{c \in \text{CONT}(v)} e(c)^\top (e(c)^\top, \mathbf{0}_{d_2}) \right)
$$

$$
+ e(o^{\text{eq}})(e(o^{\text{eq}})^\top, \mathbf{0}_{d_2})
$$

$$
+ \sum_{(v_1, v_2) \in \mathcal{E}} \sum_{q_1 \in \text{VALS}(v_1)} e(\text{op}(v_1, v_2) \circ q_1)(\mathbf{0}_{|\mathcal{A}|}, g((v_1, v_2, q_1))^\top)
$$

$$
+ \sum_{v \in \mathcal{V}} e_{\text{uni}}(\text{VALS}(v))(\mathbf{0}_{|\mathcal{A}|}, g(v)^\top)
$$

$$
+ \sum_{q \in \bigcup_{v \in \mathcal{V}_{\text{all}}} \text{VALS}(v)} e(q)(e(q)^\top, \mathbf{0}_{d_2})
$$

$$
+ e(o^{\text{cm}})(e(o^{\text{cm}})^\top, \mathbf{0}_{d_2})
$$

Here we only explain the procedure of deriving $\text{MLP}(x_T^{(2)})$ when $z_T = o^{\text{eq}}$, the other situations are straightforward. Denoting $v_j = z_{T-1}$ as its corresponding vertex, we have that

- If $v_j \in \mathcal{V}_{\text{all}} - \mathcal{V}$, it will be caught by item 6 in $W_{O_1}$. It is then easy to show that $\text{MLP}(x_T^{(2)}) = e_{\text{uni}}(\text{VALS}(v_j))$.

- If $v_j \in \mathcal{V}$ but has no parents in previous sentence, it will be caught by item $4, 5$ in $W_{O_1}$. We can calculate that

$$
W_{O_1} x_T^{(2)} = 3(\mathbf{0}_{|\mathcal{A}|}, g(v_j)^\top)^\top + \sum_{b \in \mathcal{B} - \{v_j\}} \mu_b (\mathbf{0}_{|\mathcal{A}|}, g(b)^\top)^\top, \quad \text{where } \mu_b \le 2.
$$

That means all other $g(b)$ will be filtered by the relu except for $g(v_j)$. Specifically we have that $\mathrm{relu}(W_{O_1} x_T^{(2)} + W_B) = 3(\mathbf{0}_{|\mathcal{A}|}, g(v_j)^\top)^\top$. Then it will be caught by the 4th item in $W_{O_2}$, resulting in $\mathrm{MLP}(x_T^{(2)}) = e_{\mathrm{uni}}(\mathrm{VALS}(v_j))$.

- If $v_j \in \mathcal{V}$ and has a parent $v_{j_1}$, it will be caught by item $4, 5$ in $W_{O_1}$. We can calculate that

$$W_{O_1} x_T^{(2)} = 3(\mathbf{0}_{|\mathcal{A}|}, g((v_{j_1}, v_j, q_{j_1}))^\top)^\top + \sum_{b \in \mathcal{B} - \{(v_{j_1}, v_j, q_{j_1})\}} \mu_b (\mathbf{0}_{|\mathcal{A}|}, g(b)^\top)^\top, \quad \text{where } \mu_b \le 2.$$

Then we have $\mathrm{relu}(W_{O_1} x_T^{(2)} + W_B) = 3(\mathbf{0}_{|\mathcal{A}|}, g((v_{j_1}, v_j, q_{j_1}))^\top)^\top$. It will be caught by the 3rd item in $W_{O_2}$, resulting in $\mathrm{MLP}(x_T^{(2)}) = e(\mathrm{op}(v_{j_1}, v_j) \circ q_{j_1})$.

In summary, we have that

$$\mathrm{MLP}(x_T^{(2)}) = \begin{cases} \mathbf{1}. e_{\mathrm{uni}}(\mathrm{CONT}(v_j)), & \text{if } z_T = o^{\mathrm{cm}} \text{ and } z_{T-3} = v_{j-1} \\ \mathbf{2}. (e_{\mathrm{uni}}(\mathrm{CONT}(v_j)) + e(v_j) - \frac{1}{l}\sum_{l'=1}^{l} e(c(v_j)_{l'})), & \text{if } z_T = c(v_j)_l \\ \mathbf{3}. e(o^{\mathrm{eq}}), & \text{if } z_T = v_j \\ \mathbf{4.1}. e(\mathrm{op}(v_{j_1}, v_j) \circ q_{j_1}), & \text{if } z_T = o^{\mathrm{eq}} \text{ and } v_j = z_{T-1} \text{ has a parent } v_{j_1} \text{ in previous sentence} \\ \mathbf{4.2}. e_{\mathrm{uni}}(\mathrm{VALS}(v_j)), & \text{if } z_T = o^{\mathrm{eq}} \text{ and } v_j = z_{T-1} \in \mathcal{V} \text{ but has no parents in previous sentence} \\ \mathbf{4.3}. e_{\mathrm{uni}}(\mathrm{VALS}(v_j)), & \text{if } z_T = o^{\mathrm{eq}} \text{ and } v_j \in \mathcal{V}_{\mathrm{all}} - \mathcal{V} \\ \mathbf{5}. e(o^{\mathrm{cm}}), & \text{if } z_T = q_j \end{cases}$$

As the temperature of the softmax goes to infinity, we have that

$$\mathrm{softmax}\left(\mathrm{MLP}(x_T^{(2)})\right)$$

$$\approx \begin{cases} \mathbf{1}. \mathrm{Uniform}(\mathrm{CONT}(v_j)), & \text{if } z_T = o^{\mathrm{cm}} \text{ and } z_{T-3} = v_{j-1} \\ \mathbf{2}. \mathrm{Uniform}(\mathrm{CONT}(V_j) + \{v_j\} - \{c(v_j)_{l'}\}_{l'=1}^{l}), & \text{if } z_T = c(v_j)_l \\ \mathbf{3}. P(\cdot) \text{ where } P(o^{\mathrm{eq}}) = 1, & \text{if } z_T = v_j \\ \mathbf{4.1}. P(\cdot) \text{ where } P(\mathrm{op}(v_{j_1}, v_j) \circ q_{j_1}) = 1, & \text{if } z_T = o^{\mathrm{eq}} \text{ and } v_j = z_{T-1} \text{ has a parent } v_{j_1} \text{ in previous sentence} \\ \mathbf{4.2}. \mathrm{Uniform}(\mathrm{VALS}(v_j)), & \text{if } z_T = o^{\mathrm{eq}} \text{ and } v_j = z_{T-1} \in \mathcal{V} \text{ but has no parents in previous sentence} \\ \mathbf{4.3}. \mathrm{Uniform}(\mathrm{VALS}(v_j)), & \text{if } z_T = o^{\mathrm{eq}} \text{ and } v_j \in \mathcal{V}_{\mathrm{all}} - \mathcal{V} \\ \mathbf{5}. P(\cdot) \text{ where } P(o^{\mathrm{cm}}) = 1, & \text{if } z_T = q_j. \end{cases}$$

That is exactly the same distribution given by algorithm 1. Depending on how high the temperature of the softmax layer is, the Transformer structure can approximate algorithm 1 with arbitrary preciseness.

## G    IMPLEMENTATION DETAILS

### G.1    TRANSFORMER

Our Transformer model is based on the GPT-2 architecture as implemented by Hugging Face (Wolf et al., 2020; Radford et al., 2019). We tested various combinations of layers and heads, as detailed in Section 4. The model features 720-dimensional embeddings and a context window of 2048 tokens. During encoding, we generate token indices directly and feed them into the model, adhering to the mapping provided by GPT2Tokenizer for both encoding and decoding. The AdamW optimizer is employed with a learning rate of 5e-5.

### G.2    MLP FOR LEARNING FTCT

Given that a 1-shot CoT prompt suffices for optimal model performance and excessive input length may degrade training efficiency and accuracy, we use sliding windows to cap input length. We experimented with window sizes of 150, 200, and 300 tokens, selecting the size with the best performance. Each sentence is represented by the concatenation of the one-hot encodings of its tokens, where each token is represented in a 503-dimensional space. Thus, with a window size of 300, the input dimension approximates 1e5. We evaluated MLPs with varying layer numbers and a hidden size of 1000 dimensions, as described in Section 4.3. The MLP output dimension matches the 50257 tokens of GPT2Tokenizer, with unused tokens assigned near-zero weights. For MLP we also use the AdamW optimizer with the learning rate 5e-5.

## G.3 LINEAR FUNCTION FOR PROBING

For the probing task, we use a single-layer linear neural network without activation, effectively functioning as a matrix. The input size corresponds to the Transformer's 720-dimensional embeddings, and the output size matches the 50257 tokens of GPT2Tokenizer.

## H  TESTING PERFORMANCE CURVES

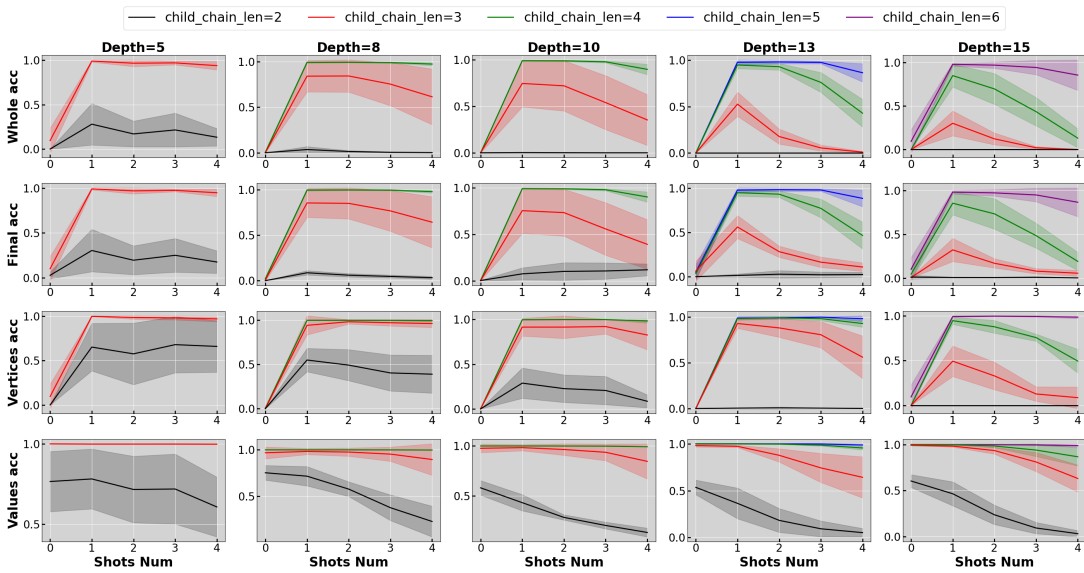

Figure 3: Testing performance of Transformers trained on FTCT with all combinations of causal depths and child chain lengths. The performance is evaluated by all four criteria in Appendix H.1.

## H.1  ALL TESTING PERFORMANCE CURVES

We summarize the criteria used to evaluate the compositional reasoning performance (including final value accuracy) as follows:

**Whole chain accuracy:** Measures if the model's generation contains all vertices and values along the reasoning chain in a correct order. For $(\widetilde{\text{inp}}^k, \widetilde{\text{lab}}^k)$ from $\mathcal{D}_{\text{test}}$, it measures whether $\text{dec}(\text{model}(\widetilde{\text{inp}}^k))$ contains all elements from $\text{dec}(\widetilde{\text{lab}}^k)$ in a correct order.

**Final value accuracy:** Measures if the model outputs the correct value of the last vertex.

**Testing vertices accuracy:** Measures if the model correctly outputs all vertices in $\text{dec}(\widetilde{\text{lab}}^k)$.

**Testing values accuracy:** Measures if the model outputs correct values of intermediate vertices, given correct preceding reasoning paths from $\mathcal{D}_{\text{test}}$.

We trained 3-layer, 3-head GPT-2-like Transformers on the FTCT training set with varying graph depths ($N$) and maximum child chain lengths ($M$). We test the performance of Transformers trained on FTCT tasks with $N = 5, 8, 10, 13, 15$ and $M = 2, 3, 4, 5, 6$. For each $(M, N)$ pair, we change the causal structures for 5 times, calculating the average and standard deviation of the testing performance of Transformers trained on them, recording the results in Figure 3.

From Figure 3 we observe that the curves of the final value accuracy mirrors the curves of whole chain accuracy, underscoring the necessity of proper reasoning paths for correct final answers

## H.2  TESTING PERFORMANCE WITH NOISY TOKENS

In the main text, we configured the training data to consist of child chains blurred by noisy tokens, while the testing data comprised the longest chains without such noise. This setup simulates real-

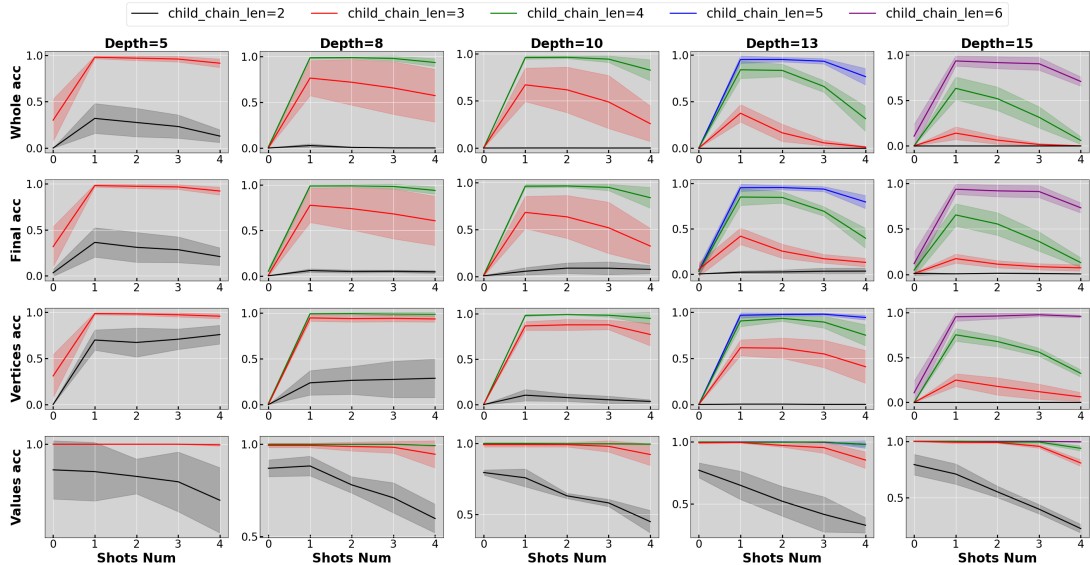

Figure 4: Testing performance of Transformers trained on FTCT where the testing data are blurred by noisy tokens. The performance is evaluated by all four criteria in Section H.2.

world scenarios where the training corpus is typically affected by noise, while users offer high-quality, noise-free prompts during testing. To demonstrate the robustness of our conclusion, we evaluate the performance of Transformers on testing data that is also blurred by noisy tokens. The key to constructing this noisy testing dataset is merging the longest chain in $\mathcal{T}(\mathcal{G})$ with randomly sampled noisy tokens from $\mathcal{V}_{\text{all}} - \mathcal{V}$, while ensuring that (1) the order of the vertices in the chain is preserved and (2) the last vertex of the chain remain as the last vertex in the merged sequence. For the causal structure depicted in Figure 1, such a merged sequence might look like

$$[F, 9, A, 100, K, 6, B, 101, H, 1, C, 103]$$

where [A, B, C] is the longest chain in $\mathcal{T}(\mathcal{G})$ and [F, K, H] are noise randomly sampled from $\mathcal{V}_{\text{all}} - \mathcal{V}$. After downstream processing, such sequnce is transformed to the input "C=?: ... F=9, ... A=100," and label "... K=6, ... B=101, ... H=1, ... C=103". To assess compositional reasoning performance on this dataset, we modify our evaluation criteria to focus only on the accuracy of non-noisy vertices in the set of knowledge points $\mathcal{V}$:

**Whole chain accuracy:** Measures if the model correctly predicts all vertices belonging to the set of knowledge points $\mathcal{V}$ and their values along the reasoning chain.

**Final value accuracy:** Measures if the model outputs the correct value of the last vertex in $\mathcal{V}$ within the sentence.

**Testing vertices accuracy:** Measures if the model correctly outputs all vertices in $\text{dec}(\widetilde{\texttt{lab}}^k)$ which belong to $\mathcal{V}$.

**Testing values accuracy:** Measures if the model outputs correct values of intermediate vertices that belongs to $\mathcal{V}$, given correct preceding reasoning paths from $\mathcal{D}_{\text{test}}$.

Figures 4 and 5 display the compositional reasoning capacity of Transformers trained on FTCT with testing data blurred by noisy tokens. The results align with those from tests without noise, indicating the robustness of our conclusions.

# I  LEAST VISITED TIMES OF ALL ADJACENT VERTICES AND THEIR VALUES

We define the set of all adjacent vertices and their values as

$$\mathcal{S}(\mathcal{G}) := \{(v_i, q_i, v_{i+1}, q_{i+1}) \mid v_i, v_{i+1} \in \mathcal{V}, (v_i, v_{i+1}) \in \mathcal{E}, q_i \in \text{VALS}(v_i), q_{i+1} = \text{op}(v_i, v_{i+1}) \circ q_i\}.$$

Only when each element in $\mathcal{S}(\mathcal{G})$ has been visited for enough times in the training data, can we ensure that the poor compositional reasoning ability of models trained on certain FTCT tasks is not

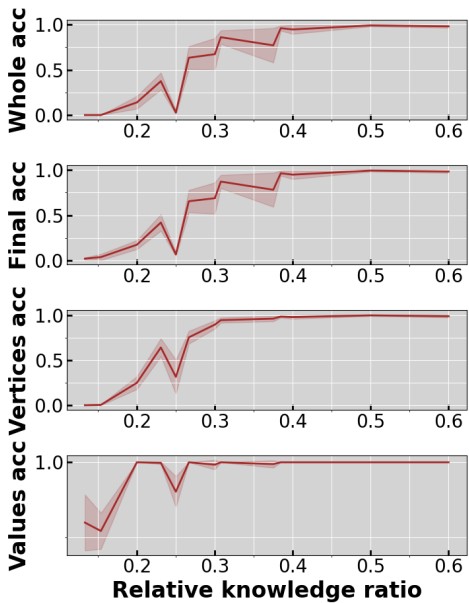

Figure 5: The relationship between the relative knowledge ratio and Transformers' compositional reasoning ability tested on the data blurred by noisy tokens.

|         | Depth 5 | Depth 8 | Depth 10 | Depth 13 | Depth 15 |
|---------|---------|---------|----------|----------|----------|
| Child 2 | 194     | 109     | 76       | 55       | 46       |
| Child 3 | 250     | 123     | 89       | 65       | 51       |
| Child 4 | —       | 140     | 91       | 67       | 57       |
| Child 5 | —       | —       | —        | 72       | 56       |
| Child 6 | —       | —       | —        | —        | 59       |

Table 4: Least visited times of training datasets generated on causal structures with various causal depths and child chain lengths.

due to the insufficient data. For any training dataset $\mathcal{D}$, we define $t_{\mathcal{D}}(v_i, q_i, v_{i+1}, q_{i+1})$ as the times the tuple $(v_i, q_i, v_{i+1}, q_{i+1})$ is visited by the data in $\mathcal{D}$. We define the least visited times of $\mathcal{D}$ as

$$T(\mathcal{D}) := \min\{t_{\mathcal{D}}(v_i, q_i, v_{i+1}, q_{i+1}) \mid (v_i, q_i, v_{i+1}, q_{i+1}) \in \mathcal{S}(\mathcal{G})\}.$$

The least visited times measure the degree of how well the $\mathcal{S}(\mathcal{G})$ is visited by $\mathcal{D}$. We demonstrate the least visited times of all our training data in Table 4. The results suggest that all necessary knowledge parts are covered sufficiently by our training dataset.

## J  PERFORMANCE OF LARGER TRANSFORMER MODELS

To assess the generalizability of our findings to more complex and sizable architectures, we trained GPT2-small (12 layers, 12 heads, 117M parameters) and GPT2-large (36 layers, 20 heads, 774M parameters) on FTCT tasks with varying causal depths and child chain lengths. We discovered that the diversity of training data used for smaller models was insufficient to leverage the larger models' generalized in-context learning ability. Consequently, we introduced auxiliary data to facilitate this capability. Specifically, the auxiliary data comprises few-shot examples with vertices and values randomly sampled from $\mathcal{V}_{\text{all}}$ and $\mathbb{Z}$, respectively. The testing performances of GPT2-small and GPT2-large are depicted in Figures 6 and 8, with the performance's relationship to relative knowledge ratio illustrated in Figures 7 and 9. We continue to observe that compositional reasoning ability

emerges with increased shot numbers and relative knowledge ratios. However, the performance of these larger models is less stable compared to smaller ones, which may be attributed to overfitting.

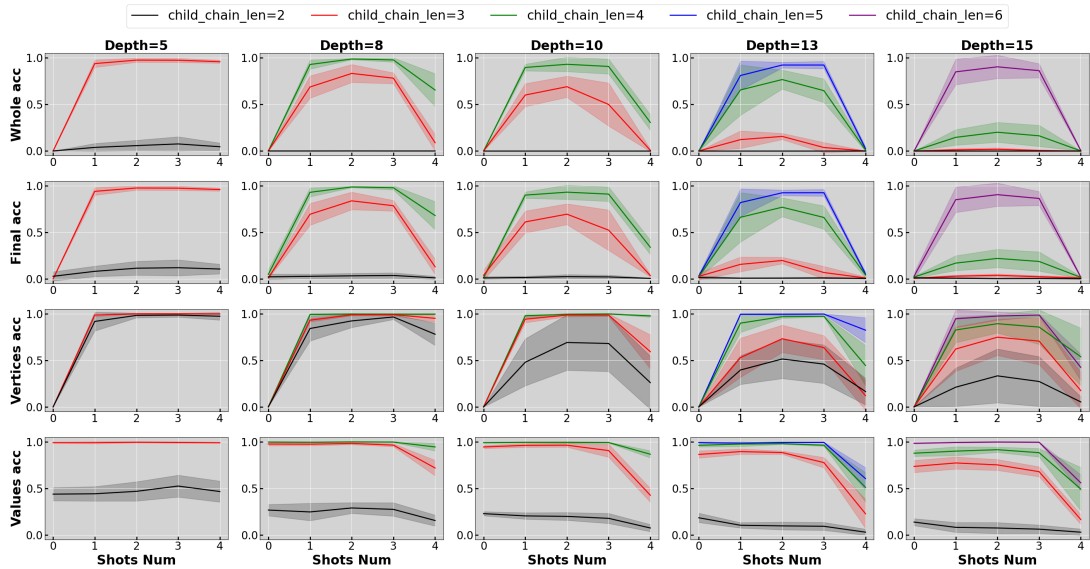

Figure 6: Zero and few-shot testing performance of GPT2 small trained on FTCT with various causal depths and child chain lengths.

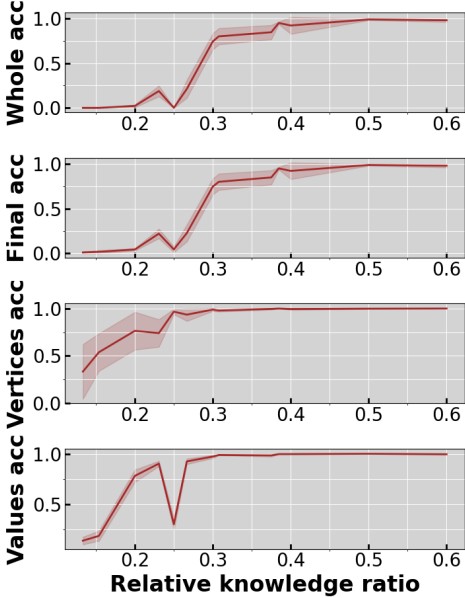

Figure 7: The relationship between the relative knowledge ratio and the compositional reasoning ability of GPT2 small.

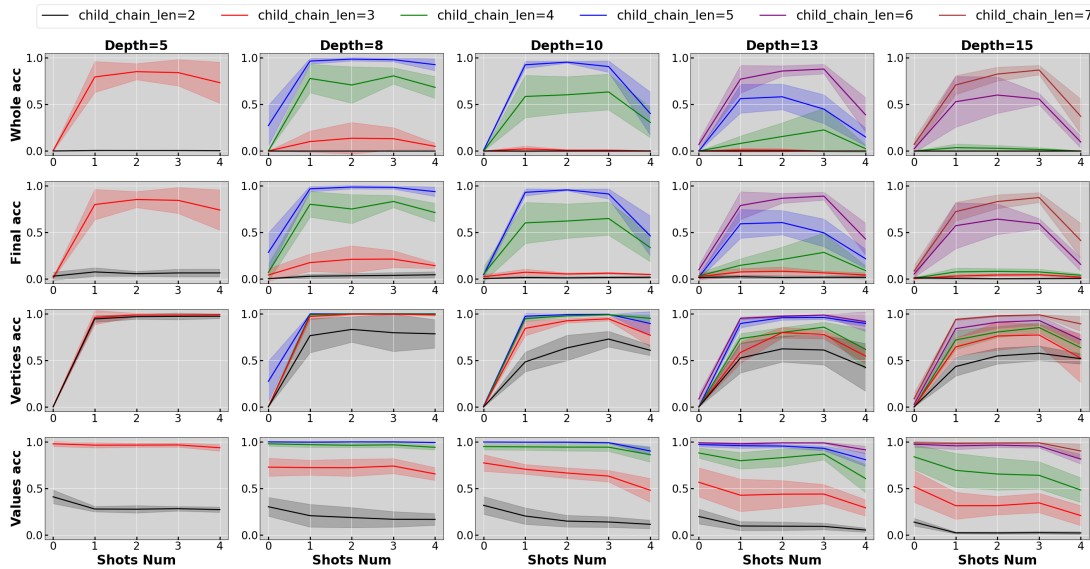

Figure 8: Zero and few-shot testing performance of GPT2 large trained on FTCT with various causal depths and child chain lengths.

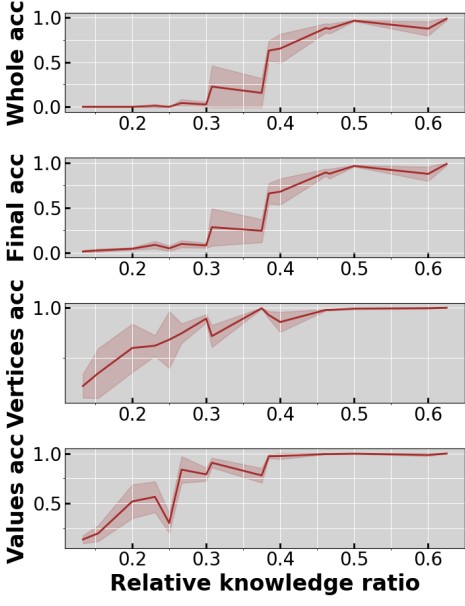

Figure 9: The relationship between the relative knowledge ratio and the compositional reasoning ability of GPT2 large.

## K  EXPLAINING THE DECREASE OF PERFORMANCE WHEN SHOTS NUMBER IS LARGER THAN ONE

In Figure 2 and Table 3, we observe a decrease in model performance when more than one few-shot prompt is used, contrary to the typical expectation that additional examples enhance in-context learning. While it is generally expected that more few-shot examples improve performance in in-context learning, performance can fluctuate or even decline as additional examples are introduced

(as noted in Figure 4b of (Garg et al., 2022) and Figures 7, 10 of (Xie et al., 2021)). Although systematic explanations are limited, it is not uncommon for performance to drop when the number of shots surpasses a particular threshold. (Agarwal et al., 2024) notably demonstrates that the optimal shot count for peak performance is often less than the maximum a model can manage, indicating that performance does not always increase monotonically with additional shots.

We propose that performance decreases when the number of shots exceeds one because additional CoT examples increase the dissimilarity between training and testing data, thereby degrading performance. To explain this, we begin by analyzing the difference between one-shot examples in testing and training data. In testing, one-shot examples typically consist of the longest chains of length $N$, whereas in training, they are child chains of length $M < N$, with the primary difference being the $N - M$ missing vertices. As the number of shots increases, each shot introduces more instances of these missing vertices, compounding the disparity between training and testing prompts. This growing difference complicates the model's ability to recognize patterns in the testing data, thus impairing performance.

In our setup, a single shot during testing already provides ample information about the vertex order needed for generating correct answers. For a $k$-shot testing example, the additional $k - 1$ shots do not add valuable information and only exacerbate the divergence between training and testing data. Consequently, we observe that testing performance peaks at 1-shot and diminishes thereafter, aligning with our expectations.

When the differences between training and testing data are limited, we observe the expected pattern of in-context learning, where performance improves with more shots and does not decline after reaching its peak. From all FTCT tasks with various depth and child chain length combinations shown in Figure 3, we select settings where performance decreases when shot numbers exceed one. These settings include "depth=8, child chain length=3," "depth=10, child chain length=3," "depth=13, child chain length=4," and "depth=15, child chain length=4." Instead of testing models' performance on the longest chains equal to their depth, we assess performance on causal chains with lengths varying from the child chain length to the depth. For instance, for models trained on an FTCT task with depth=8 and child chain length=3, we measure performance on chains with lengths 3, 4, 5, 6, and 8. The results in Figure 10 show that for tasks where test lengths are close to child chain lengths, few-shot performance does not decrease. Notably, zero-shot performance increases as test lengths near the child chain length. As the gap between child chain and test lengths widens, the performance decrease after one shot becomes evident.

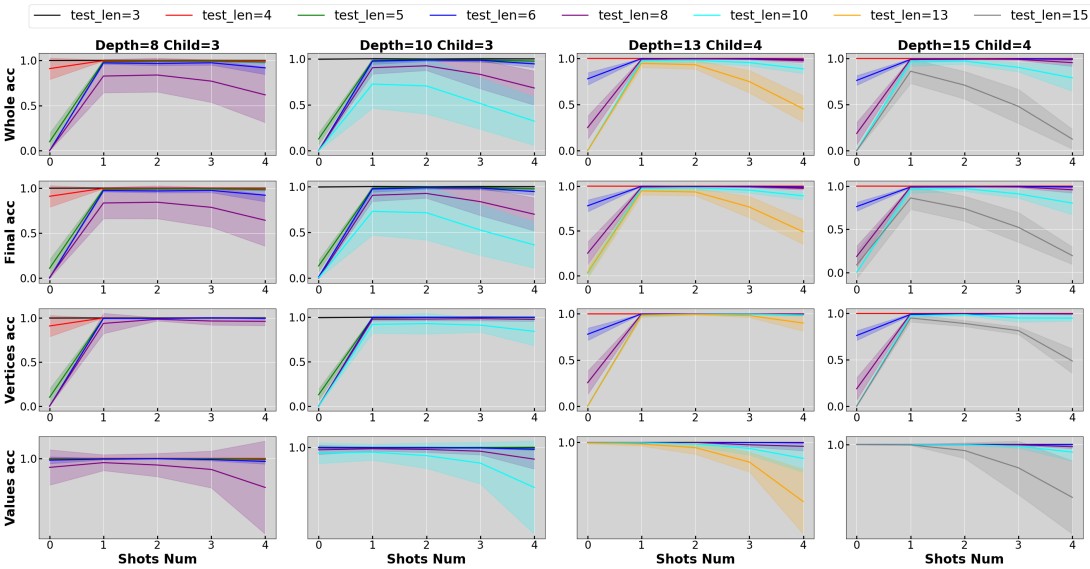

Figure 10: Performance of Transformers in reasoning causal chains with varying lengths.

## L    EMPIRICAL EVIDENCE OF THE TRANSFORMER STRUCTURE

Figure 11 and 12 demonstrate the evidence of the induction head by attention heatmap.

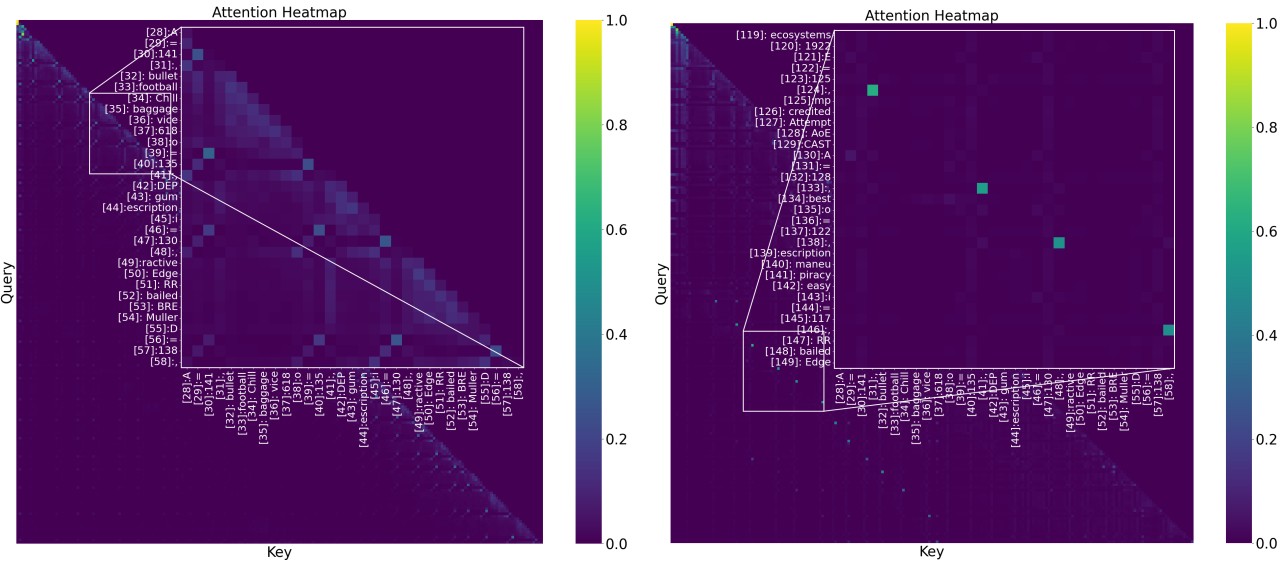

Figure 11: *Left:* Average attention heatmap of induction heads at layer 1 (test). *Right:* Average attention heatmap of induction heads at layer 2 (test).

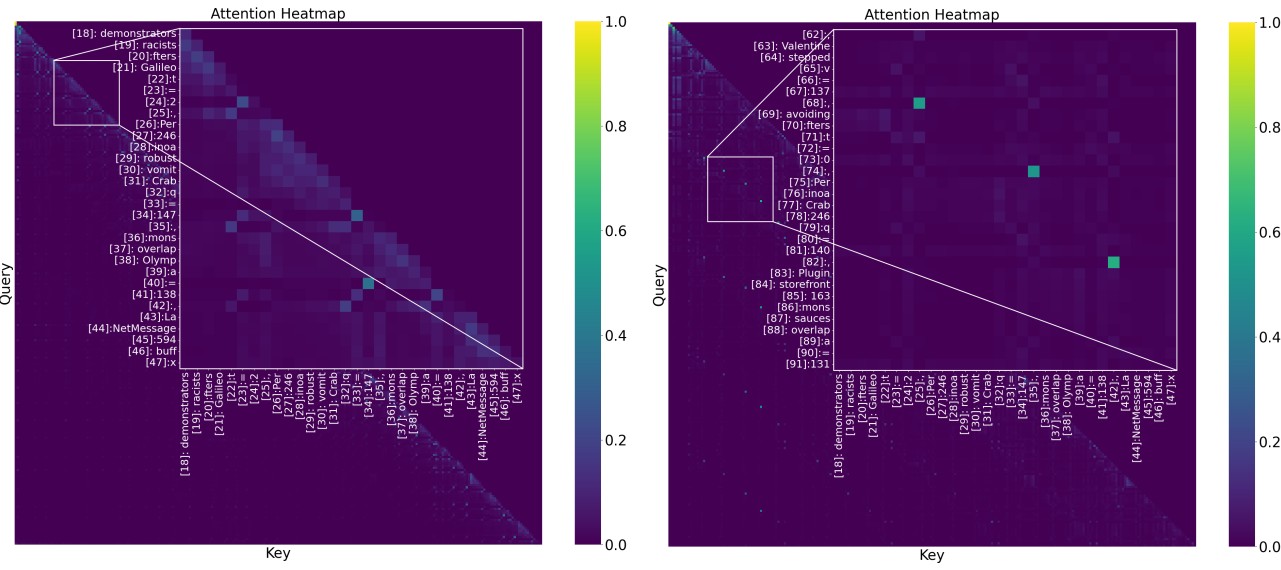

Figure 12: *Left:* Average attention heatmap of induction heads at layer 1 (train). *Right:* Average attention heatmap of induction heads at layer 2 (train).

## M    TESTING PERFORMANCE WITH INCOMPLETE INTERMEDIATE REASONING STEPS

In this section, we evaluate the capability of Transformers trained on FTCT to perform reasoning with missing intermediate steps. We conduct this test by removing intermediate vertices from the testing data. For the testing sequence [A, B, C] illustrated in Figure 1, we remove vertex B, creating

a test input formatted as "C=?: ... A=100, ... C=103 \n C=?: ... A=102," with the test label being "... C=105". For causal structures with greater depth, we define the retaining ratio $\rho$ as the ratio of the count of remaining vertices to the causal structure's depth. This ratio reflects the degree of incompleteness in the reasoning path. For example, if the longest chain in the causal structure is [A, B, C, F, G, I] and $\rho$ is set to 0.5, then the number of remaining vertices is 6 * 0.5 = 3. Since we only remove intermediate vertices, potential final sequences include [A, F, I] or [A, C, I], among others. Our evaluation criteria remain the same as the original (Section 4.1), but now both testing inputs and labels comprise incomplete reasoning paths. Given that Transformers trained on FTCT cannot generate incomplete reasoning paths autonomously in a zero-shot setting, we assess performance with 1 to 4 shots, each containing examples of incomplete reasoning paths.

Figure 13 illustrates the testing performance with retaining ratios of 0.3, 0.5, and 0.8. The results indicate that Transformers trained on FTCT exhibit limited proficiency in reasoning with incomplete intermediate steps. This limitation is likely due to training dataset bias, where adjacent vertices consistently appear consecutively in training sequences. Nonetheless, this performance shortcoming does not compromise our main findings, which demonstrate the Transformers' compositional reasoning capabilities, because the testing data invariably contain longer causal chains than the training data, regardless of whether the demonstrated reasoning path is complete or not.

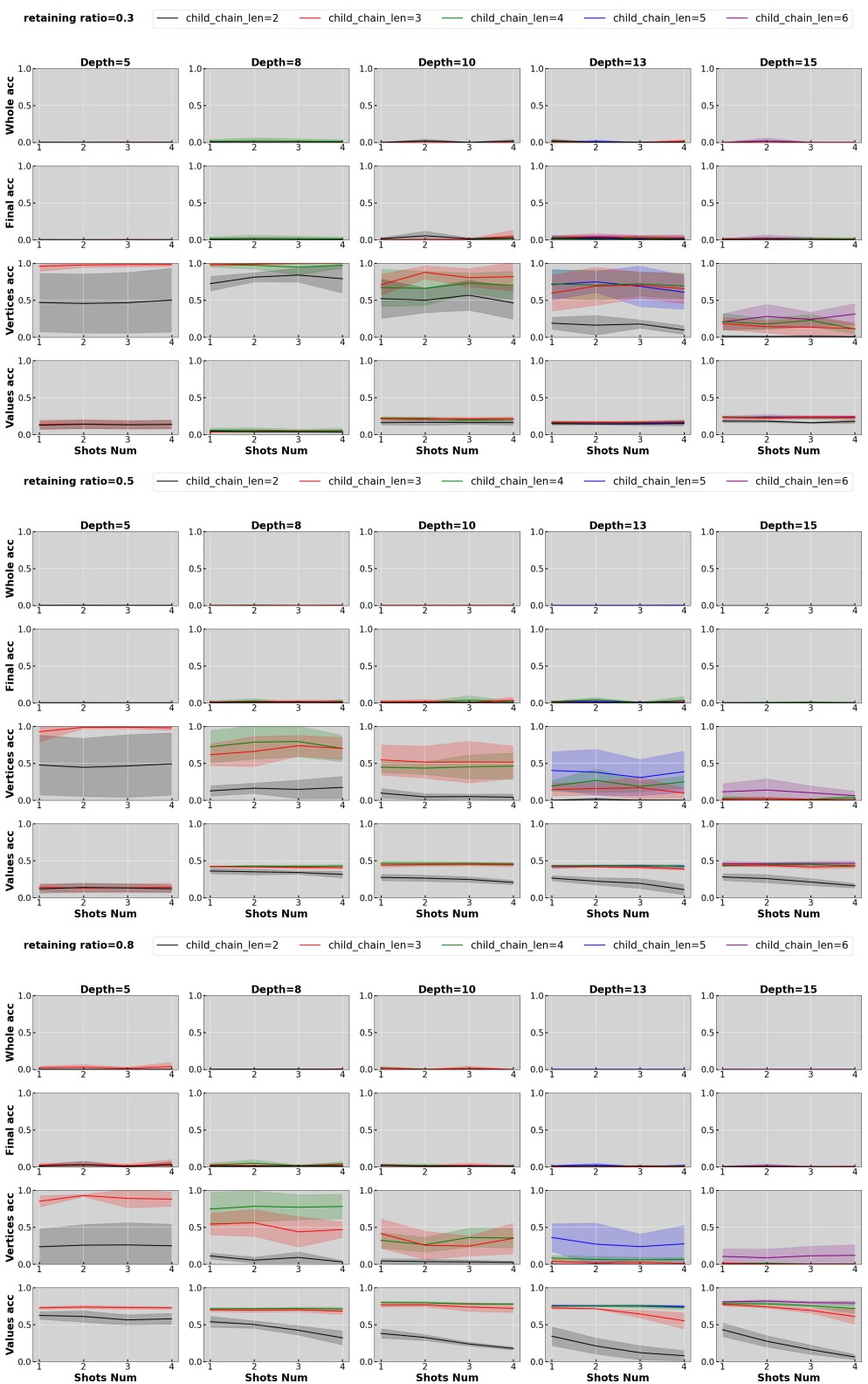

Figure 13: Testing performance of Transformers trained on FTCT with incomplete intermediate reasoning steps (retaining ratio=0.3, 0.5 and 0.8 from up to down).

