# OpenReview forum: "Are Transformers Able to Reason by Connecting Separated Knowledge in Training Data?"
_ICLR.cc/2025/Conference — ICLR 2025 Poster_

### Official Review · Reviewer_hQE3 · 2024-11-01

**Soundness:** 3
**Presentation:** 2
**Contribution:** 3
**Rating:** 6
**Confidence:** 3

**Summary:**

**Aims:** This paper seeks to understand at a mechanistic level how Transformers are able to perform compositional reasoning in a few-shot chain-of-thought setting.

**Methods:** A synthetic dataset is generated consisting of chains of nodes and edges derived from causal graphs. At training time, spurious nodes are inserted randomly into the chains; at testing time, few-shot prompts consisting of intact chains (no spurious nodes) are provided to the model. Models are tested on their ability to reconstruct full causal chains from fragmented chains learned in training, with evaluation based on accuracy in predicting both the correct vertex order and values in the chain.

**Results:**

- Zero-shot versus few-shot prompting is compared, with findings showing that few-shot CoT prompting significantly enhances performance in compositional reasoning tasks, particularly in forming the correct vertex order.
- A space of small, GPT-2-style models ranging from 42M-54M parameters are trained on the FTCT dataset. Results show that multi-layer, multi-head Transformers (minimum 2 layers and 2 heads) perform notably better, while single-layer/single-head models and MLPs perform poorly.
- The impact of training data’s relative knowledge ratio (ratio of child chain length to complete chain length) is studied, with a critical threshold (ratio ≥ 0.3) identified where compositional reasoning reliably emerges.
- Mechanisms underlying the model's success, such as induction heads for in-context learning and attention patterns facilitating parent-child relationships, are analyzed through linear probing, revealing specific mechanisms by which the model achieves compositional reasoning.

**Strengths:**

**Motivation:** The key research questions of the paper are clearly delineated in the introduction:

1) When are Transformers able to perform compositional reasoning by connecting fragmented knowledge in training data?
2) How do different training factors impact the emergence of this ability?
3) What internal mechanisms enable Transformers to develop this ability?

These questions are broadly relevant to current research and the paper is structured in a way that consistently centers these 3 questions throughout.

**Mechanistic interpretability analysis:** I especially enjoyed the approach to question (3). Broadly speaking, the authors approach this question by first demonstrating that there exists a program that solves the task (Sec. 5.1) and that this program can be approximated by a 2-layer Transformer (Sec. 5.2). Then, through linear probing experiments (Sec. 6), they give an empirical argument that the Transformers trained on FTCT have learn to implement this program. I am not an expert on probing so I can’t speak to the soundness of the methods, but I found the combination of Sec. 5-6 to be an elegant argument from a mechanistic interpretability standpoint.

**Weaknesses:**

**Novelty:** The paper claims to “introduce a learning task” (FTCT) based on causal graphs, and yet the design of this task is nearly identical to the setup in Prystawski et al. (2024). Given that the main distinction between FTCT and the prior work is the introduction of spurious nodes (line 105-106), I would expect to see this prior work—which actually *did* introduce a novel learning task—given more prominent attribution.

- (Currently this work is referenced in Line 103—”The empirical findings of our work align with the observation in (Prystawski et al., 2024)…” The wording of this reference obfuscates the underlying causal structure that this prior work likely played in informing the current paper.)

**Generality:** The key findings of this paper are framed in very broad terms:

> “The emergence of compositional reasoning is highly influenced by the data’s relative knowledge ratio and model complexity. Specifically, a relative knowledge ratio of at least 0.3 and a Transformer architecture with at least two layers and two heads are critical for achieving this ability.” (Lines 520-523)
>

However, these conclusions are all drawn relative to one synthetic dataset with a highly-specific structure; it is unclear to what extent the empirical conclusions (e.g., compositional reasoning in transformers requires a relative knowledge ratio ≥ 0.3) generalize beyond the FTCT task. To make a convincing argument that these results have meaning beyond this one benchmark, this analysis ought to be replicated on more naturalistic reasoning benchmarks where few-shot CoT prompting is commonly used.

**Clarity:** The description of the FTCT dataset/task design (Sec. 3) fails to convey a clear description of the experiment setup and requires too much work of the reader. All aspects of prompt construction are described in excruciating formal detail, making it hard to separate design choices that are key to the experiment from implementation details. Overall, the formalism in this section is a barrier to understanding what’s going on at a more basic level.

- Fig. 1 lacks basic signposting needed to convey what is going on.
    - First off, there is no caption. This is a major omission as the figure is definitely not self-explanatory.
    - The blue highlights draw the reader’s attention to spurious features of the task (noise nodes) instead of the actual purpose of the task (predicting values of causal nodes).
- Other comprehension/clarity issues in Sec. 3:
    - “We assume op(e) represents operations like (+a) or (−b)” Does this mean addition/subtraction are the *only* possible operations?
    - I don’t understand how the merge operation works from the description.
    - Some unconventional choices of notation, such as using $f$ as an index over few-shot examples.
    - What is “downside processing” - do you mean “downstream”?

**Questions:**

1. Table 1 shows that all Transformer models attain 1.0 Values accuracy, even for small models that get very low Vertices accuracy. Can you account for this discrepancy?
2. An unintuitive pattern in the results (e.g., Fig. 2 and Table 3) is that accuracy *decreases* with the number of few-shot prompts $>1$. This results stands in contrast to a standard ICL setting, where inclusion of more examples typically improves performance. It is stated that this is “possibly due to increased dissimilarity between training and testing data with more CoT examples” (Line 277-278). Why does including more CoT examples causes the test data to be OOD? If this is the case, this seems like an important confound affecting this experiment setup that may not be adequately accounted for.
3. It is interesting to contrast the results from Sec. 5-6 with Zhang et al., 2022 (“On the Paradox of Learning to Reason from Data”), who apply a similar methodology but find that gradient descent fails to discover the correct $\theta^*$ for a logical induction task with very similar structure. Is there a reason why here the training succeeds at uncovering the underlying program, whereas in previous work it does not? More generally, it would be nice to see reference to this paper in the discussion.

---

> ### Author Response · Authors · 2024-11-21
> **Response for Reviewer hQE3 (PART 1)**
>
> Thank you for the thorough reviewing and constructive advice! We address your concerns as follows.
>
> **W1:** More prominent attribution should be given to the prior work Prystawski et al.  [1].
>
> **A for W1:** To clarify how Prystawski et al.'s work has influenced our research and to outline the fundamental differences between our studies, we have revised the original discussion of their work in the related works section [Lines 107-111].
>
> - We clarify that our FTCT data construction is informed by their task's framework: “Our FTCT structure draws inspiration from their Bayesian networks” [Lines 107-108].
> - We specify the structural differences, noting that we introduce additional elements and complicate the dependency: “additionally inserting contextual noise and complicating value relationships” [Line 108].
> - Furthermore, we underscore the distinct differences in our research goals and contributions: “While they focus on locality structure's impact on CoT efficacy, we investigate how various training factors influence the emergence of compositional reasoning and conduct an in-depth analysis of the mechanisms within Transformer structures that elicit this capability.”[Lines 109-111].
>
> **W2:** The generality of our conclusions made by training highly specific model structures on synthetic data is questionable. Analysis on more naturalistic reasoning benchmarks is needed.
>
> **A for W2:** We would like to state that our main focus and contribution are not about investigating the performance of pre-trained models on natural language tasks. In fact, many empirical works have already demonstrated the symptom that well-pretrained language models do compositional reasoning on complex realistic tasks like question answering, mathematical reasoning and interdisciplinary generation ([2], [3], [4] [5]). They are shown to generate comprehensive content with elements unlikely to co-occur in training data. **We aim to scientifically validate such ability within a clear environment and explore its underlying mechanism.** To this end, it is necessary to conduct controlled and tunable experiments where reasoning paths in the test data neither appear in the training data nor are they encoded in the models' pre-existing weights. Compared with naturalistic tasks with intractable natural language corpus, we believe that a controllable synthetic task is more appropriate for our research purpose. Doing research on synthetic data is also an approach being widely used by previous interpretability studies ([6], [7], [8]).
>
> To clarify the above argument, we attach a new version of the paper that includes following modifications:
>
> - Refine the research purpose in the introduction: “This paper validates the potential of Transformers in doing compositional reasoning on synthetic dataset and investigates the inner mechanism eliciting such ability. ” [Lines 45-46].
> - Pointing out that utilizing models trained on vast natural language is not proper for our research purpose [Lines 42-44].
> - Reframe the statement about relative knowledge ratio within the scope of FTCT: “compositional reasoning emerges as the similarity between training and testing data increases. In FTCT, this is measured by the relative knowledge ratio” [Lines 69-70].
> - Adjust the conclusion: “Our research validates the potential of Transformers in doing compositional reasoning on synthetic data and investigates the inner mechanism eliciting such ability. ”[Lines 527-528].
>
> Admittedly, conducting comprehensive scientific interpretability studies on large language models trained on complex natural corpora is meaningful. However, such endeavors remain challenging amidst current research advancements. We hope our research makes a contribution to this ultimate goal.
>
> **W3:** Too many technical details in the Section 3 makes it hard to get the core idea. Problems of Figure 1. Other comprehension/clarity issues in Section 3.
>
> **A for W3:**
>
> - Issues with Figure 1:
>     - A self-explanatory caption has been added in the newly submitted revised version [Lines 176-183].
>     - To prevent the distraction caused by blue highlight, in the revised version we distinguish knowledge points and noisy nodes with gray and blue only at the beginning of the “Fragmented at Training” stage and remove all subsequent highlights to avoid confusion [Lines 163-166].

---

> ### Author Response · Authors · 2024-11-21
> **Response for Reviewer hQE3 (PART 2)**
>
> - Other comprehension/clarity issues:
>     - It is correct that addition and subtraction are the only possible operations. We clarify this in the revised version [Line 188].
>     - We have illustrated the merge process with a concrete example for clarity [Lines 200-202]. Specifically, for the causal structure in Figure 1 [Lines 162-170], suppose the sampled child chain is [A, B] and the sampled noisy vertices are [Z, H]. We then merge [A, B] and [Z, H] in to a single sequence while preserving the relative order of [A, B]. Potential outcomes include [A, Z, H, B], [A, Z, B, H], and [H, A, B, Z], of which we select [A, Z, B, H]. If the parent vertex A is assigned a value of 100 and the edge [A, B] operation is “+1”, B’s value becomes 101. We randomly sample Z and H values as 3 and 1, resulting in the sequence [A, 100, Z, 3, B, 101, H, 1]. Please refer to Appendix B [Lines 784-798] for the formal merging process description.
>     - We have changed the index of few-shot examples from $f$ to $k$, a more standard notation [Lines 208-214, 222-223, 230-243, 255-263 ...].
>     - The terminology "downside processing" actually refers to "downstream processing", which denotes the final processing step aimed at adapting sequences into natural language-like sentences. We have updated this terminology [Lines 216-217].
> - We also deleted the redundant construction details and add concrete examples in “Downstream processing” part [Lines 220-221].
>
> **Q1:** Why do all Transformer models attain 1.0 values accuracy, even for small Transformers with low vertices accuracy?
>
> **A for Q1:** We first clarify the definitions of vertices accuracy and values accuracy [Lines 262-267]:
>
> - Vertices accuracy measures whether the model correctly outputs all vertices included in the label. For instance, with an input like “… C=?:… A=100, … B=101, … C=103 \n C=?: … A=102”, the model is deemed to have accurate vertices if it outputs sentences like “….B=b, …C=c”, regardless of the specific values of b and c.
> - Values accuracy assesses whether the model correctly outputs the values of intermediate vertices, given correct reasoning paths. For values accuracy, if the input is “… A=100, … B=”, the model is considered accurate if it outputs “101” as the next token.
>
> Because the values accuracy is tested by prompting models with the reasoning paths that already have accurate vertices, model with low vertices accuracy may still pertain high values accuracy.
>
> Regarding why a one-layer Transformer achieves such performance:
>
> - As mentioned in Section 4.3 [Lines 363-365], the limited vertices accuracy is attributed to **the absence of induction heads** in a one-layer Transformer. During testing, models need to use generalized in-context learning ability to retrieve and replicate vertices accurately. As discussed in Section 6.1 [Lines 480-503], induction heads are two heads of the Transformer in different layers that facilitate in-context learning. They are unlikely to exist in one-layer Transformers, thus resulting in poor vertices accuracy.
> - The High Values Accuracy may be due to the ability of a single-layer attention mechanism to handle parent vertex retrieval and mapping memorization. The precise mechanism of it remains unclear to us and we intend to make further exploration in future research.

---

> ### Author Response · Authors · 2024-11-21
> **Response for Reviewer hQE3 (PART 3)**
>
> **Q2:** Why does accuracy decrease with more than one few-shot prompt?
>
> **A for Q2:** While it is typically expected that performance improves with more few-shot examples, it is not rare for performance to fluctuate or even decrease as additional examples are provided (see Figure 4b in [9] and Figures 7, 10 in [10]). Notably, Figure 2 in Agarwal et al. [12] demonstrates that the optimal number of shots for peak performance is often lower than the maximum number they can handle, indicating that performance does not always increase monotonically with the number of shots.
>
> We argue that performance decreases when shots number > 1  because more CoT examples increase the dissimilarity between training and testing data. To explain this,  we start by analyzing the difference between one-shot examples from the testing and training data. During testing, one-shot examples typically consist of the longest chains of length $N$. In contrast, one-shot examples during training comprise child chains of length $M < N$. As the number of shots increases, each shot introduces extra instances of these $N-M$ missing vertices, exacerbating the disparity between training and testing prompts. This growing difference makes it harder for the  model to recognize testing data patterns, thus reducing performance. In our setup, a single shot during testing already provides ample information about the vertex order needed for generating correct answers. For a $k$-shot testing example, the additional $k-1$ shots do not add valuable information and only exacerbate the divergence between training and testing data. Consequently, we observe that testing performance peaks at 1-shot and diminishes thereafter.
>
> We test the performance of Transformers in reasoning causal chains with varying test lengths in Appendix L [Lines 1613-1673]. The results (Figure 10 [Lines 1653-1673]) show that for tasks where test lengths are close to child chain lengths models are trained on, few-shot performance remains stable without decrease. As the gap between child chain and test lengths widens, the few-shot performance decrease becomes evident.  Thus, we conclude that **when differences between training and testing data are limited, the expected pattern of in-context learning appears, where performance improves with more shots and does not decline after reaching its peak. As the gap between testing causal chains and training child chains widens, the performance decrease after one shot becomes evident, indicating the influence brought by OOD tasks.**
>
> This non-monotonic pattern does not affect our conclusions and contributions, as the performance at the optimal number of shots already sufficiently reflects the compositional reasoning ability.
>
> **Q3:** In contrast to our successful results, Zhang et al. [11] find that Transformers fail to uncover the underlying program to generalize to different testing data. What explains our different outcomes?
>
> **A for Q3:** Possible reasons of this difference are as follows.
>
> - CoT prompting: Zhang et al. utilize BERT models to directly generate answers for complex reasoning tasks. In contrast, we employ few-shot CoT prompting to facilitate step-by-step reasoning in GPT2-like Transformers, which has been shown to provably enhance model’s expressivity and generalization ability ([13], [14]). Without the help of few-shot CoT, Transformers also struggle in FTCT (poor zero-shot performance in Figure 2 (left) [Lines 289-308] )
> - Statistical Features: Our FTCT dataset contains fewer "statistical features" compared to those in Zhang et al.'s datasets. As per Zhang et al., a statistical feature is a specific statistic of an example that strongly correlates with its label. The presence of numerous statistical features can lead models to learn these correlations rather than the true underlying program. Our dataset’s testing performance is segregated into vertices accuracy and values accuracy. Each vertex token relates solely to the same token in preceding few-shot examples, without obvious statistical features that correlate highly with it. It is the same for the values, which is only related with its corresponding vertices and its parent.
>
> In the revised version, we discuss this work in the related works section [Lines 139-141].
>
> We would like to emphasize that Transformers can not solve every FTCT task successfully. For those tasks with low relative knowledge ratio, Transformers also fail to generalize (Figure 2 (right) [Lines 289-308]). We observe a phase transition of the generalization ability as the relative knowledge ratio increases, indicating the key role played by data structure in eliciting models’ generalization ability.

---

> > ### Author Response · Authors · 2024-11-21
> > **Response for Reviewer hQE3 (PART 4)**
> >
> > [1] Prystawski, Ben, Michael Li, and Noah Goodman. "Why think step by step? Reasoning emerges from the locality of experience." *Advances in Neural Information Processing Systems* 36 (2024).
> >
> > [2] Press, Ofir, et al. "Measuring and narrowing the compositionality gap in language models." *arXiv preprint arXiv:2210.03350* (2022).
> >
> > [3] Zhou, Denny, et al. "Least-to-most prompting enables complex reasoning in large language models." *arXiv preprint arXiv:2205.10625* (2022).
> >
> > [4] Khot, Tushar, et al. "Decomposed prompting: A modular approach for solving complex tasks." *arXiv preprint arXiv:2210.02406* (2022).
> >
> > [5] Bubeck, Sébastien, et al. "Sparks of artificial general intelligence: Early experiments with gpt-4." *arXiv preprint arXiv:2303.12712* (2023).
> >
> > [6] Chan, Stephanie, et al. "Data distributional properties drive emergent in-context learning in transformers." *Advances in Neural Information Processing Systems* 35 (2022): 18878-18891.
> >
> > [7] Allen-Zhu, Zeyuan, and Yuanzhi Li. "Physics of language models: Part 1, context-free grammar." *arXiv preprint arXiv:2305.13673* (2023).
> >
> > [8] Bietti, Alberto, et al. "Birth of a transformer: A memory viewpoint." *Advances in Neural Information Processing Systems* 36 (2024).
> >
> > [9] Garg, Shivam, et al. "What can transformers learn in-context? a case study of simple function classes." *Advances in Neural Information Processing Systems* 35 (2022): 30583-30598.
> >
> > [10] Xie, Sang Michael, et al. "An explanation of in-context learning as implicit bayesian inference." *arXiv preprint arXiv:2111.02080* (2021).
> >
> > [11] Zhang, Honghua, et al. "On the paradox of learning to reason from data." *arXiv preprint arXiv:2205.11502* (2022).
> >
> > [12] Agarwal, Rishabh, et al. "Many-shot in-context learning." *arXiv preprint arXiv:2404.11018* (2024).
> >
> > [13] Feng, Guhao, et al. "Towards revealing the mystery behind chain of thought: a theoretical perspective." *Advances in Neural Information Processing Systems* 36 (2024).
> >
> > [14] Li, Zhiyuan, et al. "Chain of thought empowers transformers to solve inherently serial problems." *arXiv preprint arXiv:2402.12875* (2024).

---

> ### Comment · Reviewer_hQE3 · 2024-11-22
>
> I thank the authors for their detailed response to the issues raised by myself and the other reviewers. I also appreciated the effort that the authors took to taxonomize and label their responses, which made it easy to follow.
>
> I believe that the authors' responses meaningfully addressed all 3 weaknesses I raised (W1-W3) as well as all questions (Q1-Q3). I especially appreciated the tailored responses to the questions -- these were very helpful and clarifying.
>
> At this point I think the decision around acceptance of this work mainly hinges on generality (W2). I tend to agree with Reviewer EQsG that the burden of proof for this work is higher than for analogous work highlighting shortcomings:
> > Generally, using a simple and singular synthetic dataset is suitable for highlighting the shortcomings of the Transformer architecture. However, since the paper concludes that Transformers possess this capability, the experiments on this task alone are not sufficient to support such a conclusion.
>
> In response, the authors argue that performing more comprehensive experiments would be "valuable" but "challenging amidst current research advancements." I sympathize with the view that asking for more systematic benchmarking of LLMs is one of the easiest things to request from a reviewer standpoint, and one of the most burdensome things to implement from an author standpoint. Moreover, I appreciate that the authors (in response to the reviews) added new benchmarks with GPT-2 small and large. This is sufficient for me.
>
> In response to the authors rebuttals, I have updated the following scores in my review:
> - **Presentation**: 1 → 2
> - **Overall Rating**: 5 → 6

---

> ### Author Response · Authors · 2024-11-23
> **Thank you for the updated review**
>
> Thank you for your thoughtful and insightful review of our rebuttals, and for your kind adjustments to our scores. These mean a lot to us!
>
> We appreciate the concerns regarding the sufficiency of synthetic data in demonstrating the positive results of Transformers’ capability. It’s noteworthy that while some analogous studies ([1], [2], [3]) focus on demonstrating Transformers' shortcomings with synthetic data, others ([4], [5], [6]) effectively utilize simple yet targeted synthetic data to explore the potential of Transformers and analyze their mechanisms. For instance, Zhou et al. ([4]) investigate factors influencing Transformers' length generalization through experiments on the addition of two integers. The controlled and structured nature of synthetic data indeed enhances the precision and scientific rigor of such investigations, aligning well with our research focus.
>
> To further elucidate the generality of our work, we kindly wish to clarify that empirical works ([7], [8], [9], [10]) have shown the symptom indicating that real LLMs possess the compositional reasoning ability. An example is GPT-4's ability to "write a supporting letter for Electron as a US presidential candidate” ([10]), illustrating a strange combination of fields unlikely to co-occur during training. However, scientific validation and a detailed mechanistic understanding of this ability have been limited by the intractability of natural data. We believe that our experiments and analysis using controlled synthetic data contribute to this understanding. (This research approach parallels methods used in physics. For instance, to investigate the phenomenon of ball lightning observed in nature, physicists have replicated spherical luminous balls through artificial laboratory experiments ([11], [12]), thereby confirming its potential formation and providing explanations through theoretical models.)
>
> Thank you once more for engaging in this meaningful discussion and offering valuable opinions!
>
>
> [1] Sanford, Clayton, Daniel J. Hsu, and Matus Telgarsky. "Representational strengths and limitations of transformers." *Advances in Neural Information Processing Systems* 36 (2024).
>
> [2] Abbe, Emmanuel, et al. "Generalization on the unseen, logic reasoning and degree curriculum." *Journal of Machine Learning Research* 25.331 (2024): 1-58.
>
> [3] Zhang, Honghua, et al. "On the paradox of learning to reason from data." *arXiv preprint arXiv:2205.11502* (2022).
>
> [4] Zhou, Yongchao, et al. "Transformers can achieve length generalization but not robustly." *arXiv preprint arXiv:2402.09371* (2024).
>
> [5] Bietti, Alberto, et al. "Birth of a transformer: A memory viewpoint." *Advances in Neural Information Processing Systems* 36 (2024).
>
> [6] Wang, Zixuan, et al. "Transformers Provably Learn Sparse Token Selection While Fully-Connected Nets Cannot." *arXiv preprint arXiv:2406.06893* (2024).
>
> [7] Press, Ofir, et al. "Measuring and narrowing the compositionality gap in language models." *arXiv preprint arXiv:2210.03350* (2022).
>
> [8] Zhou, Denny, et al. "Least-to-most prompting enables complex reasoning in large language models." *arXiv preprint arXiv:2205.10625* (2022).
>
> [9] Khot, Tushar, et al. "Decomposed prompting: A modular approach for solving complex tasks." *arXiv preprint arXiv:2210.02406* (2022).
>
> [10] Bubeck, Sébastien, et al. "Sparks of artificial general intelligence: Early experiments with gpt-4." *arXiv preprint arXiv:2303.12712* (2023).
>
> [11] Paiva, Gerson Silva, et al. "Production of ball-lightning-like luminous balls by electrical discharges in silicon." *Physical review letters* 98.4 (2007): 048501.
>
> [12] Wu, H-C. "Relativistic-microwave theory of ball lightning." *Scientific reports* 6.1 (2016): 28263.

---

### Official Review · Reviewer_oHAw · 2024-11-02

**Soundness:** 2
**Presentation:** 2
**Contribution:** 3
**Rating:** 6
**Confidence:** 3

**Summary:**

The paper investigates how well transformers are able to perform compositional reasoning tasks. To that end, the paper introduces a new dataset methodology, namely the Fragmented Training, Chained at Testing (FTCT) that simulates how models would be presented with training data in practice (with incomplete fragments of reasoning paths with noise + context) and how well the model is able to piece together the full reasoning chain in test-time. Using this methodology, the paper runs insightful experiments that ablate different lengths of partial reasoning chains during training, different transformers and neural architectures, and number of few shot CoT examples. Through these experiments, the authors find that few shot CoT plays an important role for compositional reasoning, the impact of increasing relative knowledge ratio, and the increasing expressibility of adding layers and heads in the transformers architecture. Lastly, the paper presented some empirical evidence that you need a certain complexity of the transformers architecture to simulate the optimal program for the FTCT task in training and testing.

**Strengths:**

- The paper presented a very intriguing and creative approach to testing the ability for models to learn compositional reasoning ability
- There are some really interesting results, specifically the exact complexity (and the increased expressability) needed for the transformer architecture to optimally solve the FTCT task
- The insights regarding the few shot CoT results are of significance and spark further research in this area
- The empirical findings of how the transformers performs this task is enlightening and should spark some interest for further research in this area

**Weaknesses:**

- The clarity of this paper is lacking, especially in the notation and writing. For instance, in Figure 1, there is a seeming typo in some of the values that contradicts the setup of the dataset. Separately, some concrete examples of the data (including noise + context tokens) of the FTCT dataset would really improve the readers understanding (it took me multiple re-read to get the gist of the methodology)
- The paper's definition of compositional reasoning should be explicitly written out in the paper. The only real definition of this is in the abstract where it is stated that "For example, if someone learns ( B = f(A) ) from one source and ( C = g(B) ) from another, they can deduce ( C = g(f(A)) ) effortlessly, even without encountering ( AC ) or ( ABC ) together, showcasing their compositional generalization ability."
- With this FTCT methodology, it seems clear that the model is learning some ability to connect sequential reasoning chains together (e.g. during training the model might just as AB and BC and correctly chain ABC), but the approach does not test if the model can correctly reason about AC in test-time, which is an aspect of compositional reasoning (as mentioned in the abstract)

**Questions:**

- Although it is clear that the model is learning some ability to connect reasoning chains together (e.g. during training the model might just as AB and BC and correctly chain ABC), will the model be able to correctly chain together the values of AC? This could make for an interesting experiment where we could have some skip links in the test data and check for values accuracy
- Checking my understanding, is there a typo in Figure 1, where B=106 and B=103, should it be C=108 and C=105, respectively?
- Are there more than one set of the causal chains? The set equation in line 155 seems to suggest there is only one sequence of length n.
- Why are the noise vertices inserted in a predictable manner?
- I am curious about this 0.3 relative knowledge ratio threshold where it is reported that compositional reasoning emerges. Could it be that 0.3 is when the probability that there is at-least one occurrence for every (v_i, v_{i+1}) in the train set reaches close to 1?
- Why is there a drop in performance in Figure 2 (right) and relative knowledge of 0.25?

---

> ### Author Response · Authors · 2024-11-21
> **Response for Reviewer oHAw (PART 1)**
>
> Thank you for the thorough and detailed review! Especially for the novel angle in adding extra experiment and careful reading of technical details. We address your concerns as follows.
>
> **W1:** The clarity of the paper is lacking, especially the notation and writing. There is a typo in Figure 1. Concrete examples are needed to improve readers’ understanding
>
> **A for W1:**
>  As for the clarity and concrete example, we do the following modification to make the paper more clear and readable:
> - Notations in Section 3:
>   - In Step 1.1  "Fragmented at Training", we provide a concrete example of the training sequence [A, 100, Z, 3, B, 101, H, 1], where [A, B] is the child chain from the causal structure with values following the ``+1" operation, and [Z, H] are noise with randomly assigned values [Lines 200-202].
>   - In Step 3 "Downstream Processing", extraneous notation details have been removed. A concrete example is used to clarify the processed sentence: "H=?: ... A=110, ... Z=1, ... B=111, ... H=5 \n H=?: ... A=102, ... Z=4, ... B=103, ... H=9" [Lines 218-221]. An example of the sentence with complete context information is like “H=?: ARCH Rewards patentsA=102, Pad Influ interfZ=4, google correl latexB=103, treasureFrankH=9”. Such concrete examples are shown in the Appendix C in the revised version [Lines 832-896].
> - Notations in Section 5:
>   - We replaced the abstract notations $o^{\texttt{cm}}$ and $o^{\texttt{eq}}$ with the actual symbols, using a comma “,” and an equals sign “=” [Lines 393-394, Lines 396-397].
> - Notations in Section 6.1:
>   - Similarly, we revised notations $o^{\texttt{cm}}$ and $o^{\texttt{eq}}$ to their respective symbols “,” and “=”. We also emphasized comma tokens at different positions by enclosing them in boxes and using various colors for improved clarity [Lines 490-491].
>
> There is a typo in Figure 1. We believe that this corresponds to your second question “where B=106 and B=103, should it be C=108 and C=105, respectively?”. The answer is yes. The “C=109” and “C=106” at the lower right corner of Figure 1 should be changed to “C=108” and “C=105” according to the relationship q(C)=q(B)+2 defined by the causal structure. We sincerely appreciate your attention to detail in identifying this typo. A corrected version has now been uploaded  [Lines 170-173].
>
>
> **W2:** The formal definition of the compositional reasoning should be explicitly written out.
>
> **A for W2:**
> We add such formal definition explicitly in the introduction section of the revised version [Lines 34-37]. Specifically, we define the compositional reasoning as “the skill to integrate discrete pieces of knowledge from multiple sources to form a coherent reasoning, even in the absence of explicit examples connecting these pieces during learning.”
>
>
> **W3:** Lack of the testing performance of reasoning with incomplete intermediate reasoning steps (reasoning about AC).
>
> **A for W3:**
> Our interpretation of "reasoning about AC" refers to reasoning when not all vertices in the reasoning paths are adjacent in the causal structure—essentially the ability to reason with incomplete intermediate steps.
>
> It is true that "reasoning about AC" is an aspect of compositional reasoning. **However, our research focus, which is validating compositional reasoning ability and doing mechanism analysis , does not necessarily require testing such performance.** A model is deemed to possess compositional reasoning ability as long as it can correctly determine the value of the last vertex in an OOD testing sentence, regardless of how that value is generated. In this study, we choose to elicit Transformers’ compositional reasoning ability by training them for complete step-by-step reasoning, which does not inherently ensure proficiency in handling incomplete intermediate steps. To avoid potential misunderstandings, we have revised the abstract example to:  “For example, if someone learns ( B = f(A) ) from one source and ( C = g(B) ) from another, they can deduce the value of ( C ) by reasoning ( C=g(B)=g(f(A)) ) even without encountering ( ABC ) together”  [Lines 13-16].
>
> Additionally, we have included an experiment on reasoning with incomplete intermediate steps (reasoning about AC) in Appendix M [Lines 1722-1744]. **The results in Figure 11 [Lines 1782-1835] indicate that Transformers trained on FTCT exhibit limited proficiency when reasoning with any level of incompleteness.** This limitation likely stems from a bias in the training dataset, where adjacent vertices consistently appear consecutively in the sequences.

---

> ### Author Response · Authors · 2024-11-21
> **Response for Reviewer oHAw (PART 2)**
>
> **Q1:** Will the model be able to chain together the values of non-adjacent vertices during testing time?
>
> **A for Q1:** We've included testing performance for reasoning with incomplete intermediate steps with non-adjacent vertices (reasoning about AC) in the  Appendix M [Lines 1722-1744]. The testing results reveal that Transformers trained on FTCT are unable to effectively chain together the values of non-adjacent vertices during testing. This limitation is likely due to a bias in the training dataset, where adjacent vertices invariably appear consecutively in the training sequences. Nevertheless, as elaborated in our response to **W3**, this limitation does not affect the main conclusions presented in this paper.
>
>
> **Q2:** Is there a typo in Figure 1?
>
> **A for Q2:** There is a typo in Figure 1, and the values of “C”s should be 108 and 105 respectively.  We have submitted a correct figure in the revised version  [Lines 170-173].
>
>
> **Q3:** Are there more than one set of causal chains? It seems that there is only one sequence of length $n$.
>
> **A for Q3:** Indeed, there is only one set of causal chains, and all different chains with various vertices and lengths are encompassed within this set. As defined in Section 3.1 [Line 187], the set of chains $\mathcal{T}(\mathcal{G})$ includes all sequences of connected vertices. The index $n$ can be any natural number as long as that $[v_1, ..., v_n]$ are connected by edges. For the causal structure shown in Figure 1 [Lines 162-174], $\mathcal{T}(\mathcal{G})$ includes sequences [A, B], [B, C], [A, D], [D, E], [A, B, C], and [A, D, E].
>
>
> **Q4:** Why are the noise vertices inserted in a predictable manner?
>
> **A for Q4:** We interpret "predictable manner" to mean that within a few-shot example, the noise vertices retain the same letters in the same positions across shots. This design is intended to evoke the model's generalized in-context learning ability by encouraging it to replicate the same vertices from prior examples, thereby ensuring high vertex accuracy during testing. While the appearance of noise vertices is predictable, it does not diminish the significance of the observed emergence of compositional reasoning.
>
>
> **Q5:** Why is a relative knowledge ratio of 0.3 the threshold of the emergence of compositional reasoning? Could it be that 0.3 is when the probability that there is at least one occurrence for every $(v_i, v_{i+1})$ in the train set reaches close to 1?
>
> **A for Q5:** Our analysis shows that occurrence probability is not the determining factor for the 0.3 threshold. For every tuple of adjacent vertices and their values $(v_i, q_i, v_{i+1}, \texttt{op}(v_i, v_{i+1})\circ q_i)$, we count the number of times that it has been visited in our training data. Results indicate that every tuple is encountered at least 46 times, across tasks with varying knowledge ratios. The large volume of our training data ensures that, even with a low relative knowledge ratio, the probability of each $(v_i, q_i, v_{i+1}, \texttt{op}(v_i, v_{i+1}) \circ q_i)$ appearing in the training data is nearly 1. **Detailed experiments results are available in the Appendix I [Lines 1452-1500] of the revised paper.**
>
> What we can confirm so far is that a higher relative knowledge ratio aligns few-shot CoT prompts more closely with training data, thereby enhancing testing performance. Understanding the precise mechanism behind this specific threshold remains a future research direction.
>
>
> **Q6:** Why is there a drop in performance when the relative knowledge ratio reaches 0.25?
>
> **A for Q6:** The relative knowledge ratio of 0.25 is associated with a training configuration where the graph depth is 8 and the child chain length is 2. Testing curves in Figure 2 (left) [Lines 289-308] and Figure 3 [Lines 1358-1376] show that every training task with a child chain length of 2 performs poorly. That is the reason why the performance drop abruptly at 0.25.
>
> Our empirical findings in Figure 2 (right) [Lines 289-308] indicate a strong correlation between compositional reasoning ability and the relative knowledge ratio; however, this does not imply that compositional reasoning ability increases monotonically with the relative knowledge ratio. Indeed, if we set the relative knowledge ratio to $(M/N)^\alpha$, where $\alpha$ is a tunable parameter, an $\alpha$ can be found that results in an apparently monotonic curve.

---

> ### Comment · Reviewer_oHAw · 2024-11-27
>
> Thank you for your explanation and updates to the paper! I lean towards accepting this paper and maintain my score.

---

> > ### Author Response · Authors · 2024-11-27
> > **Thank you for the updated review**
> >
> > Thank you once again for your detailed review and inspiring opinions! They are valuable to us.

---

### Official Review · Reviewer_EQsG · 2024-11-03

**Soundness:** 2
**Presentation:** 3
**Contribution:** 2
**Rating:** 6
**Confidence:** 3

**Summary:**

This paper introduces the "FTCT" (Fragmented at Training, Chained at Testing) task to evaluate if Transformers can perform compositional reasoning similar to humans. The task involves training models on separate knowledge fragments and testing them on integrating these fragments to form complete causal graph traces. The study finds that few-shot Chain-of-Thought prompting helps Transformers combine these fragments correctly, even without seeing such combinations during training. The results indicate that model complexity and the data's knowledge ratio play a role in enabling this skill. The authors provide theoretical and empirical evidence for their claims, showing that Transformers can learn a generalizable program to aid in compositional reasoning. The findings are interesting and suggest potential areas for further exploration.

**Strengths:**

- The design of the FTCT task is well-conceived, as it effectively mimics real-world scenarios where knowledge is often fragmented and must be integrated to draw comprehensive conclusions. This setup provides a meaningful and practical benchmark to evaluate the compositional reasoning abilities of Transformers, making the study relevant and valuable for advancing our understanding of machine learning models' capabilities.
- Chapter 5, "transformer does compositional reasoning via the underlying program", is very interesting as it explores the possible underlying mechanisms and principles that allow Transformers to perform compositional generalization. This chapter goes beyond just presenting empirical results by looking into how these models might internally handle and integrate fragmented knowledge. This deeper investigation adds value by giving us a better understanding of how Transformers achieve complex reasoning tasks.

**Weaknesses:**

- While the task studied in this paper requires strong compositional generalization abilities, it is simple and singular in its form. Generally, using a simple and singular synthetic dataset is suitable for highlighting the shortcomings of the Transformer architecture. However, since the paper concludes that Transformers possess this capability, the experiments on this task alone are not sufficient to support such a conclusion. I believe that more diverse and comprehensive tasks are needed, and ideally, this capability should also be validated on complex real-world tasks.
- In the related work section, the paper discusses various tasks used to probe compositional generalization abilities. The authors mention that these existing tasks have not been studied from the perspectives of few-shot prompting and chain-of-thought reasoning. However, this distinction alone is insufficient; if the difference is merely in this aspect, it would be possible to modify existing datasets instead of creating a new one. The novelty of this newly created task is not demonstrated well. Therefore, the authors need to provide more explanation regarding the motivation and innovation behind the proposed task.
- The experiments in this paper use Transformers with relatively small parameter sizes. It is unclear whether the conclusions drawn from these experiments would hold true for larger Transformer models. This limitation raises questions about the generalizability of the findings to more complex and sizable architectures.

**Questions:**

My main concerns have already been expressed in the "weakness" section.

---

> ### Author Response · Authors · 2024-11-21
> **Response for Reviewer EQsG (PART 1)**
>
> Thank you for your thorough review and valuable opinions! We address your concerns as follows.
>
> **W1:** More diverse, comprehensive and complex real-world tasks are needed to draw the conclusion that Transformers possess the compositional reasoning ability.
>
> **A for W1:** As for the need of real world tasks, we would like to clarify that our main focus and contribution are not about proving the ability of real pre-trained language models on natural language settings. In fact, many empirical works have already demonstrated the symptom that well-pretrained language models do compositional reasoning on complex realistic tasks like question answering, mathematical reasoning and interdisciplinary generation ([1], [2], [3] [4]). They are shown to generate comprehensive content with elements unlikely to co-occur in training data. **We aim to scientifically validate such ability within a clear environment and explore its underlying mechanism.** To this end, it is necessary to conduct controlled and tunable experiments where reasoning paths in the test data neither appear in the training data nor are they encoded in the models' pre-existing weights. Compared with realistic tasks with intractable natural language corpus, we believe that a controllable synthetic task is more appropriate for our research purpose. Doing research on synthetic data is also an approach being widely used in previous interpretability studies ([5], [6], [7]).
>
> As for the diversity and comprehensiveness, we argue that the experiments on FTCT is enough for the potential validation and mechanism investigation research purposes. Nonetheless, we acknowledge that incorporating more diverse and comprehensive tasks will enhance the generalizability and impact of our work, which we aim to pursue in future research.
>
> To clarify the above argument, we attach a new version of the paper that includes following modifications:
>
> - Refine the research purpose in the introduction:  “This paper validates the potential of Transformers in doing compositional reasoning on synthetic dataset and investigates the inner mechanism eliciting such ability. ” [Lines 45-46].
> - Pointing out that utilizing models trained on vast natural language is not proper for our research purpose: “However, the complexity and ambiguity of their natural language training and testing data make it hard to scientifically validate the compositional reasoning ability and explore the underlying mechanisms.” [Lines 42-44].
> - Reframe the statement about relative knowledge ratio within the scope of FTCT: “compositional reasoning emerges as the similarity between training and testing data increases. In FTCT, this is measured by the relative knowledge ratio” [Lines 69-70].
> - Adjust the conclusion: “Our research validates the potential of Transformers in doing compositional reasoning on synthetic data and investigates the inner mechanism eliciting such ability. ”[Lines 527-528].
>
> **W2:** More explanation about the motivation and innovation behind our task is needed to demonstrate the novelty and contribution of the FTCT dataset compared with existing tasks.
>
> **A for W2:** We rewrite the “Compositional generalization” part in related works section in the new submitted version [Lines 124-141], in which we clarify the motivation and innovation of our FTCT tasks compared with existing works. The main logic of the new version is as follows.
> - Firstly, a series of works has showcased the potential of Transformers in compositional tasks where the answers are directly output without intermediate reasoning steps ([8], [9], [10], [11], [12]). In contrast, our FTCT dataset with deep causal structure allows exploration of explicit reasoning's impact on compositional generalization ability.
> - Further, empirical studies show that step-by-step reasoning enhances large language models' compositional abilities on real-world tasks like question answering and mathematical reasoning ([1], [2], [3]). However, the complexity of natural language corpora used by them complicates scientific validation compared to our synthetic data.
> - Recent studies have explored Transformers' generalized reasoning on controllable synthetic tasks ([13], [6], [14]).  In contrast, our FTCT task not only ensures controlled experimentation but also introduces measures of training-testing data similarity and establishes a distinct parent-child causal relationship, facilitating analysis of the mechanisms underlying Transformers' compositional abilities concerning data distribution and model structure.
>
> We believe that the above modifications will better demonstrate the novelty and contribution of our FTCT learning task.

---

> > ### Author Response · Authors · 2024-11-21
> > **Response for Reviewer EQsG (PART 2)**
> >
> > **W3:** It is questionable that whether the findings on the small Transfomers can be generalized to more complex and sizable architectures.
> >
> > **A for W3:** We evaluated the performance of GPT2-small (12 layers, 12 heads, 117M parameters) and GPT2-large (36 layers, 20 heads, 774M parameters) on the FTCT data, with results detailed in Appendix J [Lines 1502-1611] of the revised version. These models exhibit the same pattern: compositional reasoning ability emerges with increased shot numbers and relative knowledge ratios, suggesting our findings are generalizable to more complex and sizable architectures. However, the performance of these larger models is less stable compared to smaller ones, likely due to overfitting.
> >
> >
> > [1] Press, Ofir, et al. "Measuring and narrowing the compositionality gap in language models." *arXiv preprint arXiv:2210.03350* (2022).
> >
> > [2] Zhou, Denny, et al. "Least-to-most prompting enables complex reasoning in large language models." *arXiv preprint arXiv:2205.10625* (2022).
> >
> > [3] Khot, Tushar, et al. "Decomposed prompting: A modular approach for solving complex tasks." *arXiv preprint arXiv:2210.02406* (2022).
> >
> > [4] Bubeck, Sébastien, et al. "Sparks of artificial general intelligence: Early experiments with gpt-4." *arXiv preprint arXiv:2303.12712* (2023).
> >
> > [5] Chan, Stephanie, et al. "Data distributional properties drive emergent in-context learning in transformers." *Advances in Neural Information Processing Systems* 35 (2022): 18878-18891.
> >
> > [6] Allen-Zhu, Zeyuan, and Yuanzhi Li. "Physics of language models: Part 1, context-free grammar." *arXiv preprint arXiv:2305.13673* (2023).
> >
> > [7] Bietti, Alberto, et al. "Birth of a transformer: A memory viewpoint." *Advances in Neural Information Processing Systems* 36 (2024).
> >
> > [8] Hupkes, Dieuwke, et al. "Compositionality decomposed: How do neural networks generalise?." *Journal of Artificial Intelligence Research* 67 (2020): 757-795.
> >
> > [9] Arora, Sanjeev, and Anirudh Goyal. "A theory for emergence of complex skills in language models." *arXiv preprint arXiv:2307.15936* (2023).
> >
> > [10] Yu, Dingli, et al. "Skill-Mix: A flexible and expandable family of evaluations for AI models." *arXiv preprint arXiv:2310.17567*(2023).
> >
> > [11] Xu, Zhuoyan, Zhenmei Shi, and Yingyu Liang. "Do large language models have compositional ability? an investigation into limitations and scalability." *ICLR 2024 Workshop on Mathematical and Empirical Understanding of Foundation Models*. 2024.
> >
> > [12] Treutlein, Johannes, et al. "Connecting the dots: Llms can infer and verbalize latent structure from disparate training data." *arXiv preprint arXiv:2406.14546* (2024).
> >
> > [13] Ramesh, Rahul, et al. "How capable can a transformer become? a study on synthetic, interpretable tasks." *arXiv preprint arXiv:2311.12997* (2023).
> >
> > [14] Ye, Tian, et al. "Physics of language models: Part 2.1, grade-school math and the hidden reasoning process." *arXiv preprint arXiv:2407.20311* (2024).

---

> > > ### Comment · Reviewer_EQsG · 2024-11-25
> > >
> > > Thanks for the response. I would like to maintain my score and lean towards accepting the paper.

---

> ### Author Response · Authors · 2024-11-25
> **Thank you for the updated review**
>
> Thank you once again for your careful review and constructive questions, which are valuable to us!

---

### Official Review · Reviewer_n7TN · 2024-11-04

**Soundness:** 3
**Presentation:** 2
**Contribution:** 3
**Rating:** 6
**Confidence:** 4

**Summary:**

The work sets out to investigate whether transformers are capable of generalizing to longer reasoning chains through connecting shorter ones seen in the training stage. The authors introduce "Fragemented at Training, Chained at Testing" learning task to train a randomly initialized 3-layer 3-head GPT2-like transformer. They find that with few-shot chain-of-thought prompting, transformers can perform good compositional reasoning skills by combineing fragments together. The authors further show that the generalization performance highly correlates to model complexity (require multiple-layer attention structure) and high relative knowledge ratio of training data. The paper also discusses the internal working of the model (learn an underlying generalizable program) to interpret the transformer's generalization behaviors and provide theoretical insights on transformer's expressivity on learning a such program.

**Strengths:**

1. The paper investigates whether transformers are capable of generalizing to longer reasoning chains through connecting shorter ones seen in the training stage, which is an interesting and important research question.
2. The paper is technically sound: the trained transformers behave compositionally (with few-shot chain-of-thought prompting) and the authors provide insights on its internal workings: induction head and attention assignment, demonstrating that the transformer learn a generalizable program in its internal computing.
3. Authors also theoretically prove that Transformers have the expressivity to simulate the generalizble underlying program.

**Weaknesses:**

1. Since the experiment setting is a randomly initialized transformer trained on synthetic data, to what extent the paper's conclusion can be extended to real pre-trained language models is questionable.
2. the notations used in the paper are quite complicated, making the paper a little bit difficult for readers to follow.

**Questions:**

1. In the FTCT learning task (e.g., Figure 1), why in the $D_{train}$, we need to add noisy tokens in the token sequence? Why in the $D_{test}$ we do not add noisy tokens in the prompt?

---

> ### Author Response · Authors · 2024-11-21
> **Response for Reviewer n7TN (PART 1)**
>
> Thank you for the thoughtful review and the valuable feedback! We address your concerns as follows.
>
> **W1:** It is questionable that to what extent the conclusions made by training randomly initialized transformers on synthetic data can be extended to real pre-trained language models.
>
> **A for W1:** We would like to clarify that our primary focus and contribution are not about investigating the performance of real pre-trained language models. As a matter of fact, many empirical works have already demonstrated that real pre-trained language models exhibit symptoms of compositional reasoning by generating comprehensive content with elements unlikely to co-occur in training data ([1], [2], [3], [4]).  **We aim to scientifically validate such ability within a clear environment and explore its underlying mechanism.** To this end, it is necessary to conduct controlled and tunable experiments where reasoning paths in the test data neither appear in the training data nor are they encoded in the models' pre-existing weights. Compared with investigating real pre-trained LLMs that have learned intractable corpus, we believe that training randomly initialized Transformers on synthetic data is an appropriate method for our research purpose. Doing research on synthetic data is also an approach being widely used in previous interpretability studies ([5], [6], [7]).
>
> To clarify the above points, we attach a new version of the paper that includes following modifications:
> - Refine the research purpose in the introduction:  “This paper validates the potential of Transformers in doing compositional reasoning on synthetic dataset and investigates the inner mechanism eliciting such ability. ” [Lines 45-46].
> - Pointing out that utilizing models trained on vast natural language is not proper for our research purpose: “However, the complexity and ambiguity of their natural language training and testing data make it hard to scientifically validate the compositional reasoning ability and explore the underlying mechanisms.” [Lines 42-44].
> - Reframe the statement about relative knowledge ratio within the scope of FTCT: “compositional reasoning emerges as the similarity between training and testing data increases. In FTCT, this is measured by the relative knowledge ratio” [Lines 69-70].
> - Adjust the conclusion: “Our research validates the potential of Transformers in doing compositional reasoning on synthetic data and investigates the inner mechanism eliciting such ability. ”[Lines 527-528].
>
> We acknowledge that conducting comprehensive scientific interpretability studies on large language models trained on complex natural corpora is indeed valuable. However, such endeavors remain challenging amidst current research advancements. We hope our research contributes meaningfully towards achieving this ultimate goal.
>
> **W2:** Complicated notations make the paper difficult to follow.
>
> **A for W2:** To simplify the notations and make the paper easier to follow, we have made the following changes:
> - Notations in Section 3:
>   - In Step 1.1  "Fragmented at Training", we provide a detailed example of the training sequence [A, 100, Z, 3, B, 101, H, 1] along with its generation process [Lines 201-202].
>   - In Step 3 "Downstream Processing", extraneous notation details have been removed. A concrete example is used to clarify the processed sentence: "H=?: ... A=110, ... Z=1, ... B=111, ... H=5 \n H=?: ... A=102, ... Z=4, ... B=103, ... H=9" [Lines 218-221].
> - Notations in Section 5:
>   - We replaced the abstract notations $o^\texttt{cm}$ and $o^\texttt{eq}$ with the actual symbols, using a comma “,” and an equals sign “=” [Lines 393-394, Lines 396-397].
> - Notations in Section 6.1:
>   - Similarly, we revised notations $o^\texttt{cm}$ and $o^\texttt{eq}$ to their respective symbols “,” and “=”. We also emphasized comma tokens at different positions by enclosing them in boxes and using various colors for improved clarity [Lines 490-491].
>
> **Q1.1:** Why in $D_{train}$ we need to add noisy tokens?
>
> **A for Q1.1:** Noisy tokens are added to simulate the dependencies found in natural language corpora, where related tokens often do not appear consecutively and are interrupted by unrelated tokens. This requires Transformers to extract meaningful information from a noisy context.
>
> **Q1.2:** Why in $D_{test}$ we do not add noisy tokens?
>
> **A for Q1.2:** Prompts in $D_{test}$ are designed without noisy tokens to reflect the typical scenario in which users pose questions to language models, where the given prompts are usually of higher quality and contain less noise.
>
> We include the testing results where testing examples are blurred by noise in the same way as the training data are processed. **These results, shown in Appendix H.2 [Lines 1401-1451], display the same overall pattern as tests without noisy tokens, indicating the robustness of our conclusions.**

---

> > ### Author Response · Authors · 2024-11-21
> > **Response for Reviewer n7TN (PART 2)**
> >
> > [1] Press, Ofir, et al. "Measuring and narrowing the compositionality gap in language models." *arXiv preprint arXiv:2210.03350* (2022).
> >
> > [2] Zhou, Denny, et al. "Least-to-most prompting enables complex reasoning in large language models." *arXiv preprint arXiv:2205.10625* (2022).
> >
> > [3] Khot, Tushar, et al. "Decomposed prompting: A modular approach for solving complex tasks." *arXiv preprint arXiv:2210.02406* (2022).
> >
> > [4] Bubeck, Sébastien, et al. "Sparks of artificial general intelligence: Early experiments with gpt-4." *arXiv preprint arXiv:2303.12712* (2023).
> >
> > [5] Chan, Stephanie, et al. "Data distributional properties drive emergent in-context learning in transformers." *Advances in Neural Information Processing Systems* 35 (2022): 18878-18891.
> >
> > [6] Allen-Zhu, Zeyuan, and Yuanzhi Li. "Physics of language models: Part 1, context-free grammar." *arXiv preprint arXiv:2305.13673* (2023).
> >
> > [7] Bietti, Alberto, et al. "Birth of a transformer: A memory viewpoint." *Advances in Neural Information Processing Systems* 36 (2024).

---

> > ### Comment · Reviewer_n7TN · 2024-11-22
> > **Thanks for the response**
> >
> > Thanks for the response. Many of my previous concerns have been properly addressed (except for W1). As for the further explanation for W2 and Q1, please carefully append them to the revised version of the paper. I would like to maintain my score and lean towards accepting the paper (I raise my confidence score from 3 to 4).

---

> ### Author Response · Authors · 2024-11-23
> **Thank you for the updated review**
>
> Thank you for your constructive opinions and for increasing your confidence score. We are glad to hear that many of your concerns have been addressed by our responses.
>
> The simplifications and refinements we mentioned in response to **W2** have been incorporated into the revised version of our paper, corresponding to the lines noted in our response. Additionally, our answer to **Q1** can be found in Appendix H.2 [Lines 1401-1451].
>
> We would like to further explain the generality of our work (**W1**). Empirical studies ([1], [2], [3], [4]) indicate that real LLMs exhibit compositional reasoning abilities. For example, GPT-4 can "write a supporting letter for Electron as a US presidential candidate” ([4]), which logically combines fields not likely to co-occur during training. However, scientific validation and a detailed mechanistic understanding of this capability are limited by the complexity of natural data LLMs are trained on. We believe our experiments and analysis using synthetic data contribute to this understanding. (Same approach has been used in physics research such as the investigation of ball lightning. Physicists have replicated spherical luminous balls through artificial laboratory experiments ([5], [6]), thereby confirming its potential formation and providing explanations through hypothetical models.)
>
> Notably, many studies ([7], [8], [9], [10]) have utilized Transformers trained on synthetic data from scratch to explore their potential and analyze their mechanisms. For instance, Zhou et al. ([7]) investigate factors influencing Transformers' length generalization by training them to predict the addition of two integers. The controlled and structured nature of synthetic data, along with the unbiasedness of randomly initialized models, enhances the precision and scientific rigor of such investigations, aligning well with our research objectives.
>
> Thank you once again for your valuable opinions and insightful discussion—they mean a great deal to us.
>
> [1] Press, Ofir, et al. "Measuring and narrowing the compositionality gap in language models." *arXiv preprint arXiv:2210.03350* (2022).
>
> [2] Zhou, Denny, et al. "Least-to-most prompting enables complex reasoning in large language models." *arXiv preprint arXiv:2205.10625* (2022).
>
> [3] Khot, Tushar, et al. "Decomposed prompting: A modular approach for solving complex tasks." *arXiv preprint arXiv:2210.02406* (2022).
>
> [4] Bubeck, Sébastien, et al. "Sparks of artificial general intelligence: Early experiments with gpt-4." *arXiv preprint arXiv:2303.12712* (2023).
>
> [5] Paiva, Gerson Silva, et al. "Production of ball-lightning-like luminous balls by electrical discharges in silicon." *Physical review letters* 98.4 (2007): 048501.
>
> [6] Wu, H-C. "Relativistic-microwave theory of ball lightning." *Scientific reports* 6.1 (2016): 28263.
>
> [7] Zhou, Yongchao, et al. "Transformers can achieve length generalization but not robustly." *arXiv preprint arXiv:2402.09371* (2024).
>
> [8] Bietti, Alberto, et al. "Birth of a transformer: A memory viewpoint." *Advances in Neural Information Processing Systems* 36 (2024).
>
> [9] Wang, Zixuan, et al. "Transformers Provably Learn Sparse Token Selection While Fully-Connected Nets Cannot." *arXiv preprint arXiv:2406.06893* (2024).
>
> [10] Allen-Zhu, Zeyuan, and Yuanzhi Li. "Physics of language models: Part 1, context-free grammar." *arXiv preprint arXiv:2305.13673* (2023).

---

### Author Response · Authors · 2024-11-21
**Common Problems**

We summarize common problems concerned by multiple reviewers and our responses to them.

**Generality of Our Conclusions**

Concerns have been raised by reviewers n7TN, EQsG and hQE3 regarding whether our conclusions, derived from training randomly initialized transformers on a singular synthetic task, are sufficiently generalizable to realistic natural language scenarios. Our response is as follows:

Our primary focus and contribution do not center on evaluating the performance of pre-trained models on natural language tasks. Numerous empirical studies have already demonstrated that well-pretrained language models exhibit compositional reasoning on complex tasks such as question answering, mathematical reasoning, and interdisciplinary content generation ([1], [2], [3], [4]). These models generate comprehensive content with elements that rarely co-occur in training data. Our objective is to scientifically validate this capability in a clear setting and explore its underlying mechanisms. To achieve this, it is essential to conduct controlled and tunable experiments where reasoning paths in the test data are distinct from those in the training data and not pre-encoded in the models' weights. In comparison to naturalistic tasks that train and test on intractable natural language corpora, a controllable synthetic task is more suited to our research objectives. Moreover, using synthetic data is a widely employed approach in previous interpretability studies ([5], [6], [7]).

To clarify our research purpose and contributions, we have made modifications outlined in the **Clarifications of the Research Purpose and Contribution** part in next comment.

While comprehensive scientific interpretability studies on large language models trained on intricate natural corpora are indeed valuable, they remain challenging amidst current research advancements. We hope our research contributes effectively to this ultimate goal.

**Clarity of the Notations**

Reviewers n7TN, oHAw and hQE3 have noted that the clarity of our paper, particularly the notations in Section 3 about introducing the FTCT dataset, should be improved. To enhance the clarity of our presentation, we have made the modifications detailed in the **Notation Simplification and Presentation Improvement** part in next comment.

[1] Press, Ofir, et al. "Measuring and narrowing the compositionality gap in language models." *arXiv preprint arXiv:2210.03350* (2022).

[2] Zhou, Denny, et al. "Least-to-most prompting enables complex reasoning in large language models." *arXiv preprint arXiv:2205.10625* (2022).

[3] Khot, Tushar, et al. "Decomposed prompting: A modular approach for solving complex tasks." *arXiv preprint arXiv:2210.02406* (2022).

[4] Bubeck, Sébastien, et al. "Sparks of artificial general intelligence: Early experiments with gpt-4." *arXiv preprint arXiv:2303.12712* (2023).

[5] Chan, Stephanie, et al. "Data distributional properties drive emergent in-context learning in transformers." *Advances in Neural Information Processing Systems* 35 (2022): 18878-18891.

[6] Allen-Zhu, Zeyuan, and Yuanzhi Li. "Physics of language models: Part 1, context-free grammar." *arXiv preprint arXiv:2305.13673* (2023).

[7] Bietti, Alberto, et al. "Birth of a transformer: A memory viewpoint." *Advances in Neural Information Processing Systems* 36 (2024).

---

> ### Author Response · Authors · 2024-11-21
> **Main Modifications**
>
> We summarize the main modifications made in the revised version.
>
> **Clarifications of the Research Purpose and Contribution**
>
> - Refine the research purpose in the introduction:  “This paper validates the potential of Transformers in doing compositional reasoning on synthetic dataset and investigates the inner mechanism eliciting such ability. ” [Lines 45-46].
> - Pointing out that utilizing models trained on vast natural language is not proper for our research purpose: “However, the complexity and ambiguity of their natural language training and testing data make it hard to scientifically validate the compositional reasoning ability and explore the underlying mechanisms.” [Lines 42-44].
> - Reframe the statement about relative knowledge ratio within the scope of FTCT: “compositional reasoning emerges as the similarity between training and testing data increases. In FTCT, this is measured by the relative knowledge ratio” [Lines 69-70].
> - Change the title of section 4.2: “The Similarity between Training and Testing Data Determines the Emergence of Compositional Reasoning” [Lines 310-312].
> - Adjust the conclusion: “Our research validates the potential of Transformers in doing compositional reasoning on synthetic data and investigates the inner mechanism eliciting such ability. ”[Lines 527-528].
>
> **Notification Simplification and Presentation Improvement**
> - Notations in Section 3:
>   - In Step 1.1 "Fragmented at Training", we provide a detailed example of the training sequence [A, 100, Z, 3, B, 101, H, 1] along with its generation process [Lines 201-202].
>   - In Step 2 “Few-shot learning”, the index for few-shot examples has been changed from (f) to (k), a more standard notation indicating shot numbers [Lins 208-213].
>   - In Step 3, we change the "downside processing" in the original version to "downstream processing" which is a more conventional terminology [Lines 216].
>   - In Step 3 "Downstream Processing", extraneous notation details have been removed. A concrete example is used to clarify the processed sentence: "H=?: ... A=110, ... Z=1, ... B=111, ... H=5 \n H=?: ... A=102, ... Z=4, ... B=103, ... H=9" [Lines 218-221].
> - Notations in Section 5:
>   - We replaced the abstract notations (o^\cm) and (o^\eq) with the actual symbols, using a comma “,” and an equals sign “=” [Lines 393-394, Lines 396-397].
> - Notations in Section 6.1:
>   - Similarly, we revised notations (o^\cm) and (o^\eq) to their respective symbols “,” and “=”. We also emphasized comma tokens at different positions by enclosing them in boxes and using various colors for improved clarity [Lines 490-491].
> - The demonstration of Figure 1:
>   - The values of “C”s at the lower right corner of Figure 1 have been changed to 108 and 105 respectively [Lines 170-173] .
>   - A self-explanatory caption of Figure 1 has been added in the newly submitted revised version [Lines 176-183].
>   - We distinguish knowledge points and noisy nodes with gray and blue only at the beginning of the “Fragmented at Training” stage and remove all subsequent highlights to avoid confusion [Lines 163-166].
> - Other improvements
>   - We add the formal definition of composition reasoning in the introduction [Lines 34-37].
>   - We change the description of the ABC example in the abstract to indicate our focus on step-by-step reasoning skills [Lines 13-16].
>
> **More thorough and clear discussions in related works**
> - In the “Step-by-step reasoning” part (formerly titled “Chain-of-Thought Prompting”), we have expanded on how Prystawski et al.'s work has informed our research and delineated the fundamental differences between our studies [Lines 107-111].
> - In the “Compositional generalization” part, we have clarified the motivation and innovations of our FTCT tasks compared to existing works, focusing on step-by-step reasoning, data complexity, and the unique structure that facilitates detailed mechanism analysis [Lines 124-141].
>
> **Relaxation of Evaluation Criteria for Compositional Reasoning Ability**
>
> In the revised version, we have adjusted the whole chain accuracy standard to assess if the model's output includes all vertices and values in the correct order [Lines 255-258].  Compared to the original version that requires vertices and values to be presented consecutively, this revised approach considers additional valid reasoning paths that are disturbed by noise.
> We have updated the testing results using these revised criteria in the new version (Figure 2 [Lines 289-302] and Figure 3 [Lines 1358-1376]). The overall pattern remains consistent with the original version which does not affect our conclusions or contributions.

---

> > ### Author Response · Authors · 2024-11-21
> > **Extra Experiments**
> >
> > During the rebuttal process, we conducted the following additional experiments to address reviewers’ concerns.
> >
> > **Testing Performance with Noisy Tokens**
> >
> > In response to reviewer n7TN’s inquiry about the absence of noisy tokens in $D_{test}$, we included testing results where examples are blurred by noise in the same manner as the training data. These findings, presented in Appendix H.2 [Lines 1401-1451], reveal the same overall pattern as tests without noisy tokens, affirming the robustness of our conclusions.
> >
> > **Conclusions on Larger Transformer Models**
> >
> > To address reviewer EQsG’s concern about the generalizability of findings from small Transformers to larger architectures, we evaluated GPT2-small (12 layers, 12 heads, 117M parameters) and GPT2-large (36 layers, 20 heads, 774M parameters) on the FTCT data, with results in Appendix J [Lines 1502-1611]. These models display a similar pattern, where compositional reasoning ability emerges with increased shot numbers and relative knowledge ratios. This suggests our findings are applicable to more complex architectures, although the performance of the larger models is less stable, likely due to overfitting.
> >
> > **Testing Performance with Incomplete intermediate Reasoning Steps**
> >
> > In response to reviewer oHAw’s query about the model's ability to link values of non-adjacent vertices during testing, we included the testing performance for reasoning with incomplete steps in Appendix M [Lines 1722-1744]. Results indicate that Transformers trained on FTCT struggle to link non-adjacent vertices, likely due to a training dataset bias where adjacent vertices appear consecutively.
> >
> > This limitation does not affect the main conclusions presented in this paper,  as a model exhibiting robust compositional reasoning ability is not necessarily required to handle non-adjacent vertices directly within its test data. We can prove the compositional reasoning ability by models doing complete reasoning as long as the test data have not appeared in their entirety during training.
> >
> > **Least Visited Times for All Adjacent Vertices and Their Values**
> >
> > To certify reviewer oHAw’s hypothesis about the reason why relative knowledge ratio of 0.3 is the threshold of the phase change, we count the times that each tuple $(v_i, q_i, v_{i+1}, \texttt{op}(v_i, v_{i+1})\circ q_i)$ is visited in Appendix I [Lines 1452-1500]. Results show that every tuple is visited at least 46 times, across tasks with varying knowledge ratios, indicating that occurrence probability is not the determining factor for the 0.3 threshold.
> >
> > **Performance of Transformers in reasoning causal chains with varying test lengths**
> >
> > To address reviewer hQE3’s concern regarding the unexpected pattern where accuracy decreases with more than one few-shot prompt, we test the performance of Transformers in reasoning causal chains with varying test lengths in Appendix L [Lines 1613-1673]. The results (Figure 10 [Lines 1653-1673]) show that for tasks where test lengths are close to child chain lengths models are trained on, few-shot performance remains stable without decrease. As the gap between child chain and test lengths widens, the performance decrease after one shot becomes evident.  Thus, we conclude that when differences between training and testing data are limited, the expected pattern of in-context learning appears, where performance improves with more shots and does not decline after reaching its peak. As the gap between testing causal chains and training child chains widens, the performance decrease after one shot becomes evident, indicating the influence brought by OOD tasks.

---

### Meta-Review · Area_Chair_7pps · 2024-12-23

**Metareview:**

All reviewers made strong, positive comments about the submission, with common threads commenting on it being (i) interesting and important research (ii) technical soundness and particularly creative and interesting analysis (iii) the well-conceived nature of the FTCT task. While there were some open questions of how findings transfer to larger models and/or pretrained large-scale transformers, the authors did convincingly argued for the randomly initialised synthetic data as a valid use case while also scaling up to at least the hundreds of millions of parameters scale (GPT-2) level to check for empirical differences in model size.

While I would have liked to see a strong champion for this submission under the reviewers for the submission, the unanimity in considering the submission above the acceptance threshold, together with the lack of deeper criticism, persuades me that this submission is suitable for publication at ICLR. I would like to ask the authors to take any remaining feedback into account for the camera-ready version.

**Additional Comments On Reviewer Discussion:**

We saw a very engaging and lively discussion between reviewers and authors, with the authors, in particular, giving very well-argued and strong rebuttals to some of the authors' criticisms. The author's arguments around the valid goals and scope of this manuscript (e.g. explicitly not wanting to test large pre-trained language models and instead focusing on new, randomly initialised models) positively influenced my own weighting of the pros and cons of the submission.

---

### Decision · Program_Chairs · 2025-01-22

Accept (Poster)